

# Digitized endocasts and brains: a perspective on measurements and historical analyses of the evolution of 172 fossil and extant amniote specimens

Harry J. Jerison[1,*,†], Catherine M. Early[2], Andrew A. Farke[3] and Ashley C. Morhardt[4,*]

[1] Department of Psychiatry and Biobehavioral Sciences, University of California, Los Angeles, CA, United States of America
[2] Biology Department, Science Museum of Minnesota, Saint Paul, MN, United States of America
[3] Raymond M. Alf Museum of Paleontology, Claremont, CA, United States of America
[4] Department of Neuroscience, Washington University School of Medicine in St. Louis, St. Louis, MO, United States of America

[*] These authors contributed equally to this work.
[†] Deceased.

Corresponding author
Ashley C. Morhardt,
amorhardt@wustl.edu

## ABSTRACT

This perspective paper is intended to stimulate future research and discussion of brain evolution in amniotes by sharing 172 digitized endocasts of extinct and extant species spanning 60 million years. Using 3D digital surface scans of physical (*e.g.*, latex, plaster, resin) endocasts, we measured and compared relative endocranial volumes from dozens of extinct amniote taxa with those (endocasts or brain surface scans) of relevant extant species. Additionally, we offer calculated Encephalization Quotients and neocorticalization from digitized endocasts. Using historical methods of analysis, we find that, on average, neocortocalization of mammals increased over time, which is in agreement with recently published findings. Results also showed that, about 60 million years ago, mammalian neocorticalization averaged about 20%, increasing to a present average of 50%, and reaching a maximum of about 80% in primates within the past 10 million years. These results potentially redefine the allometric boundary between mammals and reptiles and confirm that measurements on a single species can adequately represent the brains of the entire species. We encourage other researchers to use our data, results, and conclusions as a springboard for more updated analyses.

# INTRODUCTION

## Endocasts and their utility

Cranial endocasts are casts molded by the endocranial cavity of the skull, either naturally through fossilization of interred material, artificially with materials such as plaster or latex, or virtually with computed tomography (CT) scanning and segmentation. Endocasts provide a powerful window into deep time for studying neuroanatomy and brain evolution.
Tilly Edinger, the founder of paleoneurology, described endocasts as fossilen Gehirne (*Edinger, 1929*), or "fossil brains," and by the time of her death in 1967, had compiled a then-comprehensive annotated bibliography of over a thousand vertebrate fossil genera and their cranial cavity endocasts (*Edinger, 1975*).

Throughout her career, Edinger noted that bird and mammal endocasts largely mirror brains in size and appearance, although the narrow space around a brain containing meninges, blood vessels, and cerebrospinal fluid add to the surface area and volume of an endocast compared to a brain. Indeed, endocasts are often faithful proxies for brain size, especially in extant, adult vertebrate groups such as mammals (*Haight & Nelson, 1987*; *De Miguel & Henneberg, 2001*) and birds (*e.g.*, *Iwaniuk & Nelson, 2002*; *Watanabe et al., 2019*).

## Neocorticalization and encephalization in mammals

The cerebral cortex (*i.e.,* neopallium, isocortex; *Butler & Hodos, 2005*) is an outer layer of forebrain consisting of layers of nerve cells. This layered organization of cells is a unique feature of the brain in all living mammals. In this paper, we measured neocorticalization, or the comparative increase in neocortex relative to other brain structures, as the increase in surface area of cerebral cortex dorsal to the rhinal fissure relative to the total cortical surface area (Fig. 1). Encephalization—the evolutionary increase in brain complexity or relative size reflecting environmental adaptations—has been proposed historically as a general phenomenon in many species of mammals and birds that is more or less correlated to natural selection and independent of their phylogenetic details (*e.g.*, *Jerison, 1973*; *Jerison, 1977*; *Boddy et al., 2012*; *Ksepka et al., 2020*; *Smith, 2022* and sources therein; *Van Schaik et al., 2023*; but for mammals, see *Burger et al., 2019*; *Smaers et al., 2021*). For over a century, brain-body allometry, along with the generally concerted evolution of brain regions (study: *Finlay & Darlington, 1995*; "concerted evolution" term applied: *Barton & Harvey, 2000*), have explained a large portion of brain size variation across vertebrates (*e.g.*, *Snell, 1892*; *Montgomery, Mundy & Barton, 2016*; *Moore & De Voogd, 2017*; *Kotrschal et al., 2017*; but see: *Smaers & Soligo, 2013*; *Barton & Montgomery, 2019*; *Willemet, 2019*; *Willemet, 2020*). Many birds and mammals have evolved a substantially larger brain for a given body size, or larger encephalization, compared to other vertebrates (*e.g.*, *Jerison, 1971*; *Iwaniuk, Dean & Nelson, 2005*; *Emery, 2006*; *Franklin et al., 2014*; *Ksepka et al., 2020*; *Smaers et al., 2021*).

The measurement of neocorticalization evolution is an outstanding example of a quantitative analysis made possible by digitizing data. Surface scanning and software analysis enable direct measurements from virtual models of natural or physical endocasts. This study exploits digitization technology to review 129 three-dimensional (3D) models of fossil endocasts and compare them with the endocasts and brains of 43 extant specimens. Notably, quality comparisons of human-made (*e.g.*, plaster, resin) and natural endocasts are not always possible due to breaks or lack of preservation in natural endocasts. We are unsure exactly of how our late lead author (HJJ) handled these instances—adding yet another caveat to our results. We caution future researchers to be careful in their own comparisons when using our data. For more information on specific endocasts, see Supplemental Information 1A and 1B.
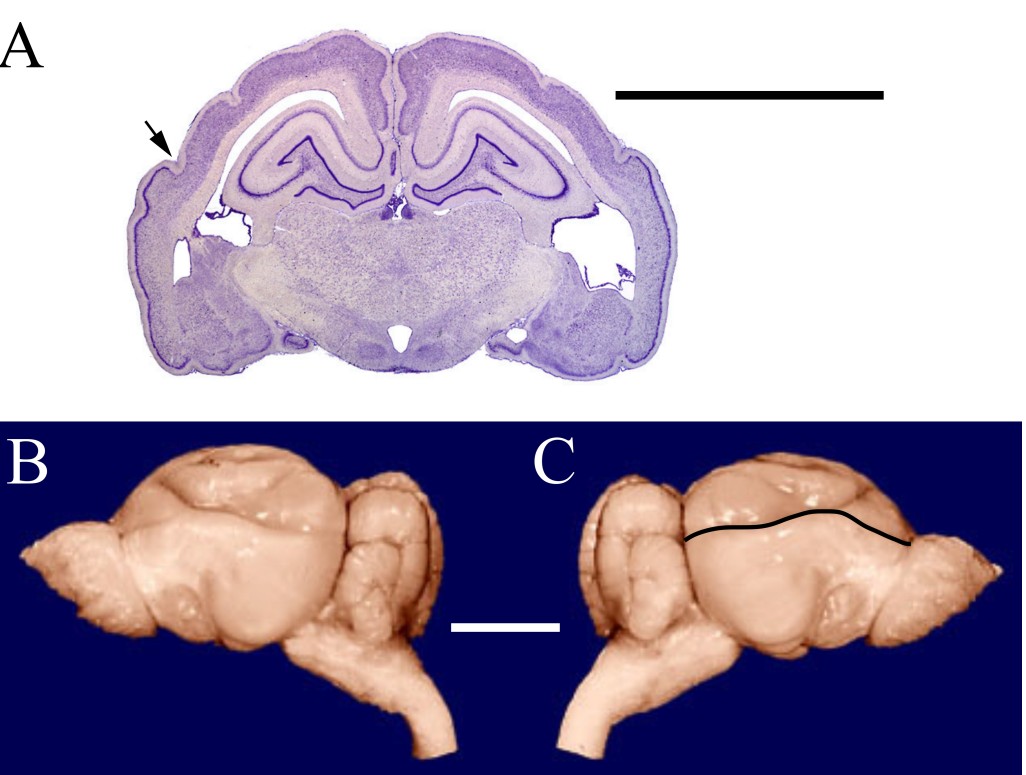

**Figure 1** **Armadillo rhinal fissure.** Armadillo brain (*Dasypus novemcinctus*; NMHM Vertebrates WISC 40-465). (A) Coronal section through the brain showing the rhinal fissure marked with an arrow, neocortex dorsal to the fissure, and paleocortex with dark lamina II ventral to the fissure. (B) Lateral views of the same brain, with (C) rhinal fissure indicated by black line. Images reproduced with permission from http://brainmuseum.org, with copyright retained by said party. Scale bars = one cm.

## Importance and limitations of this perspective study

Although components of the dataset have been presented in other venues (*e.g.*, *Long, Bloch & Silcox, 2015*, using data provided by *Jerison, 2012*), this is the most comprehensive collection to date. Equally importantly, we provide figures for nearly all of the endocasts, many for the first time in the literature. We hope that these illustrations will be a useful resource for future work on endocranial anatomy.

Lastly, we are aware of widespread leaps in the statistical calculation and determination of evolutionary trends collectively known as phylogenetic comparative methods ((PCM); *e.g.*, *Harmon, 2019*; *Adams & Collyer, 2019*). We are also aware that many of these methods are more robust and appropriate for application to studies of brain (or brain region) *vs.* body size in vertebrates (*e.g.*, see review in *Striedter & Northcutt, 2019*). The use of these methods provides rich and potentially more nuanced and/or more accurate analysis of datasets than the one presented in our paper. However, because this paper serves as the culmination of decades of work by our now-late lead author (HJJ), who passed while this paper was in review, we have elected to present methods consistent with his previous

studies for the sake of comparison. We eagerly anticipate that future work will incorporate all or parts of the dataset into more advanced and sophisticated analyses.

## MATERIALS AND METHODS

### Endocast specimens

The first author obtained the majority of the natural and latex endocasts used in this study from the Radinsky Collection at the Field Museum of Natural History (FMNH) in Chicago, most of which were collected and prepared by the late Professor Len Radinsky and catalogued by Collections Manager William Simpson. We also scanned specimens from following collections: University of Adelaide (Adelaide), Victoria, Australia; American Museum of Natural History (AMNH), New York; American Museum of Natural History (AMNH FAM)—Frick Collection at AMNH; The Natural History Museum (NHMUK), London; Carnegie Museum of Natural History (CM), Pittsburgh, Pennsylvania; National Museum of Health and Medicine Defense Health Agency Neuroanatomical Collections Division (NMHM WISC), Silver Spring, Maryland (formerly the University of Wisconsin and Michigan State Comparative Mammalian Brain Collections in Madison, Wisconsin); Falk Collection (Falk), now deposited with Mammals at AMNH; Jura-Museums Eichstaett (JME), Germany; Natural History Museum of Los Angeles County (LACM), California; The Museum of Comparative Zoology (MCZ) at Harvard University, Cambridge, Massachusetts; Muséum National d'Histoire Naturelle (MNHN), Paris; University of California Museum of Paleontology (UCB), California; Museum of the Rockies (MOR), Bozeman, Montana; National Museum of Victoria (NMV), Melbourne, Australia; Canadian Museum of Nature (NMC), Ottawa; Raymond M. Alf Museum of Paleontology at the Webb Schools (RAM), Claremont, California; Senckenberg Natural History Museum (SNHM), Frankfurt, Germany; Texas Memorial Museum (UT) and (TMM), Austin, Texas; United States National Museum (USNM), Washington, DC; University of the Witwatersrand (WITS), Johannesburg, South Africa; and Yale Peabody Museum of Natural History (YPM), New Haven, Connecticut.

Most endocasts in the dataset derive from fossilized mammals, but some endocasts and braincasts (*i.e.,* casted replicas of the actual brain) represent extant species. Four endocasts are from extinct dinosaurs. Prior to our digitization work (see below), and across several decades, collections staff or other researchers prepared non-fossilized endocasts using plaster, latex, thermoplastic, or virtual segmentation. In all cases, the cranial cavity served as a mold for each endocast, with little difference in most cases between endocasts prepared with plaster or latex from a cranial cavity or filled with fossilized matrix preserved when the skull eroded. Idenfication of features on the endocasts best matches the convention put forth by *Macrini, Leary & Weisbecker* (*2022*, see their fig 11.3).

### Digitization

HJJ digitized endocasts using surface scanning with a Cyberware Model M15 Scanner using Headus 3D Tools software (http://www.headus.com.au), which creates endocast images as complete 360° rotation producing a full scan. After performing a first set of scans, HJJ placed the object in a different orientation to expose areas previously hidden from the laser

beam before acquiring a further set of scans. HJJ then merged the final set of 16 scans into a 3D digitized image. Again, we share that there are instances in the dataset of incomplete natural endocasts, and we are unsure of how our late lead author (HJJ) handled these cases while collecting and analyzing data.

Headus outputs the following image measurements: volume in mm$^3$, surface area in mm$^2$, and length in mm. HJJ measured endocast lengths from the anterior tip of the forebrain or olfactory bulbs to the posterior hindbrain at the end of the medulla. Surface regions of the endocast, such as olfactory bulbs or neocortex, were virtually measured by HJJ by selecting the region of interest in Cyberware (now Headus 3D) and then measuring the selected region with the CySize tool. HJJ recorded measurements in centimeter-gram-seconds (cgs) unless otherwise noted. Most of the 3D models of endocasts are freely available for viewing and download on MorphoSource at the discretion of collections staff. Table S2 is a list of available endocast and MorphoSource DOIs.

## Neocorticolization

Unlike other gyri and sulci, endocast rhinal fissures are a reliable landmark of the ventral boundary of neocortex. It is a superficial landmark that is visible on the brains of living mammals (*Welker, 1990*). The rhinal fissure differentiates neocortex dorsally from paleocortex ventrally (*Kappers, 1909*). We define measurements of the surface area of forebrain dorsal to the rhinal fissure as neocortex (see example, Fig. 1).

## Brain-body allometry

Brain-body allometry relates endocast volume (representing brain size) to estimated body size. Brain-body allometry was calculated using head+body length as an independent variable and brain size as the dependent variable, as described previously (*Jerison, 1973*; *Jerison, 1991*; *Jerison, 2001a*; *Jerison, 2001b*; *Jerison, 2002*). The historical Jerison equation was used here and is consistent with previous analyses of HJJ, but we note the existence of more recent published equations in the literature (*e.g.*, *Burger et al., 2019*; *López-Torres et al., 2024*). Briefly, the power function for the regression of body size relative to body length, which reflects a surface area (the skin) for graphing against brain size, was as in *Jerison (1973)*:

$$P = 0.021\, L^{3.03} \tag{1}$$

where $P$ is body mass or volume, and $L$ is body length in the cgs system. In a few species for which accurate models of the body were prepared, body volume was determined from 3D scans of the model. We then fit brain-body size relationships in different species measured logarithmically by linear regression:

$$log\ E = \alpha\, log\ P + log\ b \tag{2a}$$

or the power function:

$$E = b\, P\alpha \tag{2b}$$

where $E$ is brain size (from the French, *encephale*); $P$ is body size (from the French, *poids*); $\alpha$ is the regression line's slope, and $log\ b$ is its $y$ intercept. $E$ and $P$ were in the same units

(ml and grams or liters and kilograms) in Eqs. (2a) and (2b) for easy interpretation. For extant species, HJJ either weighed the brains and bodies of them or derived the volumes (HJJ: no correction factor (*e.g.*, *Stephan, Frahm & Baron, 1981*) applied) from log brain size as a function of log body size graphed. Bivariate statistical regression analysis of the log brain-body relationship for vertebrate classes can estimate $\alpha$ empirically. The empirical encephalization quotient (*EQ*) in brain size in each species is its residual from the regression, but *EQ* can be different if the allometric factor is determined by theoretical analysis rather than regression. Therefore, we took *EQ* as the residual relative to Eqs. (2a) or (2b), with $\alpha$ a theoretical constant of exactly 2/3. In this way, the allometric equation transforms the 3D information of the body into a 2D map created by the brain (*Jerison, 2001a*; *Jerison, 2001b*; *Jerison, 2002*). Extensive discussion of body mass estimation sources and methods is available in File S1A.

## RESULTS AND DISCUSSION

### Important notes

With the exception of measured values taken directly from 3D surface endocasts, the following results and discussions stem from more traditional methods (*e.g.*, phylogenetically un-informed mathematical regressions of raw data) in that they do not incorporate our field's most recent understanding of: (1) phylogenetic comparative methods, (2) updated equations and appropriate applications of encephalization quotient (EQ), and (3) methods for estimating brain and brain region sizes from 2D images (*e.g.*, more refined computational models, extrapolation by artificial intelligence) and in extinct non-mammalian amniotes (*e.g.*, correction factors for brains that do not completely fill the endocranial cavity). As such, the calculated values presented here, as well as their bearing on discussions of evolutionary patterns, will require re-analyses outside the scope of this paper to ensure that they are as accurate as possible.

We feel it is important to publish the traditional findings here for two reasons. First, we wish to acknowledge faithfully the scientific contributions of our late lead author. Second, we wish to allow for any direct comparisons between the findings in this paper and those previously published by Jerison (*e.g.*, *1973*; *2007*; *2012*). We encourage those who read and/or may cite this paper to do so with this note's context in mind. We acknowledge that our dataset is non-comprehensive and that many additional published vertebrate endocasts exist. We encourage other researchers to synthesize and reanalyze all data now available. Some additional published datasets can be found in: *Dechaseaux (1958)*, *Silcox, Benham & Bloch (2010)*, *Rowe, Macrini & Luo (2011)*, *Orliac & Gilissen (2012)*, *Long, Bloch & Silcox (2015)*, *Harrington et al. (2016)*, *Bertrand & Silcox (2016)*, *Bertrand et al. (2019)*, *Benoit et al. (2019)*, *Bertrand et al. (2020)*, and *Maugoust & Orliac (2021)*, *Arnaudo & Arnal (2023)*, and Betrand et al. (*2022*; *2024b*). Additionally, the following recent papers consider relative brain size of vertebrates through time: *Tsuboi et al. (2018)*, *Ksepka et al. (2020)*, *Smaers et al. (2021)*.

Lastly, admittedly, due to the loss of our first author and his insights, we are unable to track down and cite all resources/methods involved in the compilation of body masses reported in Table 1.

Jerison et al. (2025), *PeerJ*, DOI 10.7717/peerj.19826

**Table 1** Measurements of endocasts and brains of fossils and living mammals.

| Species/Genera: | S cm$^2$ | OB cm$^2$ | S-OB cm$^2$ | NC cm$^2$ | L cm | MYA | E ml or g | P g | %NC | EQ | Taxon |
|---|---|---|---|---|---|---|---|---|---|---|---|
| **118 FOSSIL SPECIMENS INCLUDING 7 PRIMATES** | | | | | | | | | | | |
| *Adapis parisiensis* NHMUK M1340 (= FMNH PM 59259) | 24.95 | 1.42 | 23.53 | 10.17 | 4.7 | −34.1 | 8.2 | 1,600 | 43.23 | 0.5 | Primates |
| *Adapis parisiensis* FMNH PM 59275 Le Gros Clark | 26.52 | 1.55 | 24.97 | 13.25 | | −34.1 | 7.85 | 1,600 | 53.06 | 0.48 | Primates |
| *Adinotherium ovinum* FMNH P 12986 | 155.35 | 7.99 | 147.35 | 49.07 | 10.2 | −15.5 | 111.04 | 120,000 | 33.30 | 0.38 | Notoungulata |
| *Amynodon advenus* YPM VP 11453 (= FMNH PM 59231) | 239.39 | 12.41 | 226.98 | 75.77 | | −46 | 180.99 | 263,000 | 33.38 | 0.37 | Perissodactyla |
| *Anoplotherium commune* NHMUK PV M 3753 | 129.18 | 6.88 | 122.31 | 34.83 | 10.4 | −34.1 | 78.37 | 82,000 | 28.48 | 0.35 | Artiodactyla |
| *Apterodon macrognathus* FMNH PM 57147 | 115.74 | 0 | 107.68 | 34.33 | | −37 | 77.54 | 47,0075 | 31.88 | 0.5 | Carnivora |
| *Archaeolemur majori* AMNH FM 30007 (= FMNH PM 59258) | 121.19 | 1.88 | 119.3 | 76 | 7.6 | −0.01 | 95.89 | 17,000 | 63.70 | 1.21 | Primates |
| *Archaeotherium mortoni* YPM VPPU 10908 (= FMNH PM 59061) | 204 | 12 | 192 | 49 | 16.2 | −32 | 168.74 | 230,000 | 25.52 | 0.37 | Artiodactyla |
| *Arctocyon primaevus* MNHN F CR700 | 25.89 | 4.29 | 21.6 | 4.86 | 4.9 | −58 | 7.14 | 16,144 | 22.49 | 0.09 | Procreodi |
| *Arctodus simus* FMNH PM 59022 (attributed to LACM) | 542.37 | 44.34 | 498.03 | 296.05 | 16.8 | −0.03 | 654.14 | 525,763 | 59.45 | 0.84 | Carnivora |
| *Argyrocetus joaquinensis* USNM 11996 | | | | | 13.3 | −23 | 629.94 | 80,100 | | 3.01 | Cete |
| *Arsinotherium zitelli* NHMUK PV M 8539 | | | | | 18.6 | 33 | 926.6 | 1,500,000 | | 0.59 | Embrithopoda |
| *Aulophyseter morricei* USNM 11230 | | | | | 16.7 | −16 | 2,246 | 8,508,540 | | 0.45 | Cete |
| *Australopithecus africanus* Taung 1 | 241.72 | | 241.72 | 194.69 | 10.8 | −3.5 | 440 | 40,000 | 80.54 | 3.13 | Primates |
| *Australopithecus robustus* SK1585 | 356.35 | | 356.35 | 275.03 | 11.6 | −2.5 | 530 | 40,000 | 77.18 | 3.78 | Primates |
| *Bathygenys reevesi* TMM TXVP 40209-431 | 33.46 | 2.35 | 31.11 | 8.71 | 5 | −37 | 12.08 | 6,795 | 28.00 | 0.28 | Artiodactyla |
| *Borhyaena tuberata* FMNH P 13266 | 82.09 | 13.08 | 69.01 | 17.39 | 7.4 | −17 | 43.05 | 24,600 | 25.20 | 0.42 | Marsupialia |

Jerison et al. (2025), *PeerJ*, DOI 10.7717/peerj.19826

**Table 1** (*continued*)

| Species/Genera: | S cm² | OB cm² | S-OB cm² | NC cm² | L cm | MYA | E ml or g | P g | %NC | EQ | Taxon |
|---|---|---|---|---|---|---|---|---|---|---|---|
| *Aenocyon dirus* LACMHC 2300-82 | 207.83 | 15.12 | 192.71 | 114.41 | 11.7 | −0.03 | 181 | 80,000 | 59.37 | 0.81 | Carnivora |
| *Canis latrans* endocast LACMHC 3200-7 | 117.79 | 7.79 | 110 | 73.97 | | 0.03 | 97.85 | 15,000 | 67.25 | 1.34 | Carnivora |
| *Carpocyon webbi* AMNH FM 61328 (= FMNH PM 58964) | 142.9 | 13.12 | 129.78 | 61.35 | 10.6 | −13 | 100.04 | 32,000 | 47.27 | 0.83 | Carnivora |
| *Cebochoerus lacustris* FMNH PM 59051 | 34.1 | 3.84 | 30.26 | 7.01 | 5.6 | −38 | 11.9 | 8,000 | 23.17 | 0.25 | Artiodactyla |
| *Chadronia margaretae* AMNH FM 109412 (= FMNH PM 57129) | 66.47 | 6.09 | 60.38 | 23.48 | 7.7 | −35 | 28.86 | 7,500 | 38.89 | 0.63 | Cimolesta |
| *Cormohipparion occidentale* AMNH FM 71886 (= FMNH PM 59220) | 333.71 | 24.39 | 309.32 | 184 | 11.2 | −10 | 363.94 | 151,000 | 59.49 | 1.07 | Perissodactyla |
| *Coryphodon hamatus* YPM VP 11331 (= FMNH PM 59241) | 140.94 | 13.66 | 127.28 | 24.34 | 11.8 | −52 | 90.6 | 394,000 | 19.12 | 0.14 | Cimolesta |
| *Cynodictis cayluxi* FMNH PM 59013 | 33.29 | 2.5 | 30.79 | 12.54 | | −34 | 11.6 | 2,800 | 40.73 | 0.49 | Carnivora |
| *Cynohyaenodon cayluxi* FMNH PM 57153 | 32.46 | 4.58 | 27.88 | 8.32 | | −34 | 11.04 | 5,392 | 29.83 | 0.3 | Creodonta |
| *Daphoenus vetus* FMNH PM UM1 | | | 80.89 | 39.82 | | −32 | 46.87 | 24,000 | 49.22 | 0.47 | Carnivora |
| *Daphoenus vetus* FMNH PM UM1 | | | 84.96 | 39.1 | | −32 | 42.61 | 24,000 | 46.02 | 0.43 | Carnivora |
| *Desmathyus (Hesperhyus)* CM VP 1423 (=FMNH M 59066) | 122.43 | 13.59 | 108.84 | 33.38 | 8.3 | −19.5 | 71.27 | 11,291 | 30.67 | 1.18 | Perissodactyla |
| *Dinictis felina* SDSM 2431 (= FMNH PM 58866) | 88.4 | 4.9 | 83.23 | 42.14 | 6.8 | −32 | 60.1 | 37,000 | 50.63 | 0.45 | Carnivora |
| *Durodon atrox* NHMUK PV M 10173 b | | | | | 13.1 | −40 | 459.55 | 393,540 | | 0.71 | Cete |
| *Enaliarctos* sp. FMNH PM 57161 | 132 | | 132 | 80.64 | 7.2 | −22 | 118 | 82,000 | 61.00 | 0.52 | Carnivora |
| *Eomoropus amarorum* AMNH FM 5096 (= FMNH PM 59182) | 80.41 | 1.76 | 78.66 | 23.46 | 6.8 | −44 | 36.89 | 40,000 | 29.83 | 0.26 | Perissodactyla |
| *Eporeodon socialis* YPM VP 13118 (= FMNH PM 59076) | 78.72 | 6.92 | 71.8 | 23.67 | 7.6 | −23 | 41.79 | 19,400 | 32.96 | 0.48 | Artiodactyla |
| *Equus occidentalis* LACMHC 3500-17 | 573.34 | 32.01 | 541.33 | 317.55 | | −0.03 | 869 | 550,000 | 85.19 | 1.08 | Perissodactyla |
| *Eusmilus bidentatus* FMNH PM 58871 | 70.19 | 4.87 | 65.32 | 32.92 | 7.2 | −32 | 40.12 | 35,000 | 50.40 | 0.31 | Carnivora |
| *Paramylodon harlani* LACMHC 1717-33 | 469.74 | 46.57 | 423.17 | 129.54 | | −0.03 | 501.94 | 1,100,000 | 30.61 | 0.39 | Xenarthra |
| *Halitherium schinzi* SMF M 3921 | 298.77 | | 298.77 | 94.59 | 12.2 | −25 | 267 | 250,000 | 31.66 | 0.56 | Sirenia |
| *Hapalops* sp. LACM | 91.8 | 7.29 | 84.51 | 30.12 | 7.8 | −17 | 54.7 | 49,000 | 35.64 | 0.34 | Xenarthra |
| *Hemicyon cf. barbouri* AMNH FM 25530 (= FMNH PM 59030) | 251.22 | 8.25 | 242.97 | 97.95 | 12.6 | −10 | 199.28 | 82,000 | 40.31 | 0.88 | Carnivora |
| *Heptodon* sp. FMNH PM 59193 | 89.03 | 11.42 | 77.6 | 20.96 | 9.2 | −52 | 42.67 | 24,000 | 27.00 | 0.43 | Perissodactyla |
| *Hesperocyon gregarius* FMNH PM 58989 | 37.81 | 4.05 | 33.75 | 17.58 | 4.9 | −32 | 18.8 | 3,000 | 52.08 | 0.75 | Carnivora |
| *Homalodotherium cunninghami* FMNH PM 59291 | 284.85 | 15.96 | 268.89 | 70.22 | 14.3 | −17.5 | 227.3 | 400,000 | 26.12 | 0.35 | Notoungulata |

Jerison et al. (2025), *PeerJ*, DOI 10.7717/peerj.19826

**Table 1** (*continued*)

| Species/Genera: | S cm² | OB cm² | S-OB cm² | NC cm² | L cm | MYA | E ml or g | P g | %NC | EQ | Taxon |
|---|---|---|---|---|---|---|---|---|---|---|---|
| *Homotherium* sp. AMNH FM 95297 (= FMNH PM 58891) | 243.67 | 19.22 | 224.45 | 128.35 | 11.2 | −1.5 | 192.5 | 200,000 | 57.18 | 0.47 | Carnivora |
| *Hoplophoneus primaevus* UM2 PF | 73.11 | | 73.11 | 28.48 | 6.7 | −32 | 42.67 | 35,000 | 38.96 | 0.33 | Carnivora |
| *Hoplophoneus primaevus* USNM Paleobiology V 22538 | 79.38 | | 79.38 | 32.57 | 6.7 | −32 | 49.47 | 35,000 | 41.03 | 0.39 | Carnivora |
| *Hyaenodon* FMNH P 12723 | 114.55 | 8.31 | 106.23 | 32.41 | | −50 | 67.37 | 60,000 | 30.51 | 0.37 | Carnivora |
| *Hylomeryx quadricuspis* CM VP 2915 (= FMNH PM 59055) | 26.78 | 0 | 26.78 | 6.71 | 4.3 | −42 | 9.14 | 6,000 | 25.04 | 0.23 | Artiodactyla |
| *Hyrachyus modestus* YPM VP 11082 (= FMNH PM 59240) | 115.32 | 8.47 | 106.85 | 29.75 | 8.7 | −51.7 | 68.95 | 100,000 | 27.85 | 0.27 | Perissodactyla |
| *Hyracotherium* AMNH FM 55268 (= FMNH PM 59207 ) | 57.33 | 8.53 | 48.8 | 10.66 | 6.6 | −52.9 | 24.16 | 10,700 | 21.84 | 0.41 | Perissodactyla |
| *Isectolophus latidens* AMNH FM 12222 (= FMNH PM 59179) | 51.05 | 8.29 | 42.75 | 9.03 | 6.6 | −47 | 20.37 | 11,600 | 21.13 | 0.33 | Perissodactyla |
| *Leontinia gaudryi* FMNH P 13285 | 346.9 | 28.2 | 318.7 | 106.48 | 15 | −25 | 356.91 | 450,000 | 33.41 | 0.51 | Notoungulata |
| *Leptauchenia decora* AMNH FM 627 (= FMNH PM 59074) | 50.78 | 5.02 | 45.76 | 10.35 | 6.3 | −31 | 21.95 | 39,300 | 22.61 | 0.16 | Artiodactyla |
| Harry Jerison's personal collection | 15.11 | 2.08 | 13.03 | 1.87 | 3.2 | −32 | 3.61 | 500 | 14.32 | 0.48 | Leptictida |
| *Leptocyon* sp. FMNH PM 58961 | 36.41 | 2.6 | 33.81 | 15.95 | 5.2 | −20 | 14.14 | 3,260 | 47.16 | 0.54 | Carnivora |
| *Leptolambda schmidti* FMNH P 26075 | 143.3 | 17.77 | 125.52 | 6.46 | 8.7 | −56 | 98.13 | 620,000 | 5.15 | 0.11 | Cimolesta |
| *Leptolambda (Barylambda) schmidti* FMNH P 15573 | 126.79 | 0 | 126.79 | 9.71 | | −56 | 85.22 | 620,000 | 7.66 | 0.1 | Cimolesta |
| *Megalonyx jeffersoni* Harry Jerison's personal collection | 302.84 | 27.05 | 275.75 | 103.69 | 12.6 | −0.03 | 332.78 | 370,000 | 37.60 | 0.54 | Xenarthra |
| *Meniscotherium robustum* USNM V 19509 | | | | | | −53 | 14.8 | 6,500 | | 0.35 | Condylarthra |
| *Megacerops coloradensis* FMNH PM 59199 (possibly from YPM VP 12010) | 521.47 | 36.62 | 484.85 | 164.37 | 17.1 | −34 | 750 | 4,000,000 | 33.90 | 0.25 | Perissodactyla |
| AMNH FM 71150 (= FMNH PM 59208) | 248.43 | 18.23 | 230.2 | 119.03 | 12.4 | −15 | 231.79 | 105,700 | 51.71 | 0.86 | Perissodactyla |
| *Merychippus severus* LACM (CIT) 2929 | 296.34 | 65.09 | 231.25 | 106.82 | 14.5 | −15.5 | 258.77 | 110,000 | 46.19 | 0.94 | Perissodactyla |
| *Merycochoerus proprius* AMNH FM 43016 A (= FMNH PM 59081) | 142.44 | 9.53 | 132.91 | 39.48 | 10.2 | −18 | 95.74 | 122,000 | 29.70 | 0.32 | Artiodactyla |
| *Merycoidodon culbertsoni* FMNH PM UM3 | 78.05 | 0 | 78.05 | 22.57 | 7.2 | −32 | 47.25 | 68,000 | 28.92 | 0.24 | Artiodactyla |
| *Mesatirhinus junius (Eobasileus)* YPM VPPU 10041 (= FMNH PM 59197) | 227.32 | 14.56 | 212.76 | 58.23 | 12.3 | −46.5 | 189.5 | 350,000 | 27.37 | 0.32 | Perissodactyla |
| *Mesatirhinus petersoni* AMNH FM 1509 (= FMNH PM 59196) | 221.06 | 24.68 | 196.38 | 37.17 | 13.2 | −46.5 | 146.9 | 350,000 | 18.93 | 0.25 | Perissodactyla |
| *Mesocyon coryphaeus* AMNH FM 6946 (= FMNH PM 58979) | 71.01 | 4.99 | 66.02 | 26.37 | 7.2 | −25 | 36.55 | 10,000 | 39.94 | 0.66 | Carnivora |
| *Mesohippus bairdi* AMNH FM 9814 (= FMNH PM 59221) | 127.48 | 9.86 | 117.62 | 48.89 | 9.6 | −32 | 86.42 | 28,500 | 41.56 | 0.77 | Perissodactyla |
| *Mesonyx obtusidens* YPM VP 13141 (= FMNH PM 57139) | 133 | 0 | 133 | 31 | 8.2 | −48 | 96 | 65,000 | 23.31 | 0.49 | Cete |

Jerison et al. (2025), *PeerJ*, DOI 10.7717/peerj.19826

## Table 1 (*continued*)

| Species/Genera: | S cm² | OB cm² | S-OB cm² | NC cm² | L cm | MYA | E ml or g | P g | %NC | EQ | Taxon |
|---|---|---|---|---|---|---|---|---|---|---|---|
| *Mixtotherium cuspulatum* FMNH PM 59052 | 45.93 | 0 | 45.93 | 12.74 | 6.7 | −40 | 21.04 | 6,000 | 27.73 | 0.53 | Artiodactyla |
| *Moeritherium lyonsi* NHMUK PV M 9176 b | 265.93 | 34.58 | 231.35 | 78.26 | 11.6 | −37 | 233.33 | 394000 | 33.83 | 0.36 | Proboscidea |
| *Mustelictes piveteaui* FMNH PM 58907 | 35.32 | 3.25 | 32.07 | 9.04 | 4.4 | −22 | 12.61 | 12,973 | 28.20 | 0.19 | Carnivora |
| *Mylodon* LACM 157696 | 420.76 | 21.67 | 399.1 | 119.19 | 15.8 | −0.01 | 514.88 | 1,100,000 | 29.86 | 0.4 | Xenarthra |
| *Necrolemur antiquus* YPM VP 18302 (= FMNH PM 59261) | 39.18 | 2.08 | 37.1 | 14.04 | | −37 | 5.05 | 320 | 37.84 | 0.9 | Primates |
| *Nesodon umbricatus* FMNH P 13076 | 253.29 | 9.28 | 244.01 | 72.74 | 12.3 | −17 | 180.06 | 250,000 | 29.81 | 0.38 | Notoungulata |
| *Notharctus* sp. FMNH PM 59264 | 40.57 | 2.32 | 38.25 | 10 | 5 | −47 | 15.38 | 4,200 | 26.14 | 0.48 | Primates |
| *Nothrotheriops shastensis* LACMHC 1800-6 | 279.43 | 40.35 | 239.08 | 98.87 | 11.8 | −0.03 | 277.12 | 320,000 | 41.35 | 0.49 | Xenarthra |
| *Orthocynodon (Amynodon)* sp. YPM VPPU 10145 (= FMNH PM 59177) | 140.52 | 21.25 | 119.27 | 30.82 | 9 | −50 | 93.99 | 150,000 | 25.84 | 0.28 | Perissodactyla |
| *Oxydactylus* sp. FMNH P 12117 | 131.44 | 9.24 | 122.2 | 51.01 | 10.3 | −19.5 | 86.65 | 250,000 | 41.74 | 0.18 | Perissodactyla |
| *Pachyaena ossifraga* YPM VPPU 14708 | 88.51 | 7.95 | 80.56 | 8.4 | 9.2 | −53 | 32.66 | 65,000 | 10.43 | 0.17 | Cete |
| *Pachylemur (Lemur) insignis* FMNH PM 59253 | 80.37 | 0 | 80.37 | 49.61 | 6.4 | −0.01 | 57.38 | 10,000 | 61.73 | 1.03 | Primates |
| *Palaeopropithecus maximus* FMNH PM 59250 | 134.93 | 1.43 | 133.5 | 72.73 | | −0.01 | 108.33 | 50,000 | 54.48 | 0.67 | Primates |
| *Palaeosyops* sp. FMNH PM 59198 | 288 | 28.24 | 259.76 | 40.2 | 14.2 | −51.7 | 195.31 | 191,000 | 15.48 | 0.49 | Perissodactyla |
| *Panthera atrox* LACMHC 2900-1 | 326.92 | 21 | 305 | 166.65 | 13.8 | −0.03 | 338.43 | 325,000 | 54.64 | 0.6 | Carnivora |
| *Paracynarctus sinclairi* AMNH FM 61009 (= FMNH PM 58973) | 97.54 | 8.53 | 89.01 | 39.56 | 8.2 | −15 | 55.93 | 12,263 | 44.44 | 0.88 | Carnivora |
| *Paratomarctus euthos* AMNH FM 61074 | 87.6 | 8 | 79.6 | 35.6 | 7.6 | −11 | 56.3 | 10,900 | 44.72 | 0.95 | Carnivora |
| *Patriomanis americana* AMNH FM 78999 (= FMNH PM 57103) | 33.86 | 4.21 | 29.65 | 5.23 | | −34.7 | 11.21 | 3,000 | 17.65 | 0.45 | Cimolesta |
| *Phenacodus primaevus* AMNH FM 4369 (= FMNH PM 59042) | 72.75 | 10.33 | 62.42 | 10.01 | 7.7 | −54 | 30.82 | 82,000 | 16.04 | 0.14 | Condylarthra |
| *Plagiolophus minor* Harry Jerison's personal collection | 58.6 | 9.64 | 48.96 | 29.51 | | −34 | | | 60.27 | | Perissodactyla |
| *Platygonus compressus* CM VP 12888 (= FMNH PM 59058) | 138.82 | 9.52 | 129 | 74.44 | 9.7 | −0.3 | 130 | 130,000 | 57.71 | 0.42 | Artiodactyla |
| *Plesiogale paragale* NMB M.A.4641 | 44.19 | 3.22 | 40.96 | 16.3 | | −22 | 17.76 | 2000 | 39.79 | 0.93 | Carnivora |
| *Pliohippus* sp. FMNH P 15870 | 291 | 21 | 270 | 135 | | −5 | 289 | 169,700 | 50 | 0.79 | Artiodactyla |
| *Plionictis* AMNH FM 25314 (= FMNH PM 58945) | 32.97 | 2.94 | 30.03 | 12.63 | 5.1 | −15 | 10.99 | 640 | 42.07 | 1.23 | Carnivora |
| *Poebrotherium* AMNH F:AM 31700 (= FMNH PM 59167) | 81.59 | 0 | 81.59 | 33.79 | 7.5 | −33.7 | 47.82 | 29,800 | 41.41 | 0.41 | Artiodactyla |
| *Potamotherium valentoni* NHMUK PV M 29357 (= FMNH PM 58906) | 64.4 | 4.02 | 60.37 | 35.69 | 5.7 | −22 | 37.3 | 10,000 | 59.11 | 0.67 | Carnivora |
| *Procamelus grandis* AMNH FM 40425 (= FMNH PM 59160) | 365.7 | 16 | 350 | 131.07 | 14.3 | −11 | 374.21 | 200,000 | 37.45 | 0.91 | Artiodactyla |

**Table 1 (*continued*)**

| Species/Genera: | S cm² | OB cm² | S-OB cm² | NC cm² | L cm | MYA | E ml or g | P g | %NC | EQ | Taxon |
|---|---|---|---|---|---|---|---|---|---|---|---|
| *Procynodictis angustidens* AMNH FM 95590 (= FMNH PM 57168) | 53.09 | 1.32 | 51.77 | 21.56 | 6.3 | −40 | 23.33 | 6,571 | 41.64 | 0.55 | Carnivora |
| *Promartes olcotti* FMNH P 25233 | 49.16 | 3.83 | 45.33 | 16.96 | 5.2 | −28 | 24.12 | 3,000 | 37.42 | 0.97 | Carnivora |
| *Promerycochoerus superbus* YPM VP 11002 (= FMNH PM 59072) | 174.4 | 0 | 174.4 | 68.59 | 10.8 | −32 | 147.12 | 178,000 | 39.33 | 0.39 | Artiodactyla |
| *Proterotherium cavum* AMNH FM 9245 (= FMNH PM 59742) | 106.09 | 4.54 | 101.55 | 29.23 | 9.2 | −17 | 57.35 | | 28.79 | | Notoungulata |
| *Protypotherium* FMNH P 13046 | 43.62 | 2.61 | 41.01 | 13.53 | 5.7 | −16.5 | 16.69 | 9,683 | 32.98 | 0.31 | Notoungulata |
| *Pseudaelurus validus* AMNH FM 61835 (= FMNH PM 58867) | 114.43 | 5.49 | 108.94 | 50.38 | 9.7 | −15 | 71.72 | 30,000 | 46.24 | 0.62 | Carnivora |
| AMNH FM 70025 (= FMNH PM 59211) | 207.87 | 13.39 | 194.47 | 93.65 | 10.6 | −11 | 168.43 | 50,000 | 48.15 | 1.03 | Perissodactyla |
| *Pseudotypotherium pseudopachygnathum* AMNH FM 14509 (= FMNH PM 59292) | 104.6 | 5.97 | 98.64 | 48 | 8.6 | −6 | 63.71 | 80,000 | 48.66 | 0.29 | Notoungulata |
| *Pterodon dasyuroides* NHMUK PV M 25985 b | 105.05 | 0 | 105.05 | 37.19 | 9.6 | −36 | 58.51 | 37,000 | 35.40 | 0.44 | Creodonta |
| *Rhynchippus equinus* FMNH P 13410 | 158.13 | 15.85 | 142.28 | 43.14 | 8.9 | −25 | 103.56 | 32,000 | 30.32 | 0.86 | Notoungulata |
| *Smilodectes gracilis* YPM VP 12152 (= FMNH PM 56263) | 25.93 | 0 | 25.93 | 9.13 | 3.4 | −48 | 9 | 1,600 | 35.23 | 0.55 | Primates |
| *Smilodon fatalis* LACMHC 2001-199 | 256.51 | 16.45 | 240.06 | 120.3 | 11.7 | −0.03 | 216 | 250,000 | 50.11 | 0.45 | Carnivora |
| *Sthenurus cf. orientalis* FMNH PM 59245 | 141.46 | 11.48 | 129.98 | 67.35 | 8.4 | −0.5 | 107.05 | 200,000 | 51.82 | 0.26 | Marsupialia |
| *Thylacoleo carniflex* SAMA P18681 (= FMNH PM 59244) | 170.89 | 16.9 | 153.99 | 63.7 | 9.6 | −2 | 120.01 | 130,000 | 41.37 | 0.39 | Marsupialia |
| Tillyhorse YPM VP 11694 | 49.52 | 4.83 | 44.69 | 5.84 | 6.4 | −52.8 | 15 | | 13.07 | | Condylarthra |
| *Titanoides primaevus* FM NH PM 8655 | 152.94 | 17.87 | 135.06 | 18.98 | 11.7 | −59.2 | 88.35 | 172,032 | 14.05 | 0.24 | Cimolesta |
| *Typotheriopsis internum* FMNH P 14420 | 112.8 | 4.4 | 108.4 | 52.69 | 9.6 | −8 | 75.1 | 6,846 | 48.60 | 1.74 | Notoungulata |
| *Uintatherium anceps* YPM VP 11036 | 391.2 | 76 | 343.2 | 64.37 | 17.2 | −49 | 386 | 1250,000 | 18.75 | 0.28 | Dinocerata |
| *Urocyon cinereoargenteus* UCMP V 12263 | 68.05 | 4.38 | 63.67 | 36.94 | 6.5 | −0.03 | 38.95 | 5,000 | 58.02 | 1.11 | Carnivora |
| *Ustatochoerus profectus* AMNH FM 33617 (= FMNH PM 59071) | 209.04 | 11.81 | 197.23 | 64.2 | 10.6 | −12.5 | 162.67 | 24,000 | 32.55 | 1.63 | Artiodactyla |
| *Zodiolestes daimonelixensis* FMNH P 12032 | 61.59 | 3.32 | 58.26 | 29.03 | 6.1 | −21 | 31.2 | 5,000 | 49.83 | 0.89 | Carnivora |
| **22 LIVING NON-PRIMATES** | | | | | | | | | | | |
| *Aonyx cinerea (Amblyonyx)* Rad 358 | 69.95 | 2.38 | 67.57 | 44.62 | 6 | 0 | 40.59 | 3,000 | 66.04 | 1.63 | Carnivora |
| *Canis latrans* brain NMHM Vertebrates WISC 62-301 | 125.27 | 8.22 | 117.05 | 82.44 | | 0 | 72.67 | 15,000 | 70.43 | 1 | Carnivora |
| *Cerdocyon thous* AMNH Mammals 36501 (= FMNH Mammals 146294 = Rad 294) | 78.5 | 6.93 | 71.57 | 43.89 | 7.1 | 0 | 45.67 | 6,000 | 61.32 | 1.15 | Carnivora |
| *Equus caballus* (Arabian) | 487.14 | 42.71 | 444.43 | 232.74 | | 0 | 669 | 400,000 | 52.37 | 1.03 | Perissodactyla |
| *Equus caballus* (draft horse) | 595.08 | 54.18 | 540.9 | 273.23 | | 0 | 881 | 800,000 | 50.33 | 0.85 | Perissodactyla |
| *Equus sp.* (zebra) LACM Mammals 342 | 473.08 | 41.55 | 431.54 | 249.9 | | 0 | 625 | 300,000 | 57.91 | 1.16 | Perissodactyla |

**Table 1** (*continued*)

| Species/Genera: | S cm² | OB cm² | S-OB cm² | NC cm² | L cm | MYA | E ml or g | P g | %NC | EQ | Taxon |
|---|---|---|---|---|---|---|---|---|---|---|---|
| *Felis catus* FMNH Mammals 146456 (= Rad 101) | 51.92 | 2.6 | 49.31 | 28.87 | 5.5 | 0 | 25.41 | 3,000 | 58.54 | 1.02 | Carnivora |
| *Lama glama* NMHM Vertebrates WISC 65-139 | 232.01 | 5.28 | 226.74 | 144.2 | 11 | 0 | 172.22 | 150,000 | 63.60 | 0.51 | Artiodactyla |
| *Lon tra canadensis* FMNH Mammals 146394 (= Rad 129) | 91.94 | 2.31 | 89.63 | 54.43 | | 0 | 59.87 | 10,000 | 60.73 | 1.07 | Carnivora |
| *Lutra lutra* Rad 366 | 70.54 | 1.96 | 68.58 | 40.6 | 6.6 | 0 | 39.22 | 10,000 | 59.20 | 0.7 | Carnivora |
| *Macropus fuliginosus* MSU 64023 braincast | 89.85 | 3.8 | 86.05 | 38.38 | 7 | 0 | 33.83 | 23,600 | 44.60 | 0.34 | Marsupialia |
| *Nasua n arica* WISC 62-404 braincast | | | 62.74 | | | | 28 | | | | Carnivora |
| *Odocoileus virgianus* NMHM Vertebrates WISC 67-81 2 braincast | 181 | 6.6 | 174.4 | 102.24 | | 0 | 124.6 | 75,000 | 58.62 | 0.58 | Artiodactyla |
| *Odocoileus virgianus* NMHM Vertebrates WISC 67-81 1 braincast | 206.58 | 6.6 | 199.98 | 102.24 | 11.2 | 0 | 124.6 | 75,000 | 51.13 | 0.58 | Artiodactyla |
| *Phascolarctos cinereus* Maciej Henneberg Lab braincast | 71.57 | 2.46 | 69.11 | 19.57 | | 0 | 15.93 | 10,000 | 28.32 | 0.29 | Marsupialia |
| *Phascolarctos cinereus* Maciej Henneberg Lab endocast | 78.01 | 6.52 | 71.49 | 21.32 | 7.2 | 0 | 36.5 | 10,000 | 29.83 | 0.66 | Marsupialia |
| *Procyon lotor* brain WISC 61-824 | | | 60.99 | 34.37 | | 0 | 25.79 | 7,000 | 56.35 | 0.59 | Carnivora |
| *Procyon lotor* FMNH Mammals 146352 (= Rad 154) | 84.42 | 5.18 | 79.23 | 47.03 | 7 | 0 | 54.18 | 7,000 | 59.36 | 1.23 | Carnivora |
| *Taxidea taxus* Rad 360 | 87.11 | 6.83 | 80.28 | 48.3 | 7 | 0 | 60 | 10,000 | 60.16 | 1.08 | Carnivora |
| *Ursus americanus* LACM | 281.56 | 19.26 | 262.3 | 160 | 11.8 | 0 | 276.67 | 140,000 | 61.00 | 0.86 | Carnivora |
| *Ursus arctos* Kodiak Bear LACM | 479.97 | 38.33 | 441.64 | 224.44 | 18.5 | 0 | 488.55 | 700,000 | 50.82 | 0.52 | Carnivora |
| *Vombatus ursinus* NMV C7780 | | | 103.23 | 48.45 | 7 | 0 | 82.2 | 28,000 | 46.93 | 0.74 | Marsupialia |
| **19 LIVING PRIMATES** | | | | | | | | | | | |
| *Cercocebus albigena* female AMNH Mammals 52583 | | | 107.84 | 86.65 | 6.7 | 0 | 79.64 | 7,900 | 80.35 | 1.67 | Primates |
| *Cercopithecus pygenthus* male AMNH Mammals 52468 | | | 101.93 | 79.91 | 6.8 | 0 | 71.86 | 4,200 | 78.40 | 2.3 | Primates |
| *Chiropotes albinasa* female FMNH Mammals 94927 | | | 82.57 | 65.26 | 5.9 | 0 | 53 | 3,000 | 79.03 | 2.12 | Primates |
| *Colobus guereza* AMNH Mammals 52217 | | | 112.01 | 76.33 | 7 | 0 | 85.27 | 10,500 | 68.15 | 1.48 | Primates |
| *Erythrocebus patas* female infant AMNH Mammals 52574 | | | 116.93 | 91.23 | 7.1 | 0 | 90.06 | 17,000 | 78.02 | 1.14 | Primates |
| *Homo sapiens* Falk A | | | 540.59 | 432.55 | 14.3 | 0 | 945.7 | 50,000 | 80.01 | 5.81 | Primates |
| *Homo sapiens* Falk B | | | 682.38 | 530.32 | 16.3 | 0 | 1,369.71 | 70,000 | 77.72 | 6.72 | Primates |
| *Hylobates lar* Falk 386 | | | 123.4 | 78.11 | 7.1 | 0 | 99.32 | 8,000 | 63.30 | 2.07 | Primates |
| *Macacca mulatta* brain WISC 69-307 | | | 114.72 | 79.45 | 6.2 | 0 | 71.61 | 6,000 | 69.26 | 1.81 | Primates |
| *Mandrillus sphinx* AMNH Mammals 274 | | | 154.88 | 119.09 | 8.1 | 0 | 131.85 | 18,000 | 76.89 | 1.6 | Primates |
| *Nasalis larvatus* male MCZ Mammals 37328 | | | 121.94 | 89.15 | 5.9 | 0 | 97 | 14,000 | 73.11 | 1.39 | Primates |
| *Pan troglodytes* NMHM Vertebrates WISC 63-307 braincast | | | 331.52 | 267.73 | 10.1 | 0 | 307.39 | 40,000 | 80.76 | 2.19 | Primates |

Jerison et al. (2025), *PeerJ*, DOI 10.7717/peerj.19826

Peer*J*

**Table 1** (*continued*)

| Species/Genera: | S cm² | OB cm² | S-OB cm² | NC cm² | L cm | MYA | E ml or g | P g | %NC | EQ | Taxon |
|---|---|---|---|---|---|---|---|---|---|---|---|
| *Pan troglodytes* MCZ endocast | 278.46 | 1.8 | 276.66 | 196.58 | | 0 | 371.18 | 50,000 | 71.05 | 2.28 | Primates |
| *Pithecia monachus* female AMNH Mammals 75981 | | | 68.05 | 53.09 | 5.9 | 0 | 39.73 | 1,500 | 78.02 | 2.52 | Primates |
| *Presbytis johnii* female AMNH Mammals 54644 | | | 114.26 | 82.62 | 7.2 | 0 | 85.85 | 13,400 | 72.31 | 1.27 | Primates |
| *Pygathrix nigripes* male AMNH Mammals 69555 | | | 106.08 | 80.39 | 6.5 | 0 | 77.71 | 7,500 | 75.78 | 1.69 | Primates |
| *Rhinopithecus avunculas* male MCZ Mammals 13681 | | | 136.64 | 99.75 | 7.4 | 0 | 114.21 | 8,000 | 73.00 | 2.38 | Primates |
| *Simias concolor* male AMNH Mammals 103359 | | | 82.54 | 61.19 | 5.8 | 0 | 54 | 7,000 | 74.13 | 1.23 | Primates |
| *Theropithecus gelada* male FMNH Mammals 8174 | | | 146.82 | 108.74 | 8.1 | 0 | 131.08 | 17,000 | 74.06 | 1.65 | Primates |

**Notes.**

S, surface area; OB, olfactory bulb area; S-OB, surface area excluding olfactory bulbs; NC, neocortex; L, length of specimen image; MYA, millions of years ago; E, volume of specimen; P, body size; %NC, neocorticalization; EQ, encephalization quotient, re 2/3.

### Assessing neocorticalization and encephalization over 60 million years using endocasts

Table 1 provides complete summary data on the digitized scans of mammal endocasts and brains. A complete description of endocast provenance, endocast features, and body size determinations are presented in the Supplementary Results. Below, we present the digitized endocast images. Geological Time Scale (v. 6.0), published by the Geological Society of America (*Walker & Geissman, 2022*), provided dates and correlated time periods listed below.

### Edinger's early horses

To illustrate the use of endocasts in modeling neocorticalization and encephalization, the endocasts of Edinger's horses (*Edinger, 1948*), photographs of which have frequently been used to illustrate progressive brain evolution (*e.g.*, *MacFadden, 1994*; *Simpson, 1951*; subsequent sources that cite them), are first presented. Figure 2 shows scans of endocasts of five of Edinger's species and adds *Hyracotherium* (FMNH PM 59207=AMNH 55268). Edinger's "*Eohippus*" (YPM 11694, listed as "Tillyhorse" in Table 1) is from *Radinsky (1976)*. The digitized models of the bodies presented in Figs. 3 and 4 are scans of careful sculptures by *Gidley (1927)*, from which the length, surface area, and volume were determined. The endocast of *Mesohippus* (Fig. 4A) was larger, more encephalized, and much more convoluted than that of *Hyracotherium* (Fig. 3A). Figure 5 shows the remainder of the endocasts of Edinger's equoid genera: three fossil genera and three recent genera, including a zebra and two domesticated horses (a pony and a draft horse, reflecting body size variations within the domesticated species).

### Paleocene fossils

The earliest digitized mammalian endocasts presented here are of the very large and heavy late Paleocene *Titanoides* and *Barylambda,* and the smaller *Arctocyon,* with the Paleocene sampled, here defined as about 66 Ma to 56 Ma. The Paleocene-Eocene boundary in our sample is somewhat artificial; the *Phenacodus* specimen is an individual that had survived into the early Eocene, but it should be representative of Paleocene members of the genus. These species are shown in Fig. 6.

### Early eocene fossils

The Early Eocene dates used here are from 56 Ma to 42 Ma, and the endocasts of *Coryphodon*, *Palaeosyops*, *Heptodon*, and *Isectolophus* are shown in Fig. 7; Edinger's Eocene "*Eohippus*" and *Hyracotherium* are shown in Figs. 2 and 3; *Hyrachyus*, *Orthocynodon* ("*Amynodon*"), *Amynodon*, and *Eomoropus* are shown in Fig. 8; *Mesatirhinus junius*, *Mesatirhinus petersoni*, *Pachyaena*, and *Mesonyx* are shown in Fig. 9; and *Smilodectes* and *Notharctus* are shown in Figs. 10A and 10B.

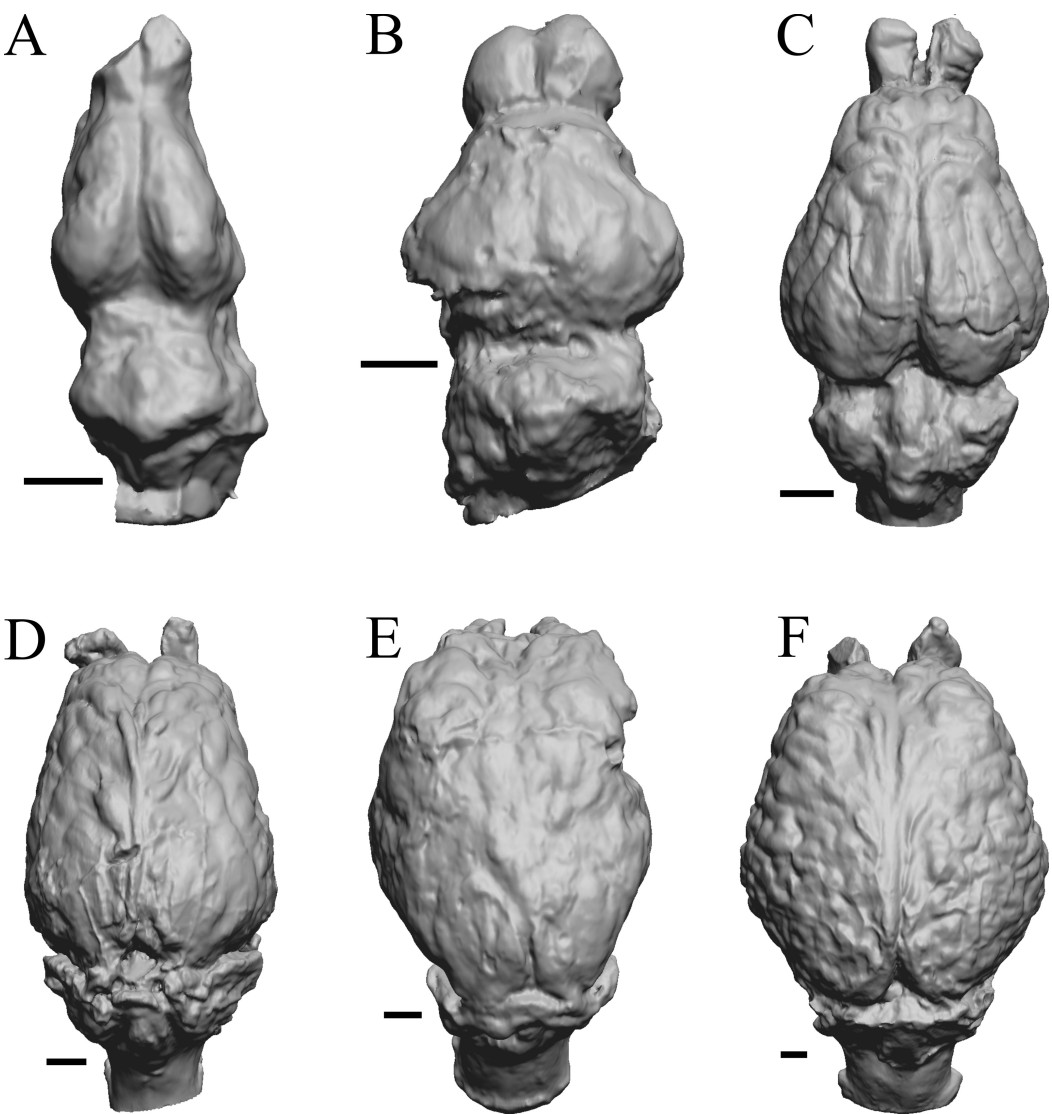

**Figure 2** **Five Edinger equoids plus one.** Digitized images of six fossil endocasts sketched by *Edinger (1948)* and *Radinsky (1976)*. Endocasts are in dorsal view with the rostral pole pointed at the top of the figure. (A) Edinger's Eocene "*Eohippus*" ("Tillyhorse") (YPM VP 11694). (B) Eocene "*Hyracotherium*" *tapirinum* (AMNH FM 55268 = FMNH PM 59207). (C) Oligocene *Mesohippus bairdi* (AMNH FM 9814 = FMNH PM 59221). (D) Miocene *Merychippus isonesus* (AMNH FM 71150 = FMNH PM 59208). (E) Mio-Pliocene *Pliohippus* (FMNH P 15870). (F) Pleistocene La Brea Horse *Equus occidentalis* (LACMHC 3500-17). Scale bars = one cm.

## Later Eocene fossils

Here, "Later Eocene" is 42 Ma to 34 Ma. The endocast images of *Necrolemur* and *Adapis* are shown in Fig. 10; *Uintatherium*, *Menodus* (*Titanotherium*), *Moeritherium*, and *Arsinotherium* in Fig. 11; *Pterodon*, *Cynodictis*, *Cynohyaenodon*, and *Procynodictis* in Fig. 12;

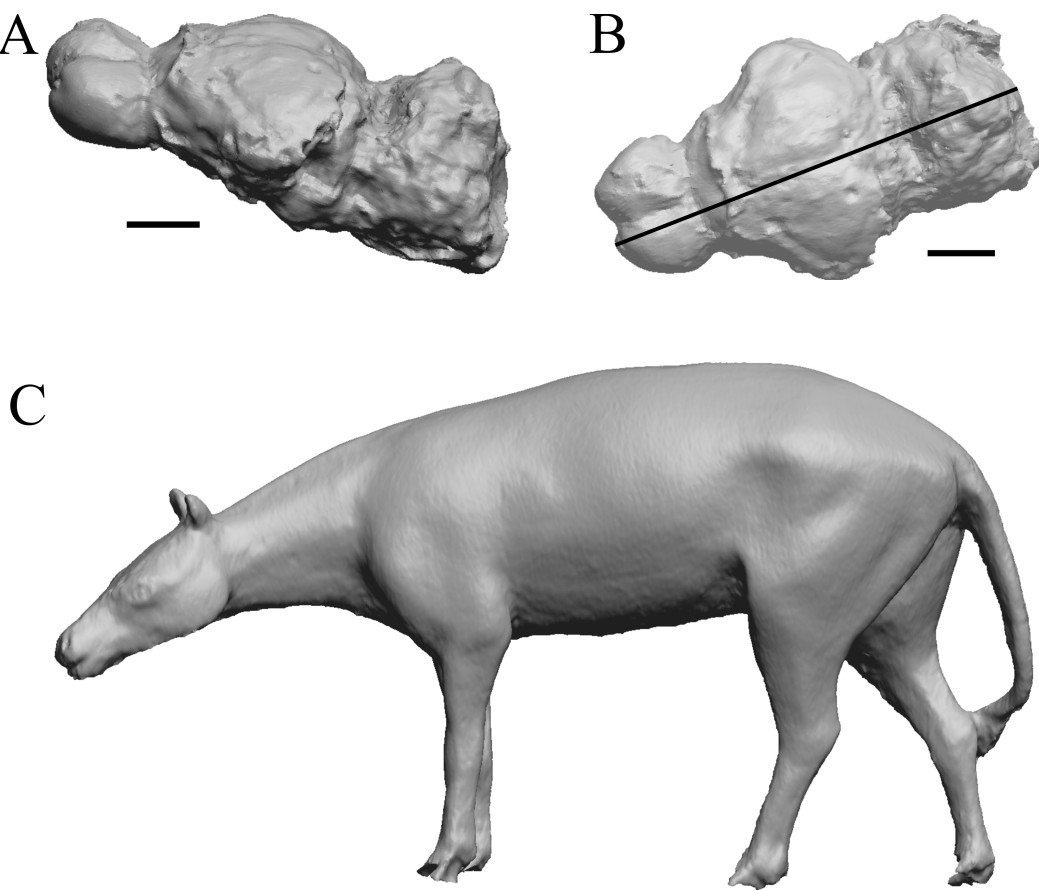

**Figure 3** *Hyracotherium* **endocast and body.** (A) 3D image of the *Hyracotherium* endocast (AMNH FM 55268 = FMNH PM 59207, *Radinsky, 1976*) in left dorsolateral view. (B) Image of endocast in dorsal view prior to rendering with rostral pointing left; measured length (black line) marked by software. (C) Scan of model sculpted by *Gidley (1927)*. Scale bars = one cm. Figure 4. *Mesohippus* endocast and body. (A) *Mesohippus bairdi* endocast in left dorsolateral view (AMNH FM 9814 = FMNH PM 59221). (B) Tessellated image of the endocast in left dorsolateral view, showing length line. (C) Digital image of the Gidley model, with length markings and author's notes. Scale bars = one cm.

*Cebochoerus, Hylomeryx, Mixtotherium,* and *Chadronia* in Fig. 13; and *Anoplotherium, Patriomanis, Poebrotherium,* and *Bathygenys* in Fig. 14.

## Oligocene fossils

The "Oligocene" samples date from 34 Ma to 23 Ma. These dates anchor the analysis of neocorticalization as changes with the passage of time. *Daphoenus, Dinictis, Eusmilus,* and *Hoplophoneus* are shown in Fig. 15; *Merycoidodon, Mesohippus, Promerycochoerus,* and *Hesperocyon* in Fig. 16; *Leptictis (Ictops), Leptauchenia, Halitherium,* and *Hapalops* in Fig. 17; *Leontinia, Rhynchippus, Archaeotherium,* and *Promartes* in Fig. 18; and *Mesocyon* in Fig. 19A.

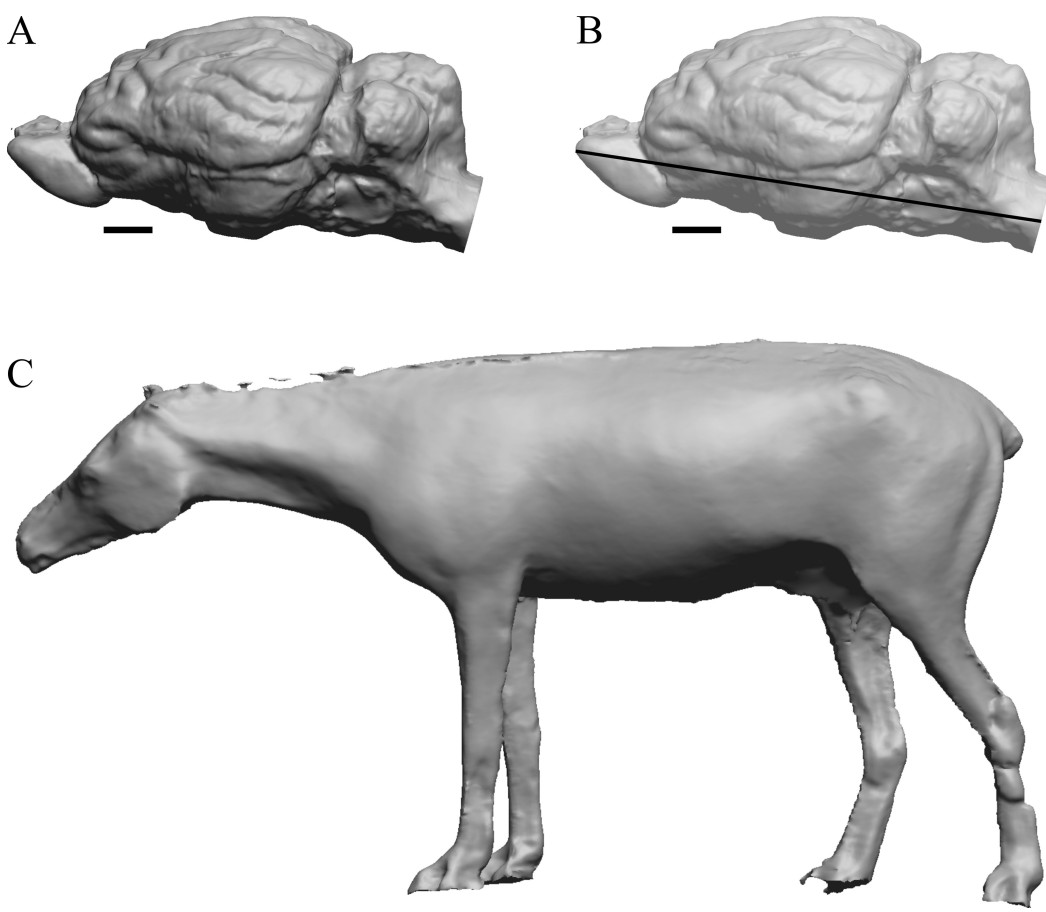

**Figure 4 Mesohippus. endocast and body.** (A) *Mesohippus bairdi* endocast in left dorsolateral view (AMNH FM 9814 = FMNH PM 59221). (B) Tessellated image of the endocast in left dorsolateral view, showing length line. (C) Digital image of the Gidley model, with length markings and author's notes. Scale bars = one cm.

## Mio-Pliocene fossils

This group dates from 23 Ma to mid-Pliocene, about 3 Ma. *Mustelictis, Leptocyon,* and *Eporeodon* are shown in Fig. 19; *Enaliarctos, Potamotherium, Plesiogale,* and *Zodiolestes* in Fig. 20; *Desmathyus (Hesperhyus), Oxydactylus, Homalodotherium,* and *Borhyaena* in Fig. 21; *Protypotherium, Proterotherium, Nesodon,* and *Merycochoerus* in Fig. 22; *Adinotherium, Merychippus (Atavahippus), Plionictis,* and *Pseudaelurus* in Fig. 23; *Paracynarctus, Ustatochoerus* (note: endocast volume is large due to expanded cerebellum, but relative telencephalon size is comparable that of modern species), *Carpocyon ("Osteoborus"),* and *Pseudhipparion* in Fig. 24; *Paratomarctus, Hemicyon, Pseudotypotherium,* and *Tyopotheriopsis* in Fig. 25; and *Cormohipparion, Procamelus, Homotherium,* and *Mylodon* in Fig. 26.

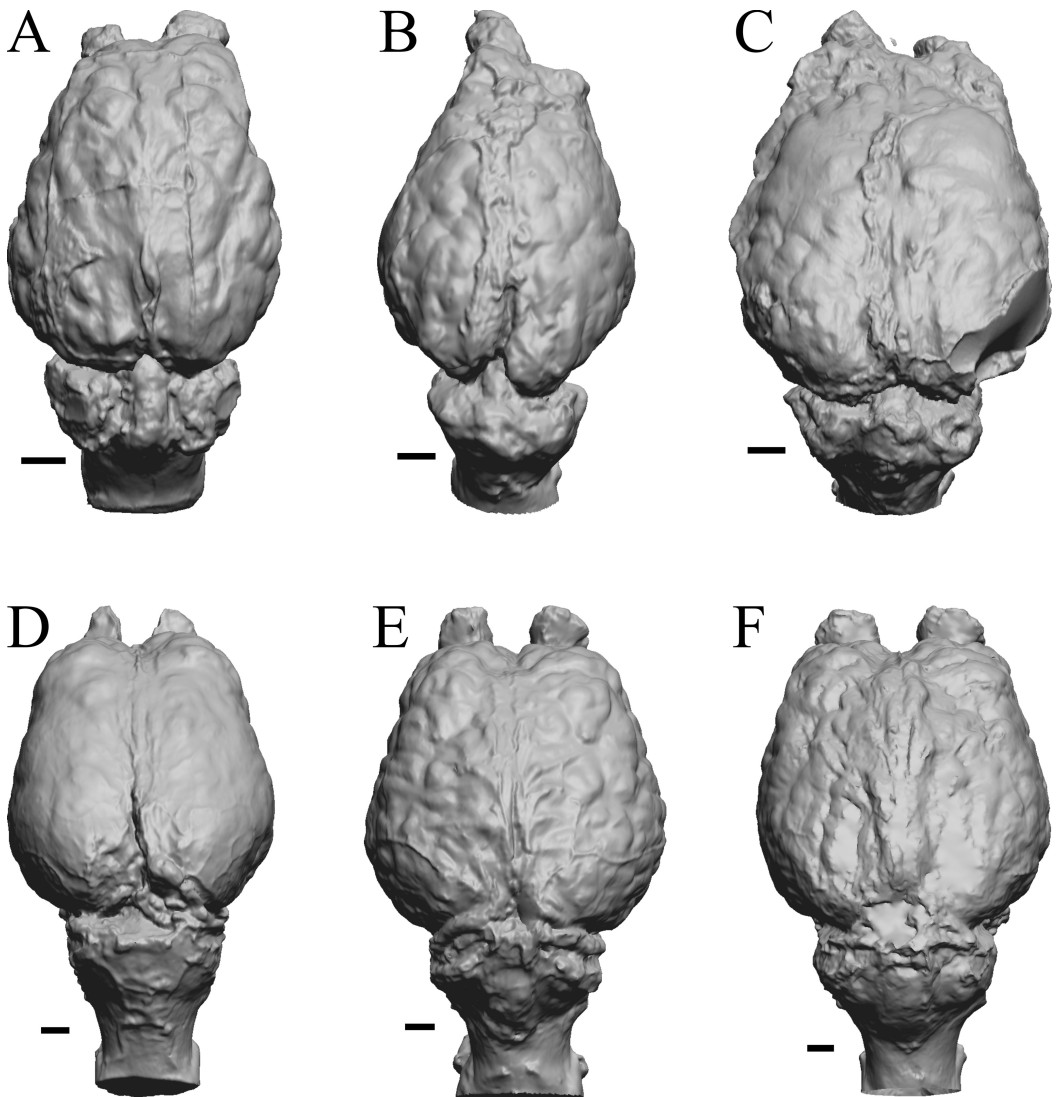

**Figure 5 Six of Edinger's later horses; additional digitized Edinger endocasts.** Endocasts are in dorsal view with the rostral pole pointed at the top of the figure. (A) *Pseudhipparion* (AMNH FM 70025 = FMNH PM 59211). (B) *Neohipparion* (FMNH P 15871). (C) *Cormohipparion* (AMNH FM 71886 = FMNH PM 59220). (D) *Equus* sp. (zebra, LACM Mammals 342). (E) *Equus caballus* (Arabian horse, LACM). (F) *Equus caballus* (draft horse, LACM). Scale bars = one cm.

## Plio-pleistocene and recent fossils

*Glossotherium, Arctodus, Aenocyon dirus,* and *Megalonyx* are shown in Fig. 27; *Nothrotheriops, Panthera, Smilodon,* and *Urocyon* are shown in Fig. 28; *Platygonus, Sthenurus, Thylacoleo,* and *Archaeolemur* are shown in Fig. 29; *Pachylemur (Lemur) insignis,* and *Palaeopropithecus, Australopithecus robustus,* and *Australopithecus africanus* in Fig. 30.

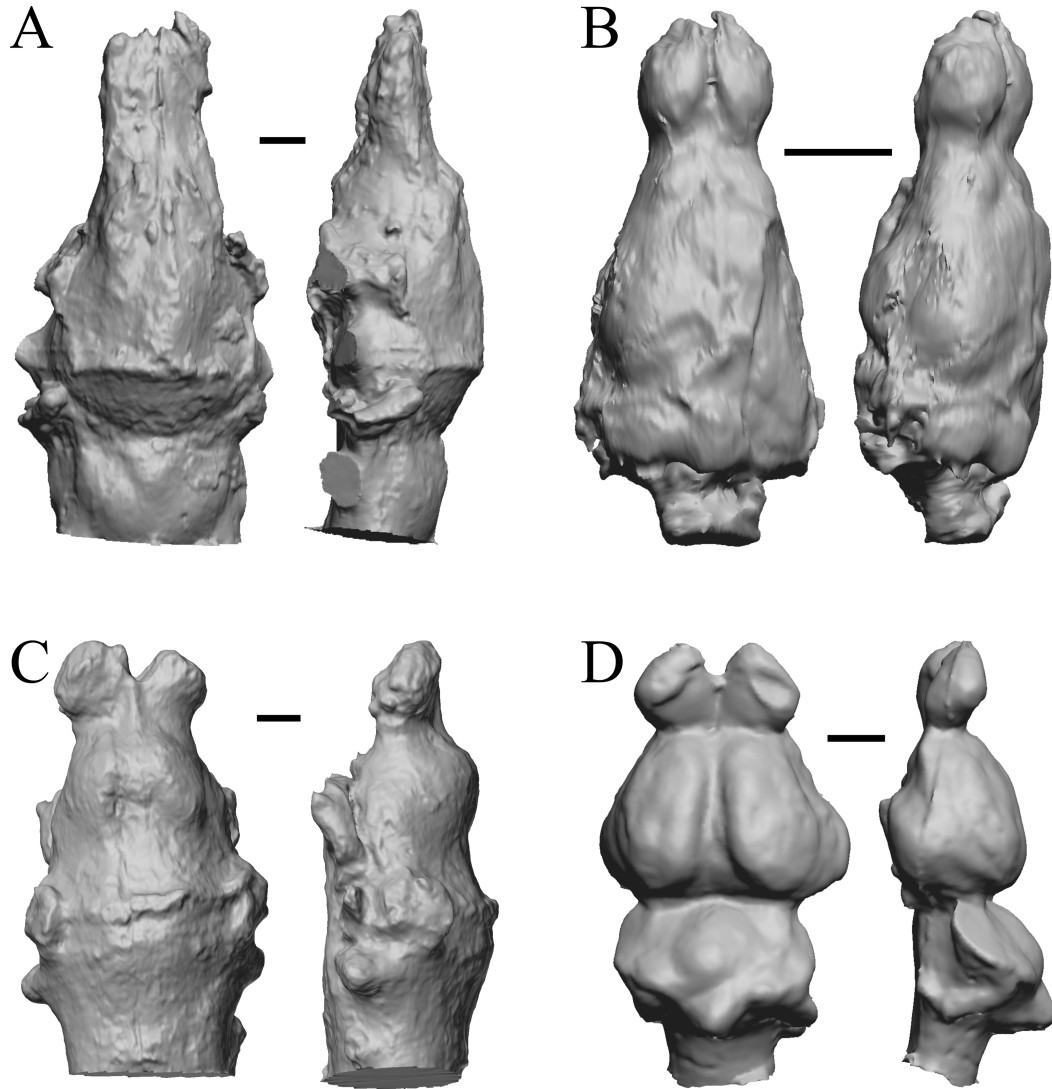

**Figure 6** *Titanoides, Arctocyon, Barylambda, Phenacodus.* All endocasts in dorsal (left) and left lateral (right) views with rostral pole pointed at the top of the figure. (A) *Titanoides primaevus* (FMNH PM 8655). (B) *Arctocyon primaevus* (MNHN F CR700). (C) *Barylambda schmidti* (FMNH P 26075 and FMNH P 15573). (D) *Phenacodus primaevus* (AMNH FM 4369 = FMNH PM 59042). Scale bars = one cm. Further details may be found in Additional Information.

## Cetacean fossils

As in the living cetacean brain, no rhinal fissure exists in these fossils and thus no indication of an olfactory bulb or tract; therefore, we could not assess neocorticalization. See scans of the three fossil whales in the left panel of Fig. 31.

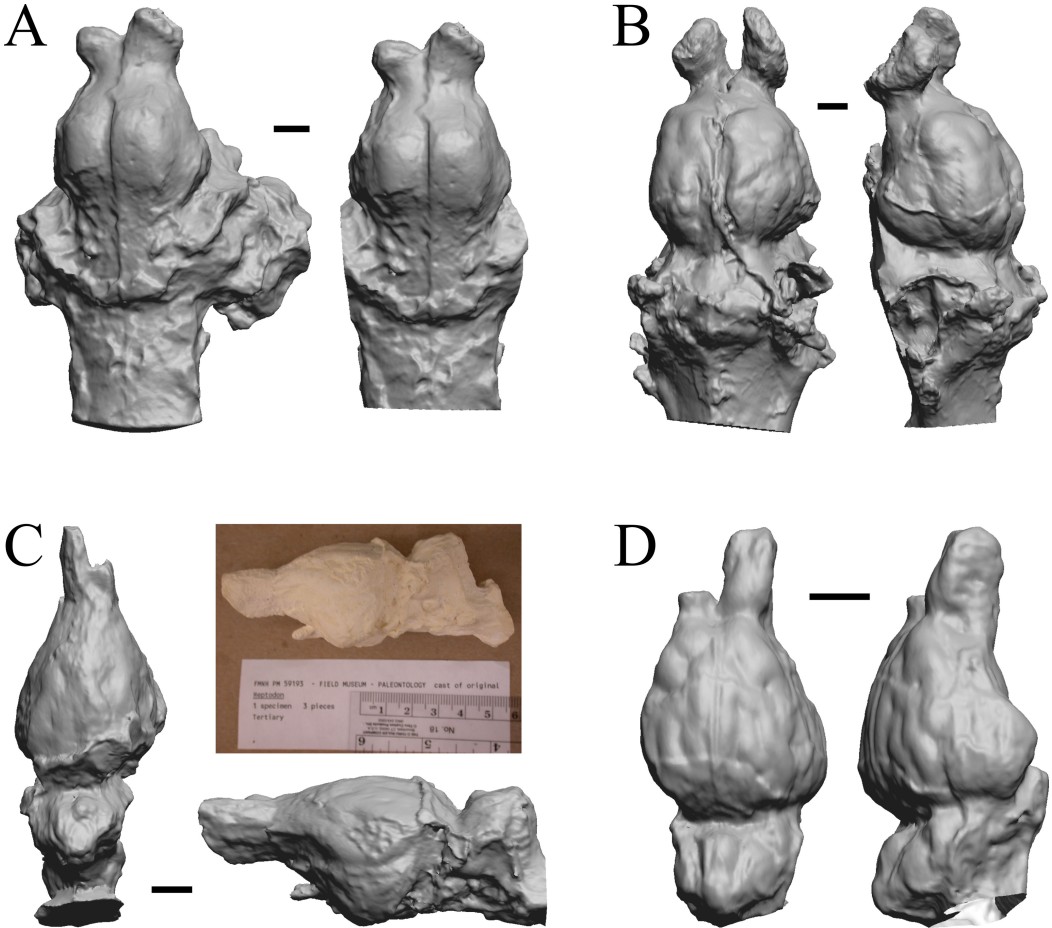

**Figure 7** *Coryphodon, Palaeosyops, Heptodon, Isectolophus* **endocasts.** (A) *Coryphodon hamatus* (YPM VP 11331 = FMNH PM 59241) in left dorsolateral (left) and dorsal (right) views with rostral pole pointed at the top of the figure. (B) *Palaeosyops leidyi* (FMNH PM 59198) in dorsal (left) and left lateral (right) views with rostral pole pointed at the top of the figure. (C) Three views of *Heptodon* sp. (FMNH PM 59193) Vertically oriented endocast in dorsal view at left; photograph of specimen above center; horizontally oriented endocast in left lateral view below center. (D) Endocast of *Isectolophus latidens* (AMNH FM 12222 = FMNH PM 59179) in dorsal (left) and left lateral (right) views with rostral pole pointed at the top of the figure. Scale bars = one cm. Further details may be found in Additional Information.

## Extant non-primate mammals

*Aonyx, Ursus* (Black Bear), *Canis latrans,* and *Felis catus* are shown in Fig. 32; *Cerdocyon, Odocoileus, Ursus* (Kodiak), and *Lama* are shown in Fig. 33; *Lutra lutra, Lutra canadensis, Procyon* endocast and braincast, and *Nasua* are shown in Fig. 34; and *Phascolarctos, Macropus, Vombatus,* and *Taxidea* in Fig. 35.

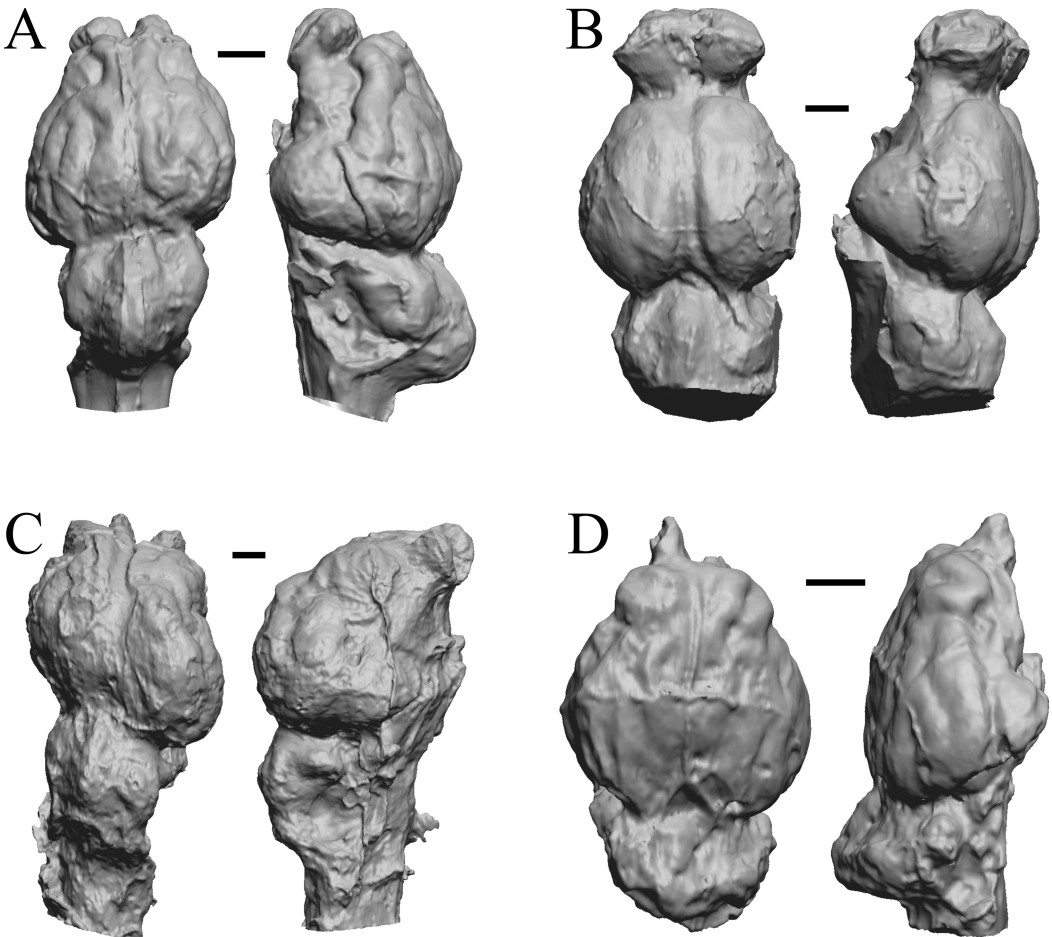

**Figure 8** *Hyrachyus, Orthocynodon* (**"*Amynodon*"**), *Amynodon, Eomoropus* **endocasts.** (A) *Hyrachyus modestus* (YPM VP 11082 = FMNH PM 59240) in dorsal (left) and left lateral (right) views with rostral pole pointed at the top of the figure. (B) *Orthocynodon* sp. (YPM VPPU 10145 = FMNH PM 59177) in dorsal (left) and left lateral (right) views with rostral pole pointed at the top of the figure. (C) *Amynodon advenus* (YPM VP 11453 = FMNH PM 59231) in dorsal (left) and right lateral (right) views with rostral pole pointed at the top of the figure. (D) *Eomoropus amaorum* (AMNH FM 5096 = FMNH PM 59182) in dorsal (left) and right lateral (right) views with rostral pole pointed at the top of the figure. Scale bars = one cm.

## Extant primates

*Chiropotes*, Mandrill, Homo-Falk A, and Homo-Falk B are shown in Fig. 36, and primate endocasts and braincasts (16 left hemisphere endocasts) are shown in Fig. 37.

## Brain size—uses and limitations

For closely-related extant mammal species, brain size, and especially relative brain size, has been put forth as a generally effective tool for estimating many traits such as information processing capacity (*i.e.,* "intelligence"; *e.g.*, *Roth & Dicke, 2005*) and sources therein, but also note the authors' emphasis on number of cortical neurons rather than brain size;

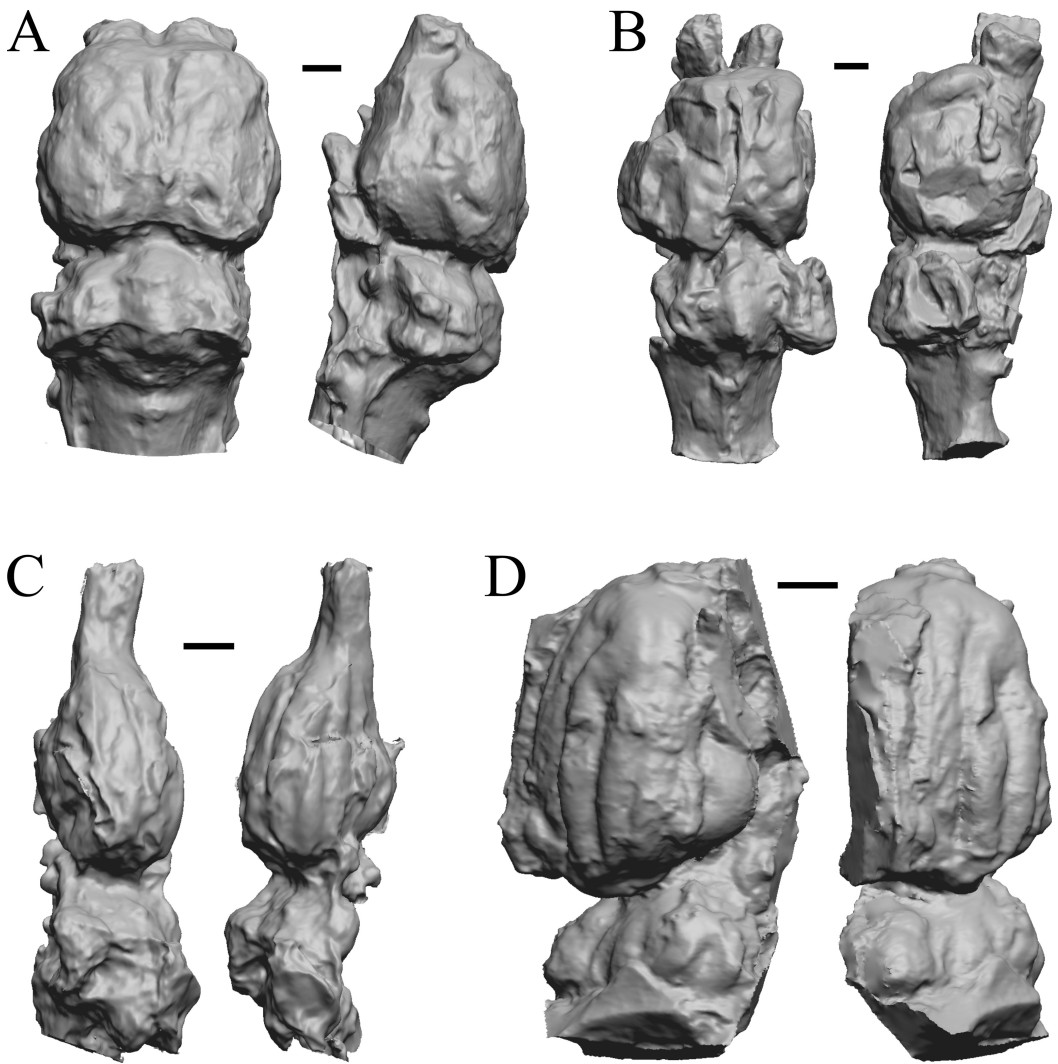

**Figure 9** *Mesatirhinus junius, Mesatirhinus petersoni, Pachyaena, Mesonyx* endocasts. (A) *Mesatirhinus junius* (YPM VPPU 10041 = FMNH PM 59197) in dorsal (left) and left lateral (right) views with rostral pole pointed at the top of the figure; the lateral view on the right shows the rhinal fissure. (B) *Mesatirhinus petersoni* (AMNH FM 1509 = FMNH PM 59196) in dorsal (left) and right lateral (right) views with rostral pole pointed at the top of the figure. (C) *Pachyaena ossifraga* (YPM VPPU 14708) in dorsal (left) and left lateral (right) views with rostral pole pointed at the top of the figure; the forebrain and hindbrain are linearly aligned, similar to many Eocene species. (D) *Mesonyx obtusidens* (YPM VP 13141 = FMNH PM 57139) in right lateral (left) and dorsal (right) views with rostral pole pointed at the top of the figure; about half of the brain and matrix was present, and excess matrix was removed when preparing the digital image. Scale bars = one cm.

(*Striedter & Northcutt, 2019*) and sources therein, (*Smaers et al., 2021*); but see *Van Schaik et al., 2021*), the size of certain gross brain regions (*e.g.*, *O'Keefe & Nadel, 1978*; *Stephan, Frahm & Baron, 1981*; *Stephan, Baron & Frahm, 1991*; *Jerison, 1991*; *Barton & Harvey, 2000*; *Baddeley, 2007*; *Brown et al., 2016*), brain and cortex surface areas (*Jerison, 1982*;

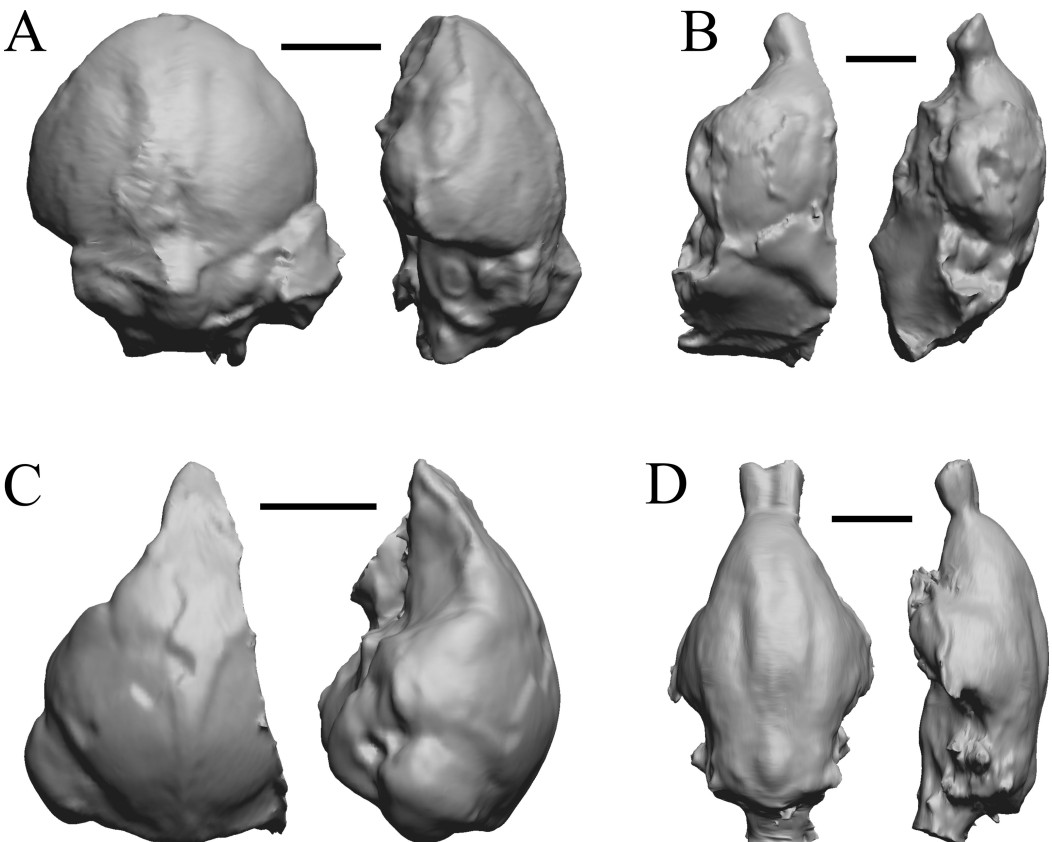

**Figure 10** *Smilodectes, Notharctus, Necrolemur, Adapis* **endocasts.** All endocasts in dorsal (left) and left lateral (right) views with rostral pole pointed at the top of the figure. (A) *Smilodectes gracilis* (YPM VP 12152 = FMNH PM 59263). (B) *Notharctus tenebrosus* (FMNH PM 59264); estimating neocorticalization was difficult because of the fragmented endocast, which was of approximately half the brain region and included posterior "brain" and matrix. (C) *Necrolemur antiquus* (YPM VP 18302 = FMNH PM 59261), mostly left hemisphere. (D) *Adapis parisiensis* (NHMUK M1340 = FMNH 59259). Scale bars = one cm.

*Ridgway & Brownson, 1984*; *Haug, 1987*; this study), and neuronal density (*Herculano-Houzel, 2010*; *Herculano-Houzel, 2017*; *Dicke & Roth, 2016*). Yet, lack of functional demand for, or gradual disuse of, brain regions can result in reorganization of the cortical projections with important changes in the details of the brain maps (*e.g.*, *Qi, Stepniewska & Kaas, 2000*). Importantly, changes in cortical organization are rarely if ever observable on endocasts and thus represent a limitation to how much can be extrapolated from comparative analyses of extant taxa. Indeed, uniquely specialized behavioral capacities and related brain structures+functions evolved, and it remains unclear if endocasts consistently and fully capture the frequency and/or nuances of these instances of reorganization through deep time. We note that some studies have been successful at showing evolutionary impact of functional demands on the evolution of brain size and shape (*e.g.*, *Bertrand et al., 2021*; *Bertrand et al., 2024a*; *Bertrand et al., 2024b*; *Schwartz et al., 2023*).

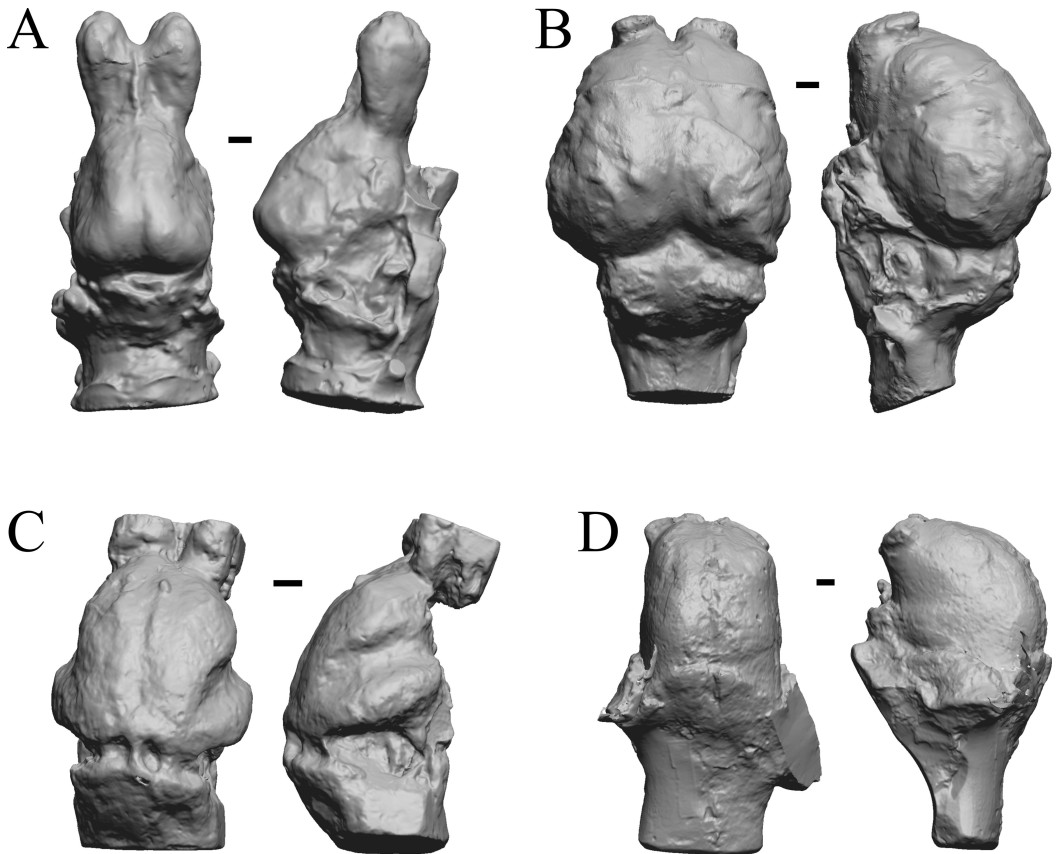

**Figure 11** **_Moeritherium, Arsinotherium_ endocasts.** (A) _Uintatherium anceps_ (YPM VP 11036) in dorsal (left) and right lateral (right) views with rostral pole pointed at the top of the figure. (B) _Menodus (Titanotherium) ingens_ (FMNH PM 59199) in dorsal (left) and left lateral (right) views with rostral pole pointed at the top of the figure. (C) _Moeritherium_ (NHMUK PV M 9176 b) in dorsal (left) and right lateral (right) views, with rostral pole pointed at the top of the figure. (D) _Arsinotherium zitelli_ (NHMUK PV M 8539) in dorsal (left) and left lateral (right) views with rostral pole pointed at the top of the figure; rhinal fissure is not visible. Scale bars = one cm. Further details may be found in Additional Information.

Yet, endocasts remain a rich source of information about brain size, shape, and composition in extinct taxa, especially if the uniformitarian hypothesis (_Simpson, 1970_) holds for the neurobiology and physiology of vertebrates. Further, especially in mammals and birds, fidelity between brain and endocast produces highly detailed endocasts (_e.g._, _Jerison, 1969_; _Jerison, 1973_; _Jerison, 1977_; _De Miguel & Henneberg, 1998_; _Iwaniuk & Nelson, 2002_; _Macrini et al., 2007_; _Watanabe et al., 2019_; _Early et al., 2020_), and these close brain:endocast relationships appears to extend far into the fossil record (_e.g._, _Rowe, Macrini & Luo, 2011_; _Balanoff, Smaers & Turner, 2016_ and sources therein). There are rare but specific cases where evidence from fossils sometimes contests the uniformitarian hypothesis. For example, the endocast volume of the Eocene _Hyracotherium_ in this study is 24 ml, and given endocast shape, we roughly estimate the approximate volume of its

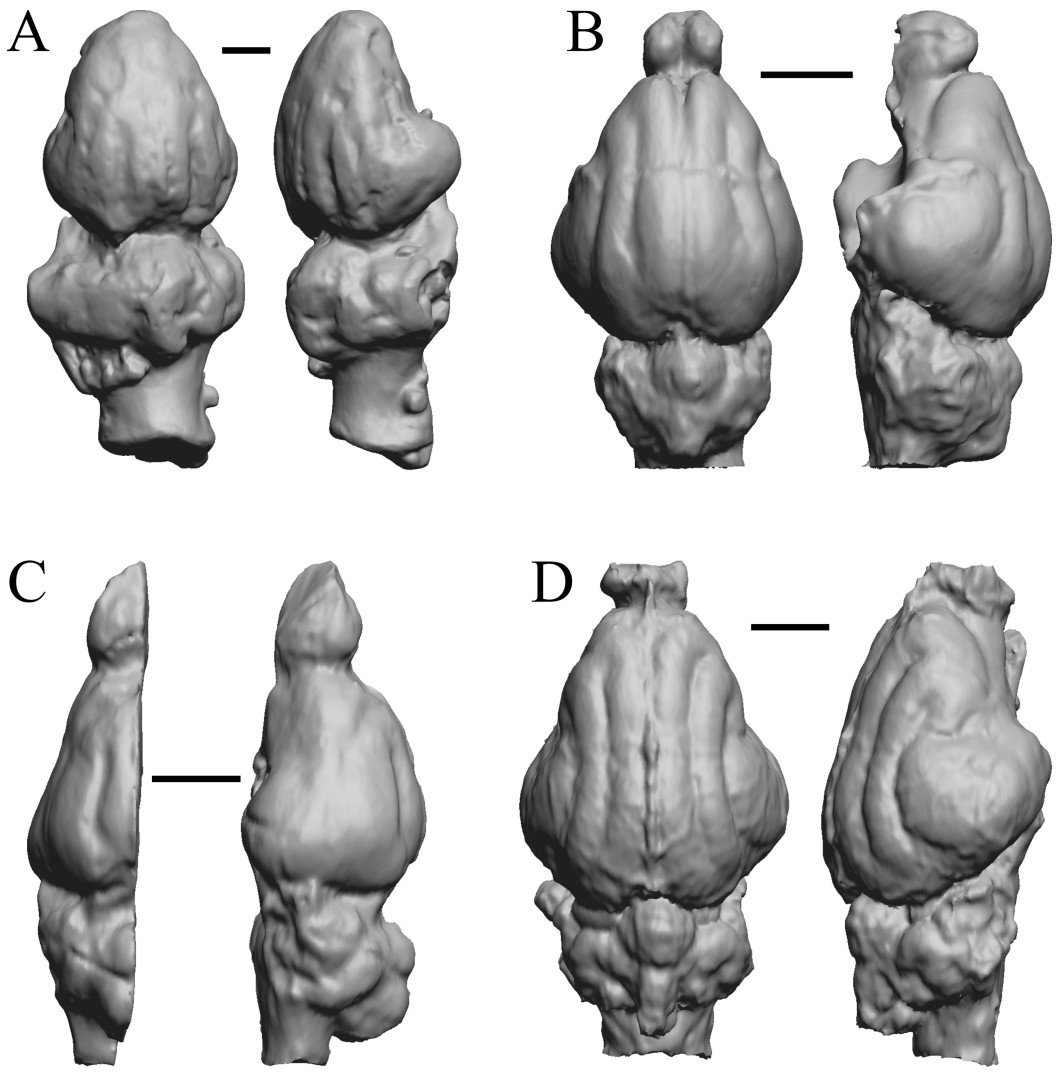

**Figure 12 *Pterodon, Cynodictis, Cynohyaenodon, Procynodictis* endocasts.** (A) *Pterodon dasyuroides* (NHMUK PV M 25985 b) in dorsal (left) and right lateral (right) views with rostral pole pointed at the top of the figure; olfactory bulbs not preserved. (B) *Cynodictis cayluxi* (FMNH PM 59013) in dorsal (left) and left lateral (right) views with rostral pole pointed at the top of the figure. (C) *Cynohyaenodon cayluxi* (FMNH PM 57153) in dorsal (left) and left lateral (right) views with rostral pole pointed at the top of the figure; (D) *Procynodictis angustidens*, (AMNH FM 95590 = FMNH PM 57168) in dorsal (left) and right lateral (right) views with rostral pole pointed at the top of the figure. Scale bars = one cm. Further details may be found in Additional Information.

cerebellum as at least five ml—about twice as large as expected given comparative data. Although these cases inject complexity in the interpretation of quantitative trends, there are nevertheless present interesting demonstrations of difference that have potential to shape our understanding of how, when, and, perhaps why, brain regions evolve.

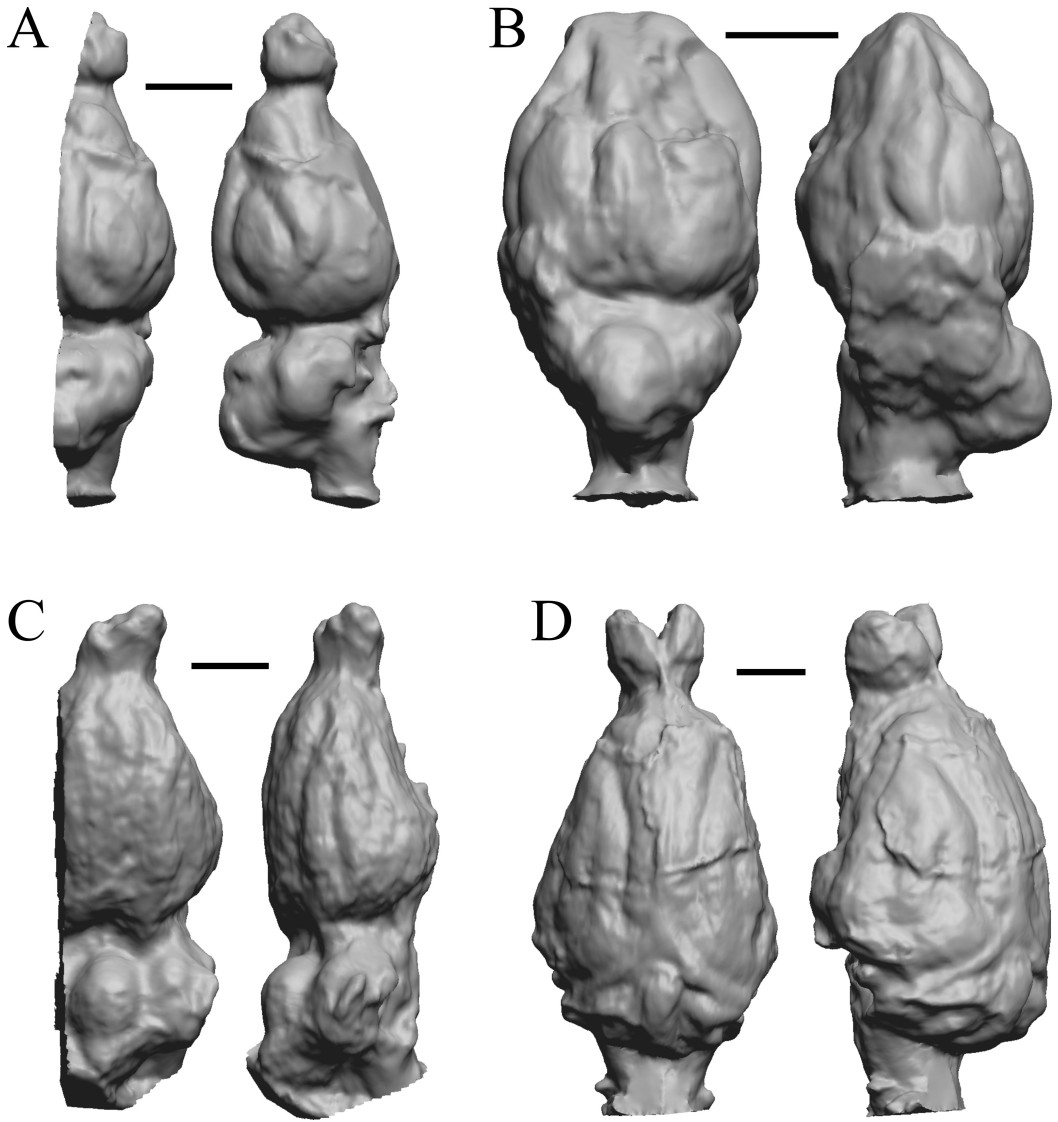

**Figure 13** *Cebochoerus, Hylomeryx, Mixtotherium, Chadronia* **endocasts.** (A) *Cebochoerus lacustris* (FMNH PM 59051) in dorsal (left) and right lateral (right) views with rostral pole pointed at the top of the figure; only a partial endocast was available. (B) *Hylomeryx (Sphenomeryx) quadricuspis* (CM VP 2915 = FMNH PM 59055) in dorsal (left) and left lateral (right) views with rostral pole pointed at the top of the figure; olfactory bulbs missing. (C) *Mixtotherium cuspidatum* (FMNH PM 59052) in dorsal (left) and right lateral (right) views with rostral pole pointed at the top of the figure. (D) *Chadronia margaretae* (AMNH FM 109412 = FMNH PM 57129) in dorsal (left) and left lateral (right) views with rostral pole pointed at the top of the figure. Scale bars = one cm. Further details may be found in Additional Information.

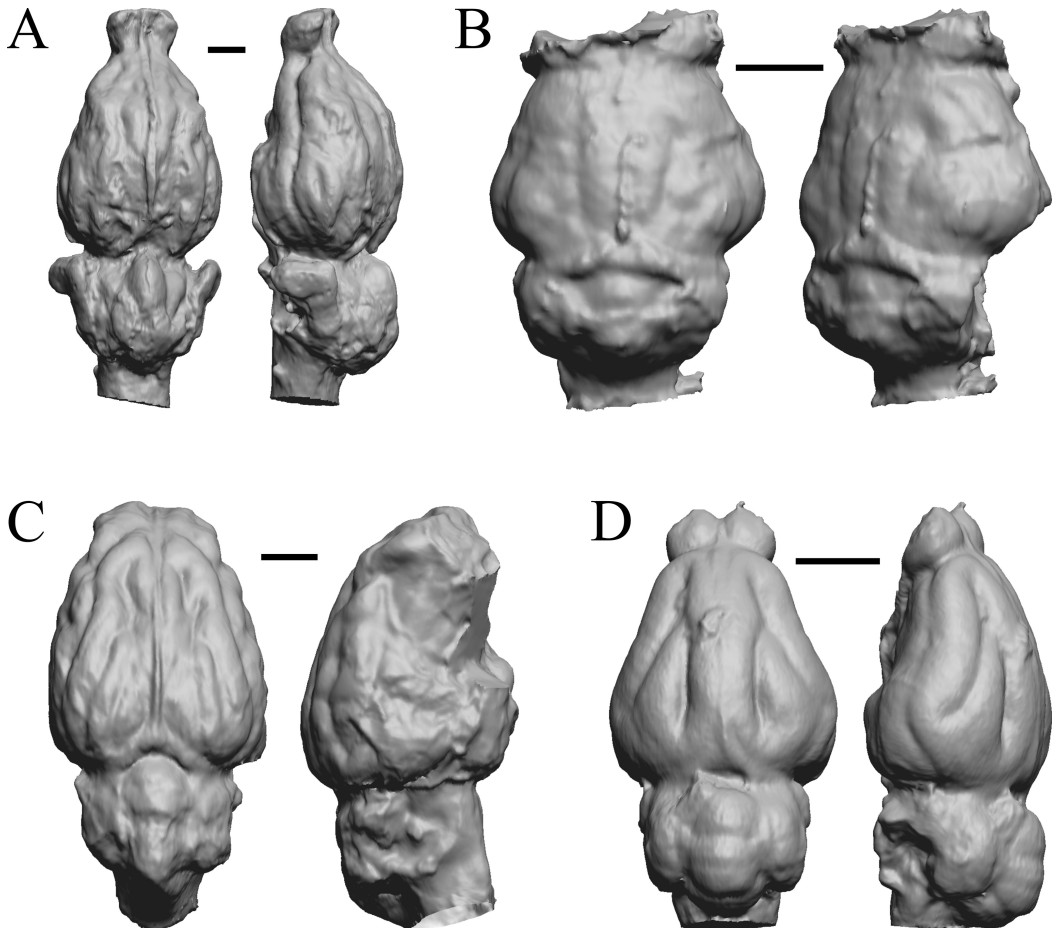

**Figure 14  *Anoplotherium, Patriomanis, Poebrotherium, Bathygenys* endocasts.** (A) *Anoplotherium commune* (NHMUK PV M 3753) in dorsal (left) and left lateral (right) views with rostral pole pointed at the top of the figure. (B) *Patriomanis americana* (AMNH FM 78999 = FMNH PM 57103) in dorsal (left) and right dorsolateral (right) views with rostral pole pointed at the top of the figure. (C) *Poebrotherium* (AMNH F:AM 31700 = FMNH PM 59167) in dorsal (left) and right lateral (right) views with rostral pole pointed at the top of the figure. (D) *Bathygenys reevesi* (TMM TXVP 40209-431) in dorsal (left) and left lateral (right) views with rostral pole pointed at the top of the figure. Scale bars = one cm. Further details may be found in Additional Information.

## Mammalian olfactory bulbs

Evolutionary studies of amniote olfactory bulbs are a challenge, especially for non-mammals, because olfactory bulbs may be broken, distorted, or not visible on natural endocasts. Because of the uncertainties, our analyses of neocorticalization exclude the olfactory bulbs from the measurement of the endocast. However, all digitized endocast files and associated figures in this paper include the olfactory bulb for potential future study. For additional information on specific cases, see Supplemental Information 1B.

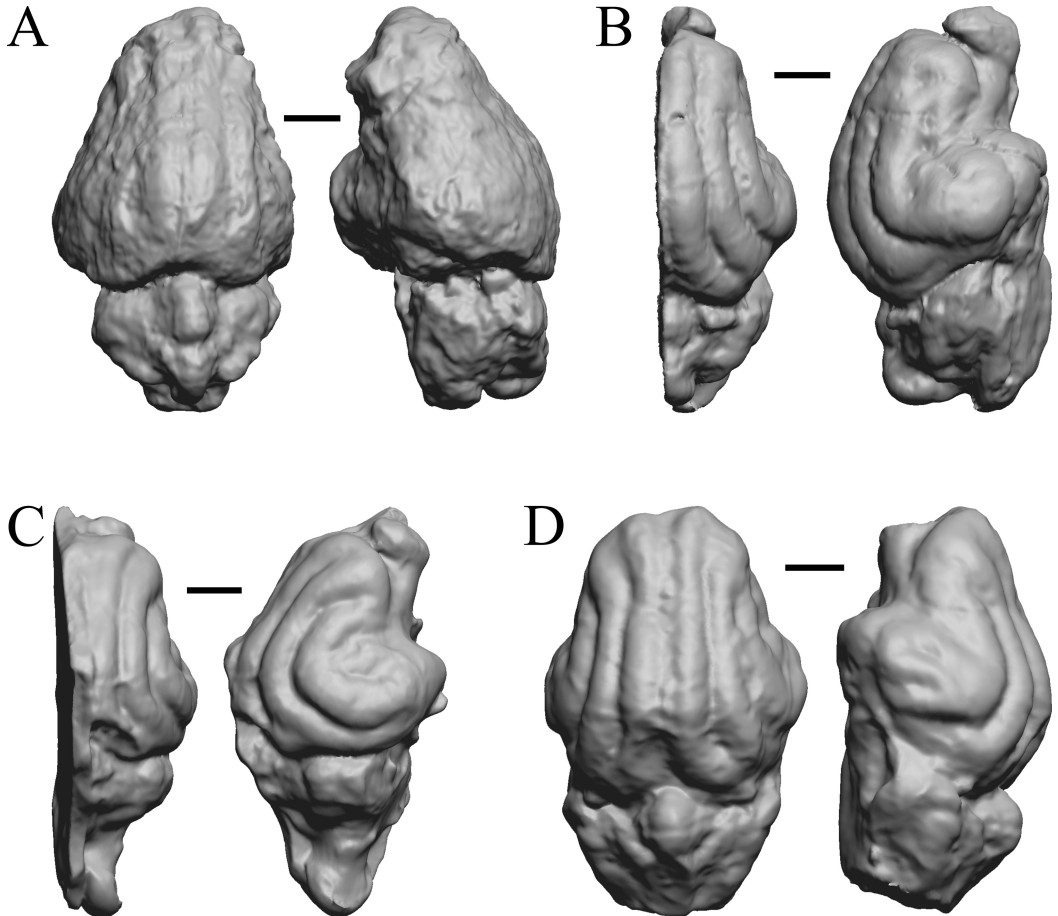

**Figure 15** *Daphoenus, Dinictis, Eusmilus, Hoplophoneus* **endocasts.** (A) *Daphoenus vetus* (FMNH PM UM1) in dorsal (left) and left lateral (right) views with rostral pole pointed at the top of the figure; olfactory bulbs were not preserved in the specimen. (B) *Dinictis felina* right hemisphere (SDSM 2431 = FMNH PM 58866) in dorsal (left) and right lateral (right) views with rostral pole pointed at the top of the figure. (C) *Eusmilus bidentatus* (FMNH PM 58871) in dorsal (left) and right lateral (right) views with rostral pole pointed at the top of the figure; only about half of the brain is present in the right hemisphere. (D) *Hoplophoneus primaevus* (USNM Paleobiology V 22538) in dorsal (left) and left lateral (right) views with rostral pole pointed at the top of the figure; olfactory bulbs not recovered. Scale bars = one cm.

## Midbrain exposure in mammals

Dorsal midbrain (tectal) exposure is striking in the koala (*Phascolarctos cinereus*) brain (*Haight & Nelson, 1987*) but is obscured on the endocast by the overlying confluence of sinuses (Fig. 35A). The morphological and/or functional underpinnings(s) of this prominent tectum is/are not yet well understood. Interestingly, some, but not all, extant bats have visible tecta on their endocasts (*Maugoust & Orliac, 2023*), and the colliculi in at least some extinct bats are visible in endocasts (*Maugoust & Orliac, 2021* and sources therein). Thus, we find no predictable pattern for when or how to model an exposed dorsal

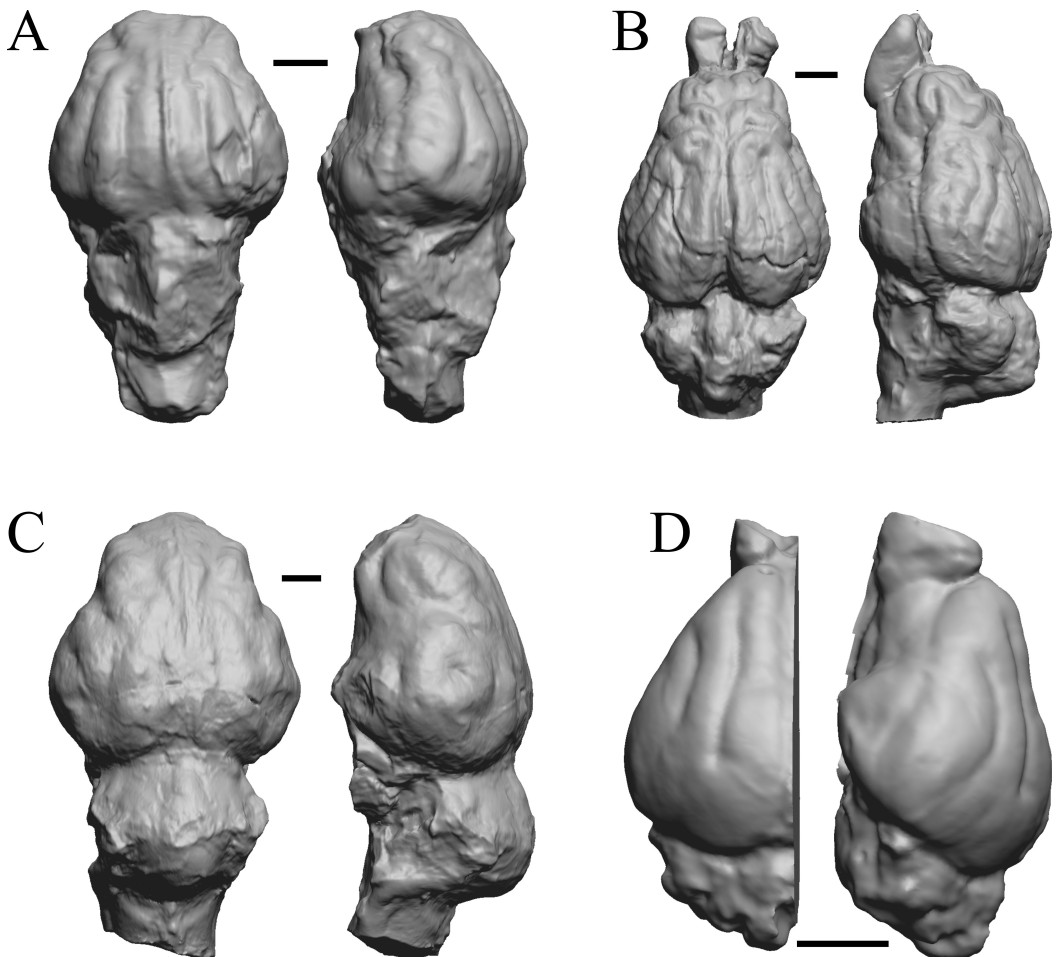

**Figure 16** *Merycoidodon, Mesohippus, Promerycochoerus, Hesperocyon* **endocasts.** All endocasts in dorsal (left) and left lateral (right) views with rostral pole pointed at the top of the figure. (A) *Merycoidodon culbertsoni* (FMNH PM UM3); the scanned endocast did not include an olfactory bulb region. (B) *Mesohippus bairdi* (AMNH FM 9814 = FMNH PM 59221). (C) *Promerycochoerus superbus* (YPM VP 11002 = FMNH PM 59072). (D) *Hesperocyon gregarius* (FMNH PM 58989). Scale bars = one cm. Further details may be found in Additional Information.

midbrain in extinct mammalian taxa unless there is direct evidence for an exposed tectum on the endocast (*e.g.*, four separate bumps for the corpora quadrigemina).

## Increased neocorticalization in mammals

We graph increase in neocortical surface ratio (forebrain surface area/total surface area of the brain or endocast) over geologic time in Fig. 38. In Fig. 38, extant species line up as the vertical column of points at 0 Ma. At 60 Ma ($X = -60$), the average neocorticolization for mammals sampled is 15%. Indeed, we reasonably approximate the earliest sampled extinct species, *Arctocyon* and *Titanoides*, as neocorticalized at 22.5% (compare to 10.3% with olfactory bulbs included in *Bertrand et al., 2022*) and 14.1%, respectively. Today (0

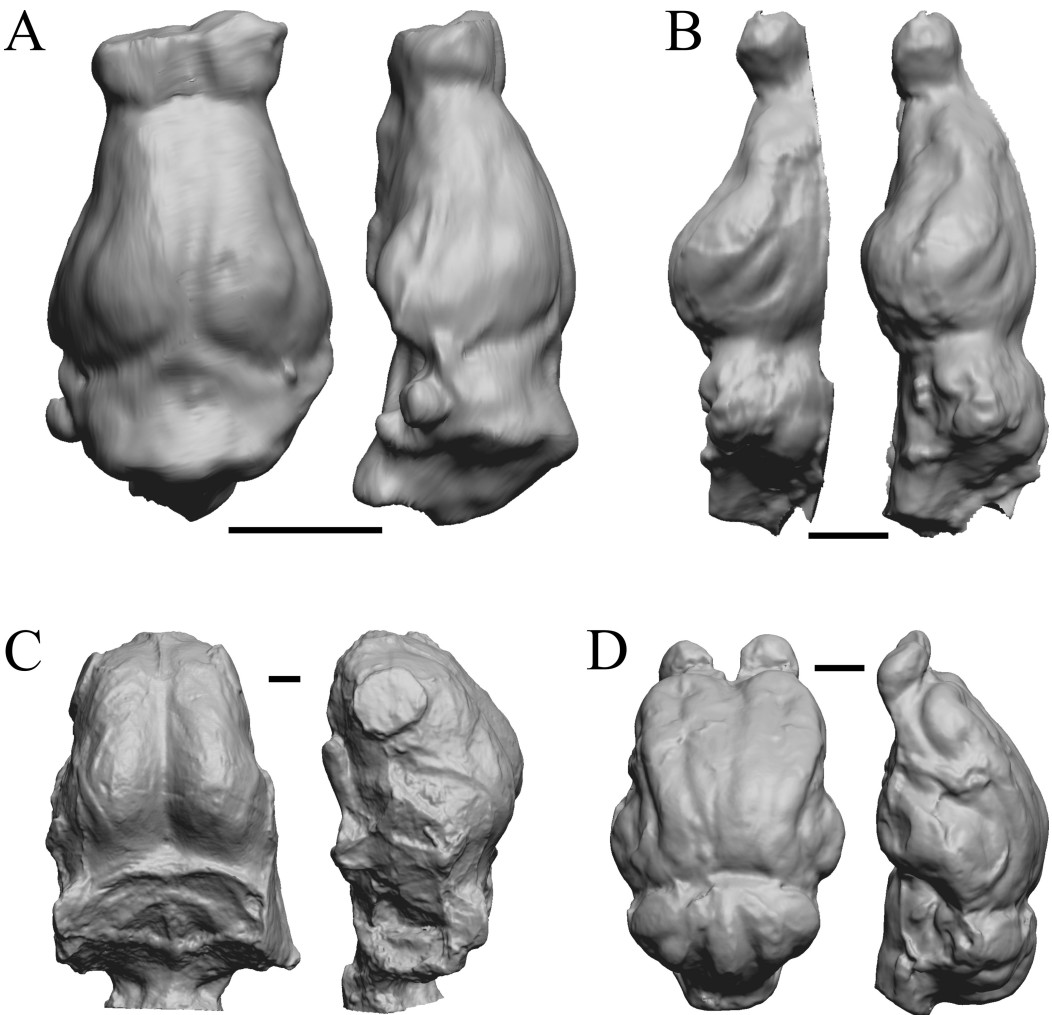

**Figure 17** *Leptictis (Ictops), Leptauchenia, Halitherium, Hapalops* **endocasts.** All endocasts in dorsal (left) and left lateral (right) views with rostral pole pointed at the top of the figure. (A) *Leptictis* (=*Ictops acutidens* Douglass). (B) *Leptauchenia decora* left hemisphere (AMNH FM 627 = FMNH PM 59074). (C) *Halitherium schinzi* (SMF M 3921); the endocast had a small postorbital extension, which was removed. (D) *Hapalops* sp. (Harry Jerison's personal collection, attributed to LACM). Scale bars = one cm.

Ma, $X = 0$), average neocorticaliation is 58%. These results are similar to the preliminary study of *Jerison (2012)* that reported an increase of 5% neocorticalization per 10 million years. For further useful comparisons and updates to these findings, see *Bertrand et al., 2022* and *Bertrand et al., 2024a*.

Briefly, we note here that the euprimates included in our analysis and that of *Jerison (2012)* come out to be "above average" with respect to neocortical size. However, we defer to the results and discussion of *Long, Bloch & Silcox (2015)* for more in-depth consideration and analyses of stem primates.

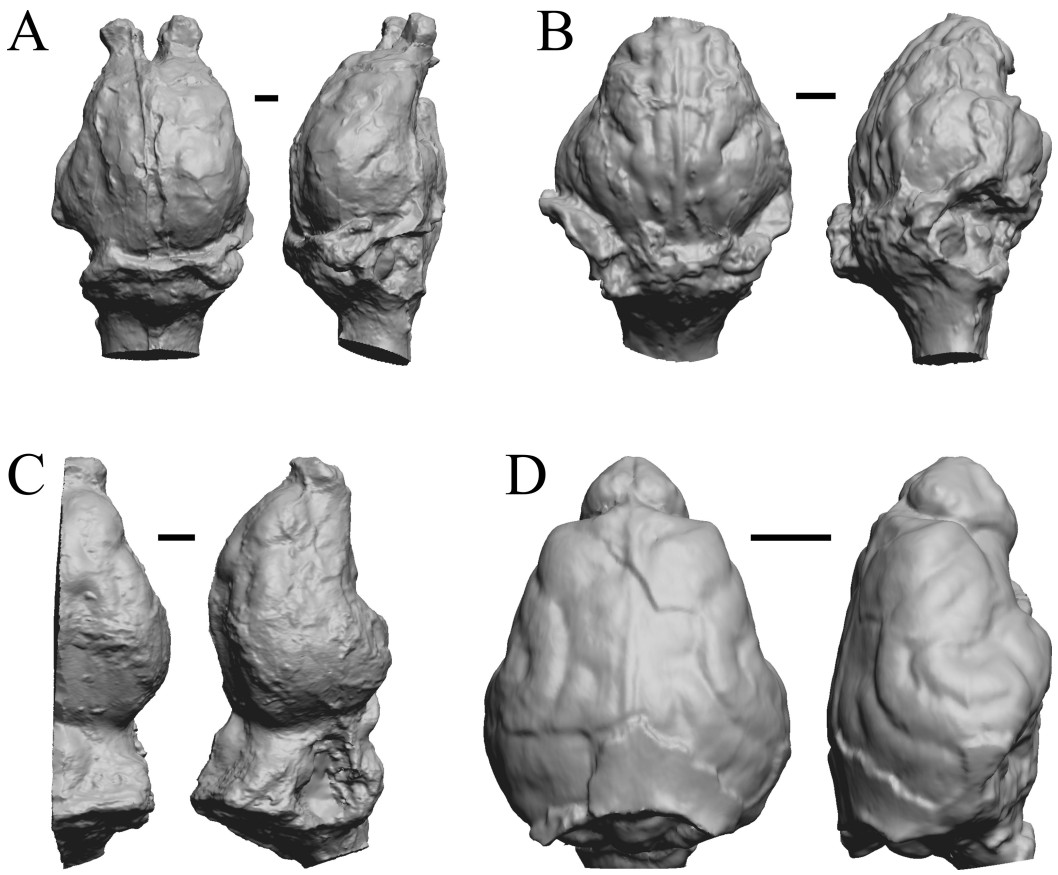

**Figure 18** *Leontinia, Rhynchippus, Archaeotherium, Promartes* **endocasts.** All endocasts in dorsal (left) and right lateral (right) views with rostral pole pointed at the top of the figure. (A) *Leontinia gaudryi* (FMNH P 13285). (B) *Rhynchippus equinus* (FMNH P 13410). (C) *Archaeotherium mortoni* right hemisphere (YPM VPPU 10908 = FMNH PM 59061). (D) *Promartes olcotti* (FMNH P 25233). Scale bars = one cm.

## Neocorticalization and encephalization in mammals

As similarly reported in *Jerison (2012)*, graphing mammalian neocortical surface area against *EQ* (Fig. 39) demonstrates that neocortal expansion plateaus at about 80% once EQ reaches ~2.0. That this plateau exists suggests that neocorticalization is constrained by factors unrelated to brain size, which itself shows no upper-limit fall-off. Certainly, topics of neural packing constraints (*e.g.*, *Assaf et al., 2020*) and the phylogenetic conservation of mammalian order connectomes (*Suarez et al., 2022*), scaling of brain matter composition (*Ardesch et al., 2022*), and neuronal wiring costs in mammals (*Huang & Yu, 2023*) are hot topics in the literature. We look forward to future illuminating discoveries in these fields.

## Mammalian encephalization

As a between-species trait, brain size is determined primarily by body size, and that is its "allometric" factor. Further, mammalian relative brain size enlarged beyond expectations

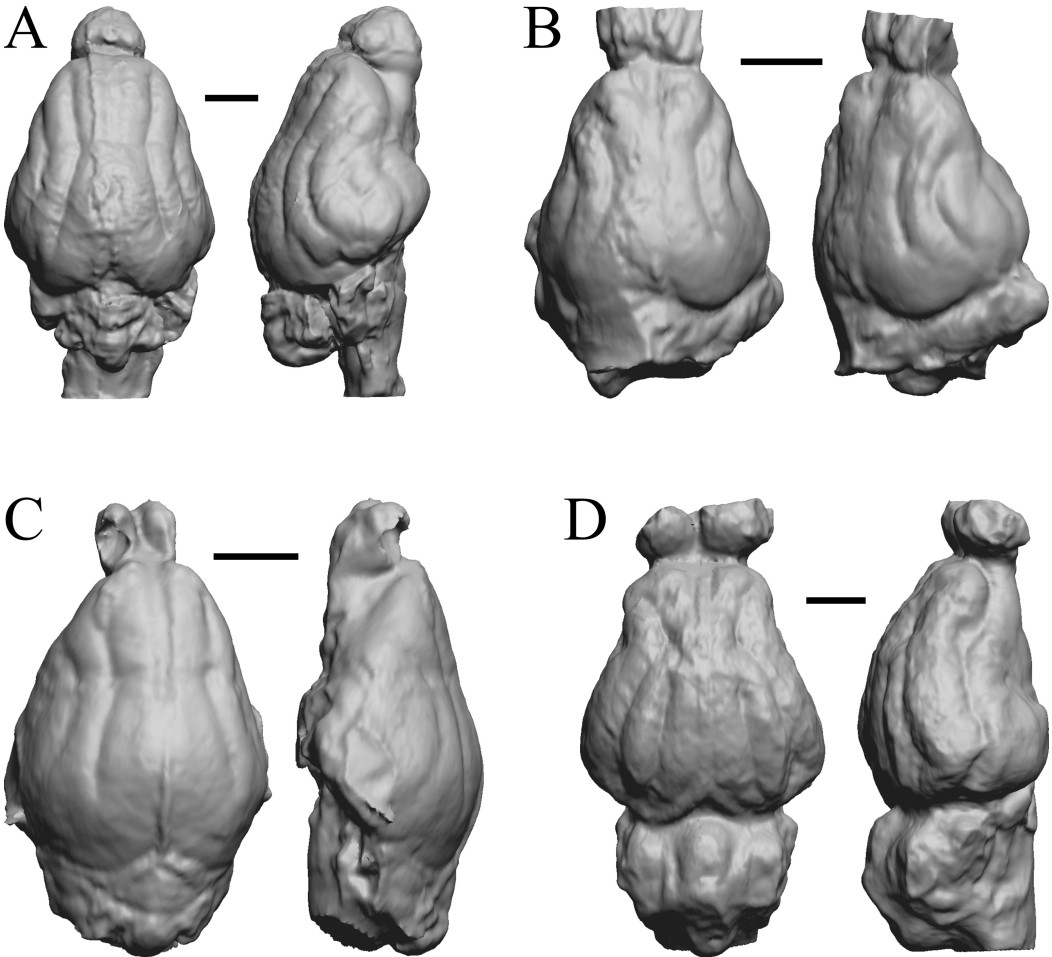

**Figure 19  *Mesocyon, Mustelictis, Leptocyon, Eporeodon* endocasts.** (A) *Mesocyon coryphaeus* (AMNH FM 6946 = FMNH PM 58979) in dorsal (left) and right lateral (right) views with rostral pole pointed at the top of the figure. (B) *Mustelictis piveteaui* (FMNH PM 58907) in dorsal (left) and right dorsolateral (right) views with rostral pole pointed at the top of the figure. (C) *Leptocyon* sp. (FMNH PM 58961 = F:AM 49063) in dorsal (left) and left lateral (right) views with rostral pole pointed at the top of the figure. (D) *Eporeodon socialis* (YPM VP 13118 = FMNH PM 59076) in dorsal (left) and right lateral (right) views with rostral pole pointed at the top of the figure. Scale bars = one cm.

set by the trend of non-mammalian vertebrates (Fig. 40). (Notably, independent variations in body size impact changes in relative brain size, too.) Importantly, we remind readers that this result should not be taken at face-value or as a novel finding because our analysis does not include data or results from recent studies on the evolution of relative brain size in mammals (*e.g., Bertrand et al., 2022*). About 80% of the variance in brain size in extant mammals is attributable to body size differences (Fig. 40B; $r = 0.81$). The residual from the allometric regression of log brain size on log body size is the statistic that defines an encephalization quotient (EQ). Thus, across species, EQ presumably describes at least some of the remaining 20% variance, which is evidenced by the variability of EQ in our sample.

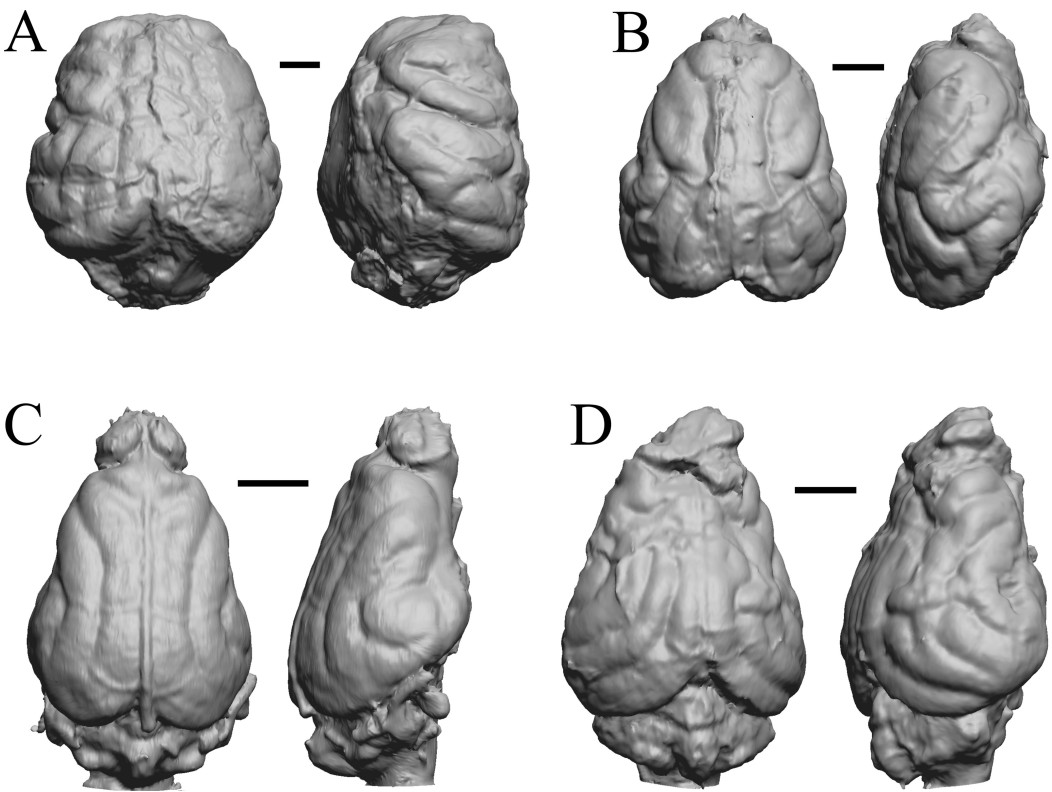

**Figure 20** *Enaliarctos, Potamotherium, Plesiogale, Zodiolestes* **endocasts.** (A) *Enaliarctos* sp. (FMNH PM 57161) in dorsal (left) and left ventrolateral (right) views with rostral pole pointed at the top of the figure; olfactory bulbs and partial hindbrain missing. (B) *Potamotherium valetoni* (NHMUK PV M 29357 = FMNH PM 58906) in dorsal (left) and right lateral (right) views with rostral pole pointed at the top of the figure. (C) *Plesiogale paragale* (NMB M.A.4641) in dorsal (left) and right lateral (right) views with rostral pole pointed at the top of the figure. (D) *Zodiolestes daimonedlixensis* (FMNH P 12032) in dorsal (left) and right lateral (right) views with rostral pole pointed at the top of the figure. Scale bars = one cm.

## Mammal-reptile boundary

Brain-body relationships in large numbers of living amniotes (mammals, $N = 647$; birds, $N = 219$; and reptiles, $N = 59$) have been described historically using convex polygons to better understand inter-class allometric relationships (*Jerison, 2007*; Fig. 40A). Extant birds and mammals show similar encephalization, with the bird polygon overlapping a portion of the larger mammalian polygon. Results for extant reptiles show them to be less encephalized (see also *Van Dongen, 1998* for larger sample size), and their polygon rests below those for birds and mammals. Non-amniote vertebrates (not graphed in Fig. 40) fall within or below the reptile polygon (further evidence: *Van Dongen, 1998*, *Jerison, 2001b*). Electric fish, however, are within the mammalian range, cartilaginous fish overlap the reptilian and mammalian ranges, and ''agnathans'' form a small polygon at the lower margin of the main fish polygon (for details, see *Jerison, 2000*).

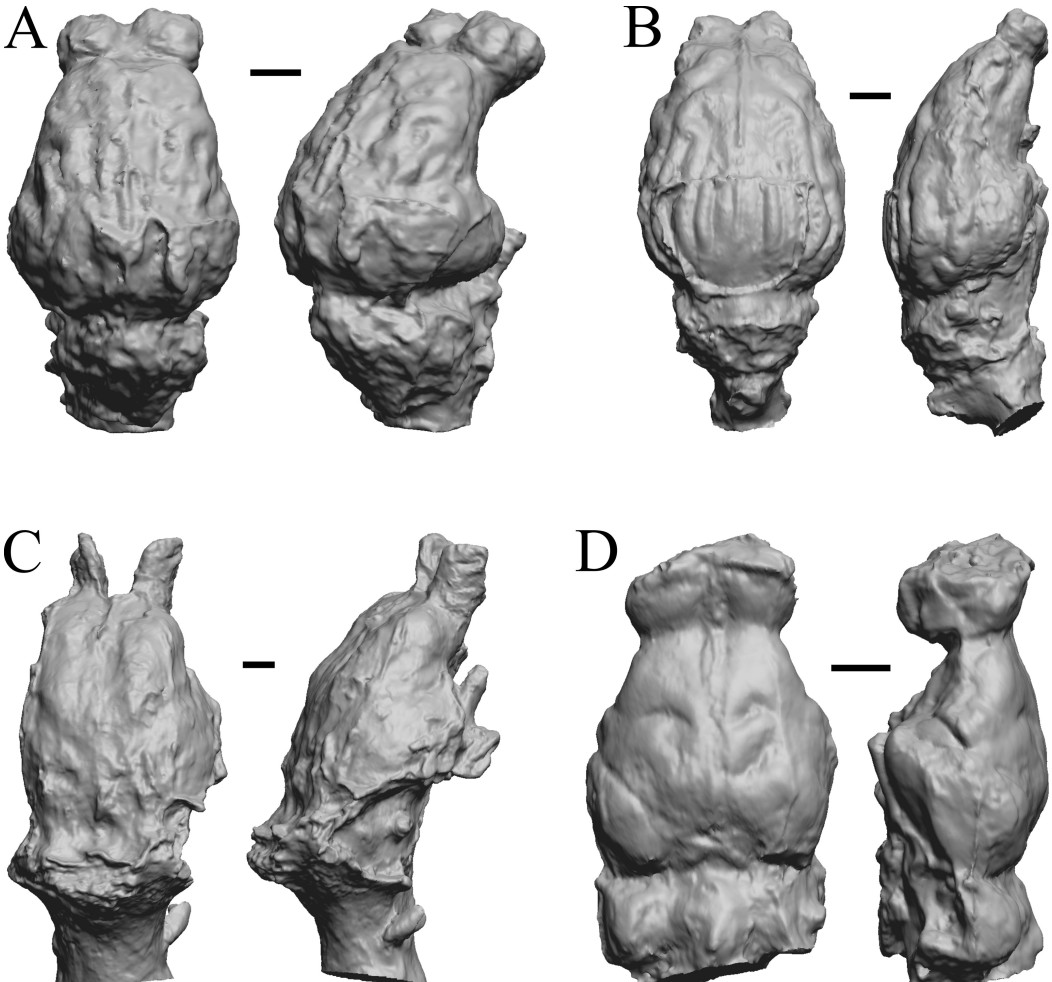

**Figure 21 *Desmathyus (Hesperhyus), Oxydactylus, Homalodotherium, Borhyaena* endocasts.** (A) *Desmathyus* sp. (*Hesperhyus*) (CM VP 1423 =FMNH M 59066) in dorsal (left) and right dorsolateral (right) views with rostral pole pointed at the top of the figure. (B) *Oxydactylus longipes* (FMNH P 12117) in dorsal (left) and right lateral (right) views with rostral pole pointed at the top of the figure. (C) *Homalodotherium* sp. (FMNH PM 59291) in dorsal (left) and right dorsolateral (right) views with rostral pole pointed at the top of the figure. (D) *Borhyaena tuberata* (FMNH P 13266) in dorsal (left) and left lateral (right) views with rostral pole pointed at the top of the figure.

Re-examining these historical polygons in light of this perspective study, we added all digitized data and their regression line in Fig. 40B; only the fossil data and their regression line were added to Fig. 40C. Unsurprisingly, only a few of the extinct mammals fell below the lower boundary of the extant mammalian polygon, representing a potential "starting point" for the encephalization that took place as mammals evolved to reach the present lower limit.

However, the datum contributed by *Arctocyon primaevus* is surprising and represents a mammalian point falling within the (extinct) dinosaur polygon. This requires

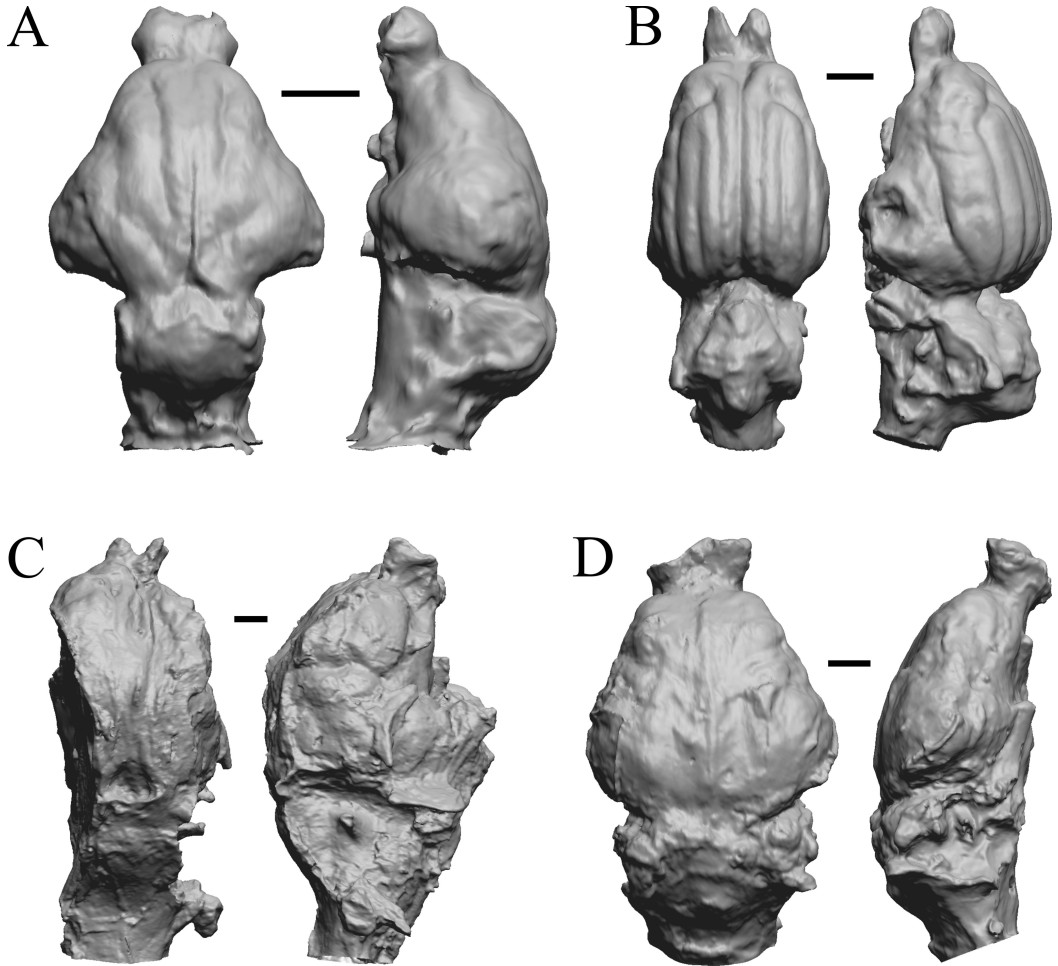

**Figure 22 *Protypotherium, Proterotherium, Nesodon, Merycochoerus* endocasts.** (A) *Protypotherium australe* (FMNH P 13046) in dorsal (left) and left lateral (right) views with rostral pole pointed at the top of the figure. (B) *Proterotherium cavum* (AMNH FM 9245 = FMNH PM 59742) in dorsal (left) and left lateral (right) views with rostral pole pointed at the top of the figure. (C) *Nesodon imbricatus* (FMNH P 13076) in dorsal (left) and right lateral (right) views with rostral pole pointed at the top of the figure; part of left hemisphere missing. (D) *Merycochoerus proprius* (AMNH FM 43016 A = FMNH PM 59081) in dorsal (left) and right lateral (right) views with rostral pole pointed at the top of the figure. Scale bars = one cm.

reconsideration of previous conclusions about the allometric border between mammals and reptiles (*Jerison, 1973*), as dinosaurs have hitherto been considered a natural extension of the polygon of extant reptiles. The datum on *Arctocyon* is robust, with the endocast prepared by *Russell & Sigogneau-Russell (1965)* quite brain-like (Fig. 6), and the measurements accurate. Although the body size was originally uncertain, a reanalysis of the skeletal material by *Argot (2013)* is definitive, making the body size estimate as good as it can be. Interestingly, and relevant to this finding for *Arctocyon*, *Bertrand et al. (2022)* found a temporary lag in

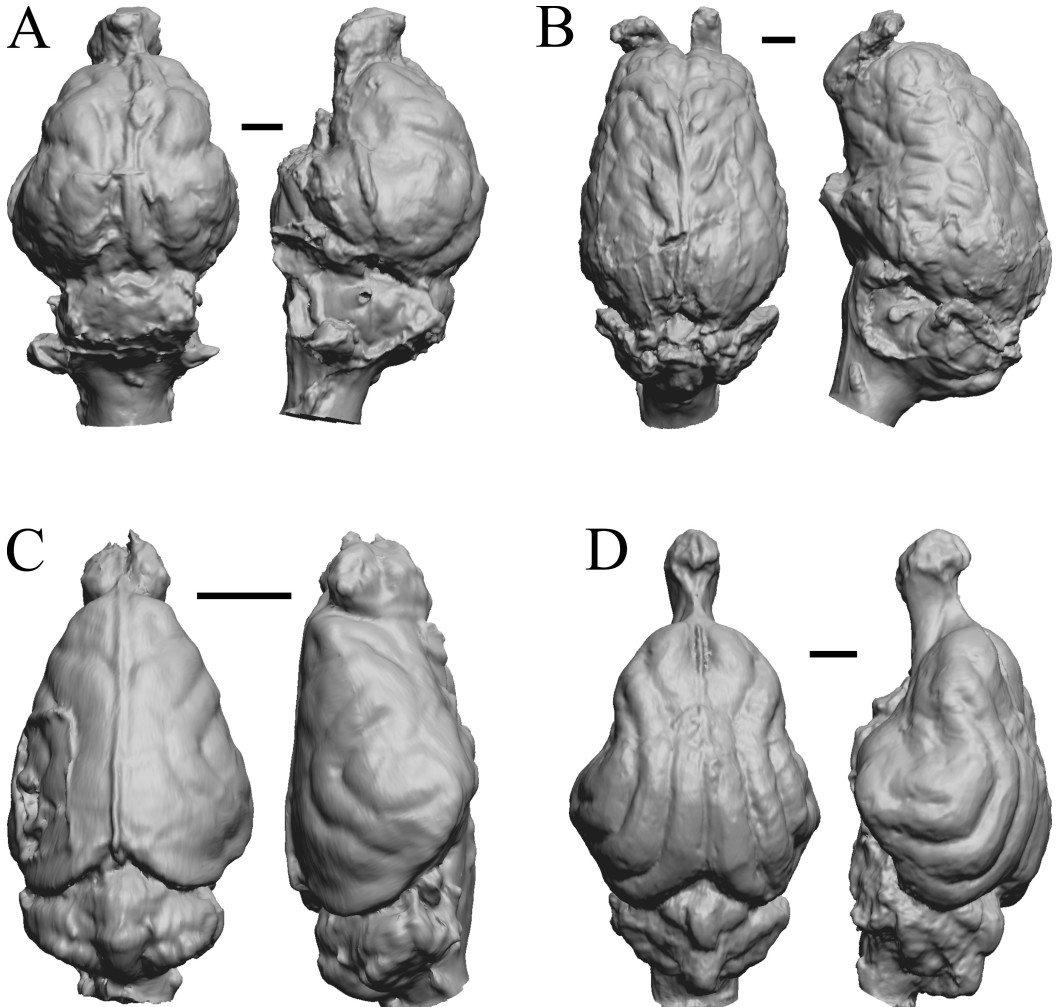

**Figure 23** *Adinotherium, Merychippus, Plionictis, Pseudaelurus* **endocasts.** (A) *Adinotherium ovinum* (FMNH P 12986) in dorsal (left) and left lateral (right) views with rostral pole pointed at the top of the figure. (B) *Merychippus isonesus* (AMNH FM 71150 = FMNH PM 59208) in dorsal (left) and left lateral (right) views with rostral pole pointed at the top of the figure. (C) *Plionictis* sp. (AMNH FM 25314 = FMNH PM 58945) in dorsal (left) and right lateral (right) views with rostral pole pointed at the top of the figure. (D) *Pseudaelurus validus* (AMNH FM 61835 = FMNH PM 58867) in dorsal (left) and left lateral (right) views with rostral pole pointed at the top of the figure. Scale bars = one cm. Further details may be found in Additional Information.

relative brain size for placental mammals in the Paleocene as compared to the Mezozoic due to body size increasing prior to brain size.

Therefore, the area of the polygon drawn for dinosaurs requires reconsideration. Fig. 40D summarizes a new view of mammal-dinosaur allometric relationships. HJJ redrew the upper boundary of the reptile-dinosaur convex polygon, and instead of connecting foci of the extant reptile polygon to a convex polygon that included speculations about
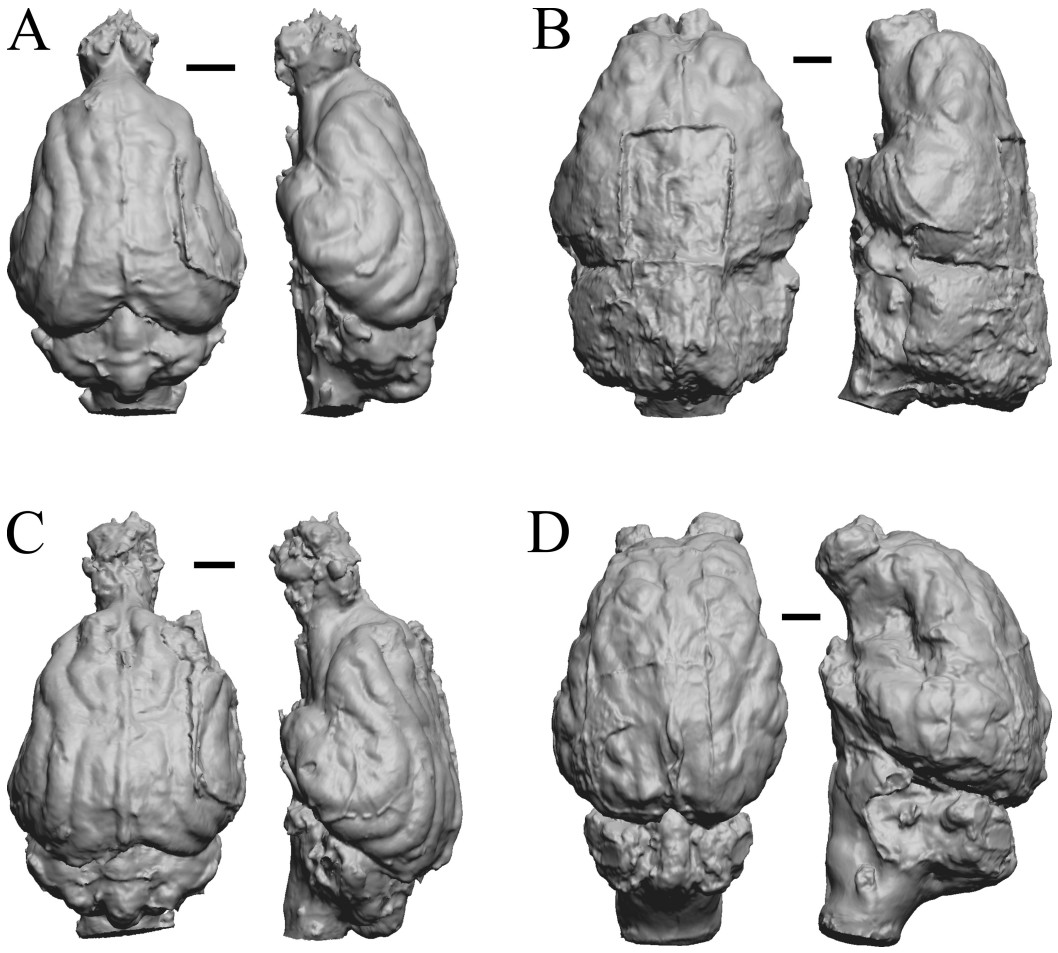

**Figure 24 *Paracynarctus, Ustatochoerus, Carpocyon ("Osteoborus"), Pseudhipparion* endocasts.** All endocasts in dorsal (left) and left lateral (right) views with rostral pole pointed at the top of the figure. (A) *Paracynarctus sinclairi* (AMNH FM 61009 = FMNH PM 58973). (B) *Ustatochoerus profectus* (AMNH FM 33617 = FMNH PM 59071); (C) *Carpocyon webbi* (AMNH FM 61328 = FMNH PM 58964). (D) *Pseudhipparion gratum* (AMNH FM 70025 = FMNH PM 59211). Scale bars = one cm.

dinosaur brain sizes, in Fig. 40D, HJJ extended the reptile polygon (dotted line) to include larger body sizes. The earlier drawing assumed that dinosaur brains were half the volume of their endocasts and bounded their assumed brain sizes. The new extended boundary of the reptile polygon is a brain boundary (not a brain-endocast boundary) assumes that living reptile brain sizes would best estimate dinosaur brain sizes, with a parallel lower boundary drawn through the smallest reptile brain sizes to complete the new convex polygon. This finding is generally corroborated by the findings of *Morhardt* (*2016*, chapter 3), which incorporate modern phylogenetic methods.

The newly drawn boundaries do not depend on prior estimates of brain-endocast relationships in dinosaurs. Rather, they assume that dinosaur brains would follow similar

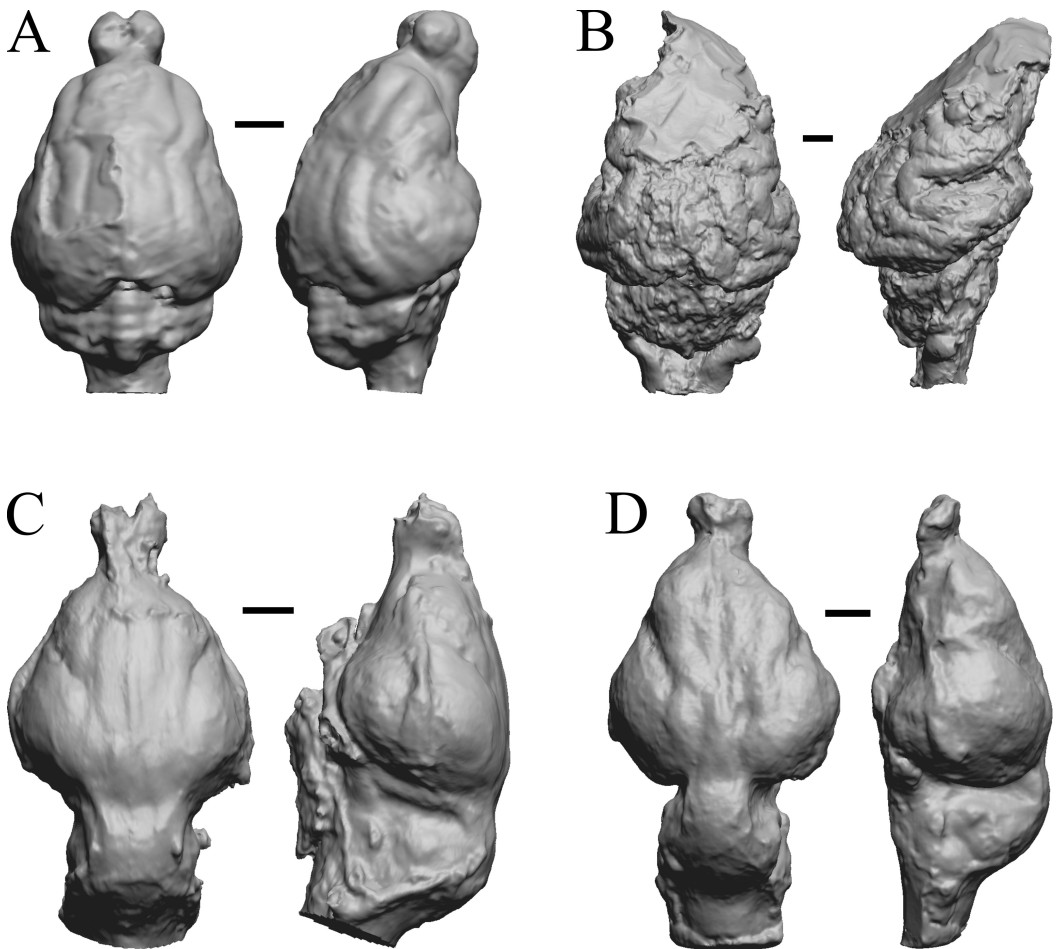

**Figure 25** ***Paratomarctus, Hemicyon, Pseudotypotherium, Tyopotheriopsis* endocasts.** (A) *Paratomarctus euthos* (AMNH FM 61074) in dorsal (left) and right dorsolateral (right) views with rostral pole pointed at the top of the figure. (B) *Hemicyon cf. barbouri* (AMNH FM 25530 = FMNH PM 59030) in dorsal (left) and right lateral (right) views with rostral pole pointed at the top of the figure. (C) *Pseudotypotherium pseudopachygnathum* (AMNH FM 14509 = FMNH PM 59292) in dorsal (left) and left lateral (right) views with rostral pole pointed at the top of the figure. (D) *Typotheriopsis internum* (FMNH P 14420) in dorsal (left) and left lateral (right) views with rostral pole pointed at the top of the figure. Scale bars = one cm.

size rules as living reptile brains (again, see *Morhardt, 2016*, chapter 3; but see *Caspar et al., 2024* for updates and further details on patterns for specific dinosaur groups and their impact on the findings of this study). In Fig. 40D, fifty-nine data points from extant reptiles (*Platel, 1979*) were overlain on the reptile polygon to help visualize how the polygon was drawn, as well as to show that the original polygon, established from far fewer data points (*Jerison, 2007*), remains useful. The maxima of the extant reptile polygon are a 134 kg crocodile (*Crocodylus acutus*) and a 205 kg alligator (*Alligator mississippiensis*), with their brains measuring 15.6 g and 14.08 g, respectively. Using these new boundaries, the lowest

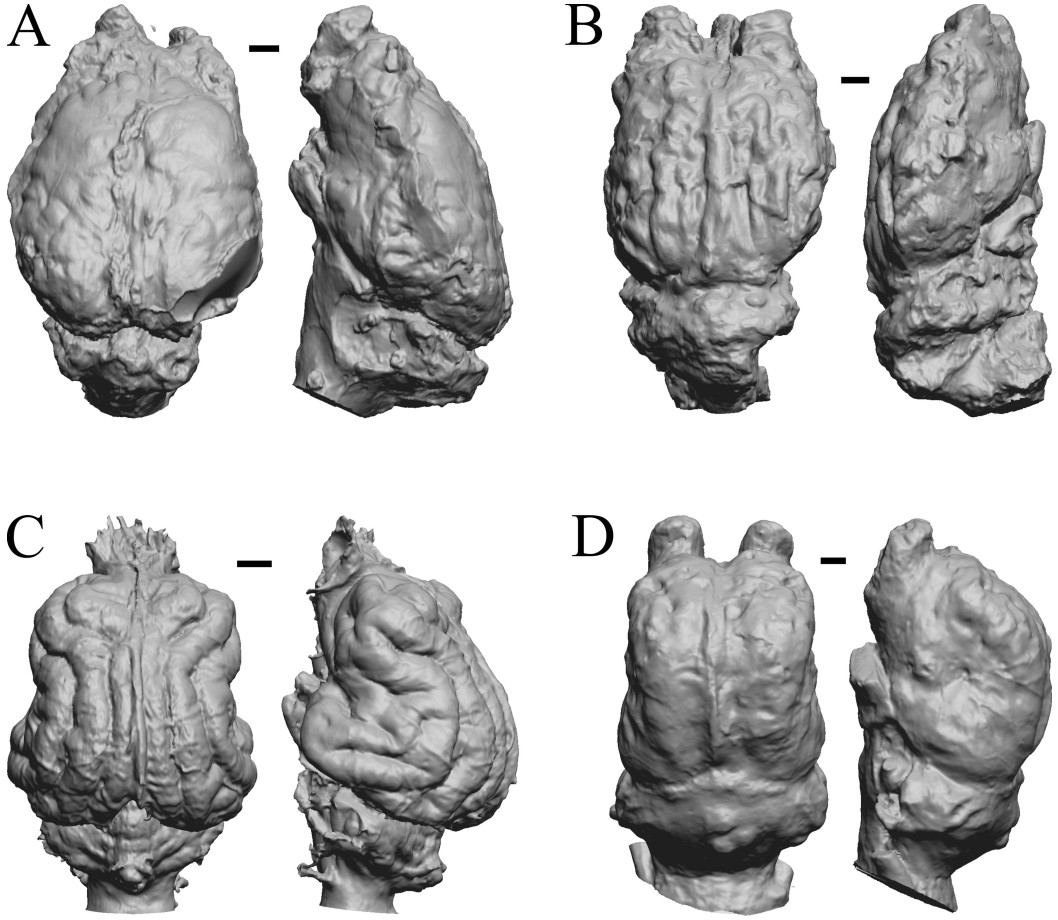

**Figure 26** *Cormohipparion, Procamelus, Homotherium, Mylodon* **endocasts.** (A) *Cormohipparion occidendale* (AMNH FM 71886 = FMNH PM 59220) in dorsal (left) and left lateral (right) views with rostral pole pointed at the top of the figure. (B) *Procamelus grandis* (AMNH FM 40425 = FMNH PM 59160) in dorsal (left) and right lateral (right) views with rostral pole pointed at the top of the figure. (C) *Homotherium* sp. (AMNH FM 95297 = FMNH PM 58891) in dorsal (left) and left lateral (right) views with rostral pole pointed at the top of the figure. (D) *Mylodon* sp. *Owen* (1840) (LACM 157696) in dorsal (left) and left lateral (right) views with rostral pole pointed at the top of the figure. Scale bars = one cm.

mammalian point, *Arctocyon primaevus*, now lies above the reptile polygon (Fig. 40D), still supporting the hypothesis that there is a distinct mammal-reptile boundary.

## Exponential increases in mammalian cortical surface area

Surface-area-to-volume relationships for extant and extinct mammal taxa are presented in Fig. 41, with endocast surfaces of all sampled taxa (extinct and extant) regressed against their respective endocast volumes in Fig. 41A, and *cortical* surface areas of all sampled extant taxa regressed against brain size (volume) in Fig. 41B. The regression equation in Fig. 41B is a power function with the exponent 0.91. That this exponent is much greater than 2/3 (*i.e.,* consistent scaling of a 3D object and its surface area; see exponent in Fig. 41A) shows

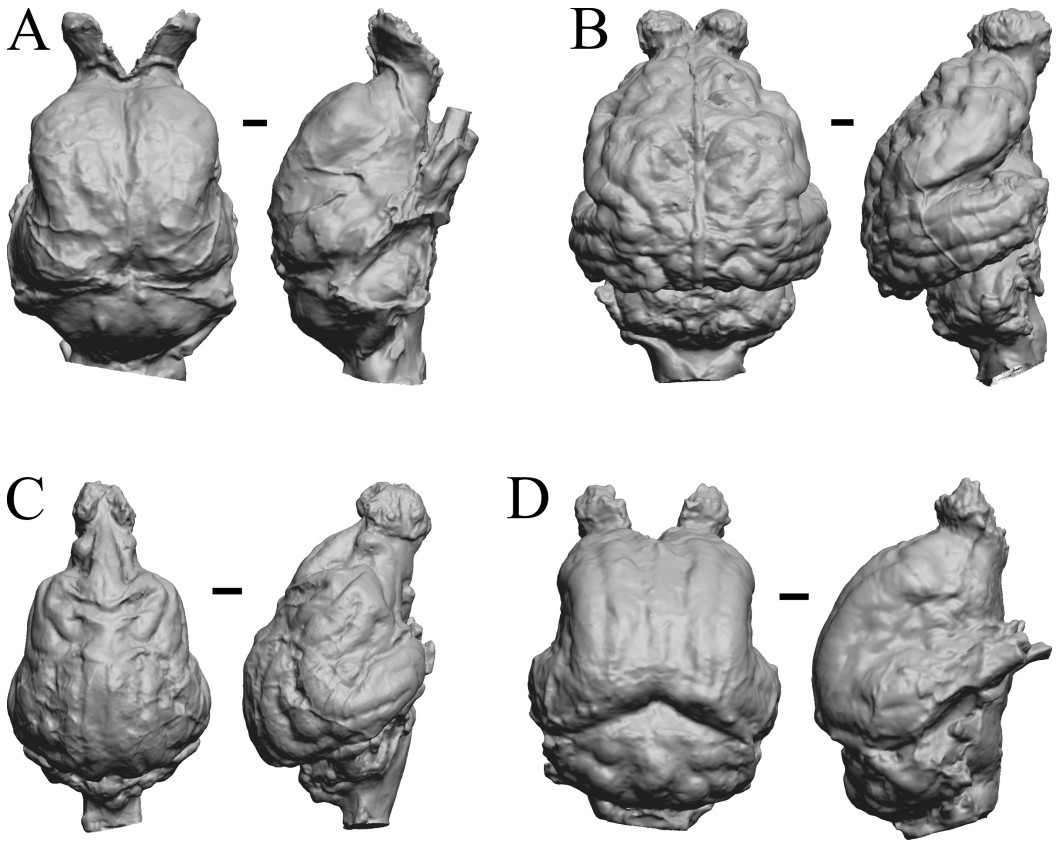

**Figure 27** *Glossotherium, Arctodus, Aenocyon dirus, Megalonyx* **endocasts.** All endocasts in dorsal (left) and right lateral (right) views with rostral pole pointed at the top of the figure. (A) *Paramylodon harlani* (LACMHC 1717-33). (B) *Arctodus simus* (*Tremarctotherium*) (FMNH PM 59022, attributed in HJJ's notes to LACM). (C) *Aenocyon dirus* (LACMHC 2300-82). (D) *Megalonyx jeffersoni* (Harry Jerison's personal collection). Scale bars = one cm.

that as brain size increases, the *rate* at which surface area increases accelerates in a highly predictable ($r = 0.996$) fashion. We attribute this rate change to exponential increases in convolutedness. To clarify, although convolutedness appears here to be almost entirely a function of brain size, there is further evidence that species do differ in convolutedness, at least at the ordinal level (*e.g.*, see manatees and beavers in this study). The difference is small (*Pillay & Manger, 2007*), and it reflects the patterning of convolutions in orders of mammals (*Welker, 1990*; *Van Essen, 1997*), in particular of ungulates compared to other orders. Another caveat to consider here is the possibility for interspecific variation in convolution patterns on endocasts (*Welker, 1990*).

## Proper mass

As stated earlier, despite general consistency in mammalian brain:brain region scaling, instances have been identified in which brain regions undergo statistically significant size change relative to other closely related species and in response to evolutionary changes

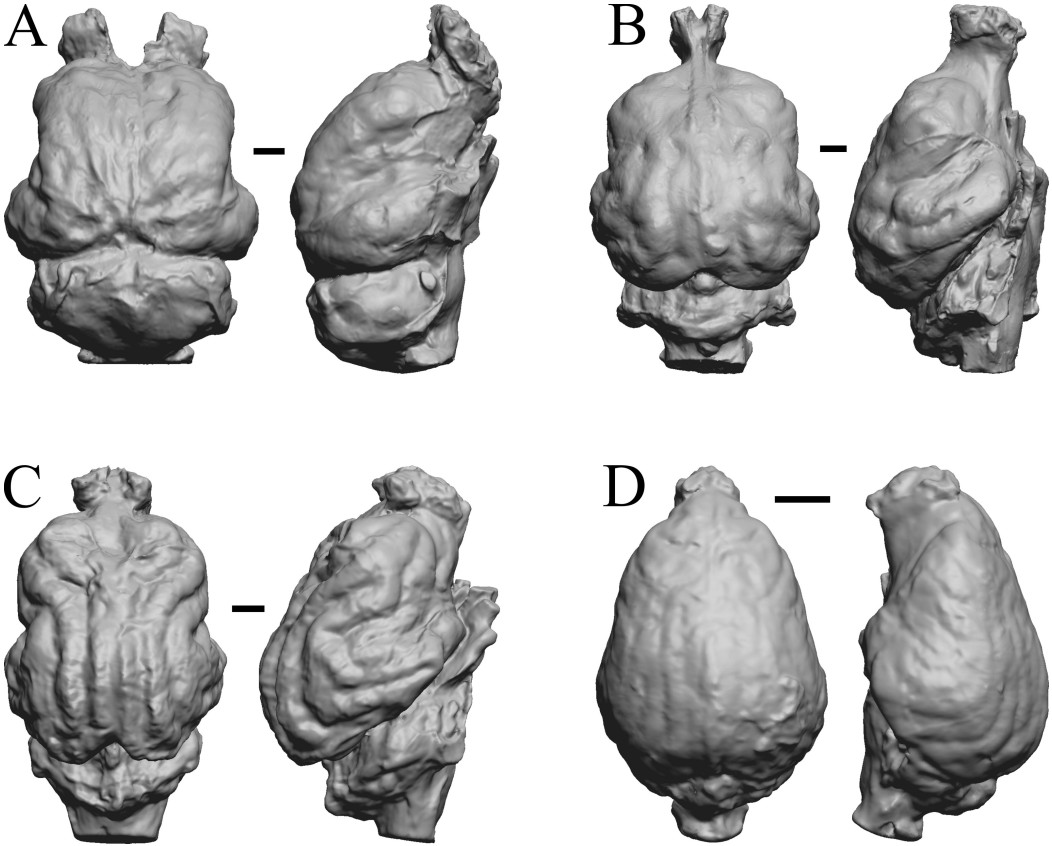

**Figure 28** ***Nothrotheriops, Panthera, Smilodon, Urocyon* endocasts.** (A) *Nothrotheriops shastensis* (LACMHC 1800-6) in dorsal (left) and right lateral (right) views with rostral pole pointed at the top of the figure. (B) *Panthera atrox* (LACMHC 2900-1) in dorsal (left) and right lateral (right) views with rostral pole pointed at the top of the figure. (C) *Smilodon fatalis* (LACMHC 2001-199) in dorsal (left) and right lateral (right) views with rostral pole pointed at the top of the figure. (D) *Urocyon cinereoargenteus* (UCMP V 12263) in dorsal (left) and left lateral (right) views with rostral pole pointed at the top of the figure. Scale bars = one cm.

in function and/or behaviors (*i.e.,* "principle of proper mass" (PPM); *e.g., Jerison, 1973; Jerison, 2001a; Jerison, 2001b; Butler & Hodos, 2005*). The most dramatic of these features in mammals is the evolutionary enlargement of forebrain and neocortex. One caveat to PPM captured in this paper is the lack of difference in cortical convolutions between raccoons (*Procyon lotor*) and coatimundi (*Nasua narica*) despite validated differences in their foraging styles and how those styles map electrophysiologically in the brain (*Welker & Campos, 1963; Johnson, 1990; Welker, 1990;* but see *Boch et al. (2024)*) for a potential unifying hypothesis related to the expansion of the postcruciate gyrus). Thus, although the principle of proper mass is intuitive and often demonstrated, it is not always a foolproof assumption in the analysis of endocasts.

**Table 2** Intra-species analysis of selected *Bathygenys reevesi* specimens.

| I.D. # | E ml | S cm² | nc cm² | nc/S |
|---|---|---|---|---|
| 443D | 11.362 | 28.5 | 8.22 | 0.288421 |
| 443F | 10.98 | 31.5 | 8.64 | 0.274286 |
| 443H | 11.386 | 34 | 8.3 | 0.244118 |
| 443I | 11.914 | 34.5 | 8.32 | 0.241159 |
| 443J | 13.557 | 28.8 | 7.62 | 0.264583 |
| 443K | 13.981 | 30.8 | 8.24 | 0.267532 |
| 443L | 10.386 | 28.3 | 6.96 | 0.245936 |
| 443X | 11.422 | 25.8 | 7.1 | 0.275194 |
| Mean | 11.8735 | 30.275 | 7.925 | 0.26265363 |
| SD | 1.253082 | 2.994161 | 0.62094 | 0.01719269 |
| SD/M (CV) | 0.105536 | 0.098899 | 0.07835 | 0.06545764 |
| CV percentage | 10.60% | 9.90% | 7.80% | 6.50% |

**Notes.**

E, endocast volume; S, endocast surface area; nc, neocortex area; nc/S, neocortex re surface area; SD, standard deviation; M, mean; CV, statistical coefficient of variation.
Specimens are from the Texas Natural History Museum, where each specimen number is preceded by "TMM" (*e.g.*, TMM 443D is the specimen label at the museum).

## Within-species variation

A previous analog analysis of twenty natural endocasts—collected at the Reeves Fossil Bed in the Big Bend area of Texas, Chadronian, end of the Eocene (*Wilson, 1971*) and regarded as variants of a single species, *Bathygenys reevesi*—revealed that endocast volumes were between 10 and 12 ml and normally distributed, with a coefficient of variation (CV) of about 10% (*Jerison, 1979*). The same analysis showed that a CV of ~10% was also a good fit for the data of other extant and fossil brains and endocasts including house cats, chimpanzees, living and fossil equoids, and living and fossil hominins (*Jerison, 1979*). Here, we calculated CVs here using data from eight *Bathygenys* endocast specimens lacking olfactory bulbs for comparison with the general sample (Table 2). Using digitized endocast data, the CVs for endocast volume, surface area, neocortical area, and neocortical:surface area ratio were 10.6%, 9.9%, 7.8%, and 6.5%, respectively. This compares to a previous analysis of digitized CT images of 157 *Bathygenys* samples, which showed higher CVs for the length of olfactory bulbs (CV = 15.8%), width of the hypophyseal endocast (CV = 16.3%), and cerebellum (13%) (*Macrini, 2009*). Although higher, we conclude that these CV values are similar enough to the current results to raise no important questions about the adequacy of measurements on a single specimen of a single species to represent its brain as the information-processing organ. Certainly, the issue of within-species variation is question-specific, with high variability at the species level being more relevant to finer (*e.g.*, intra-Family) *versus* broader (*e.g.*, inter-Order) comparisons.

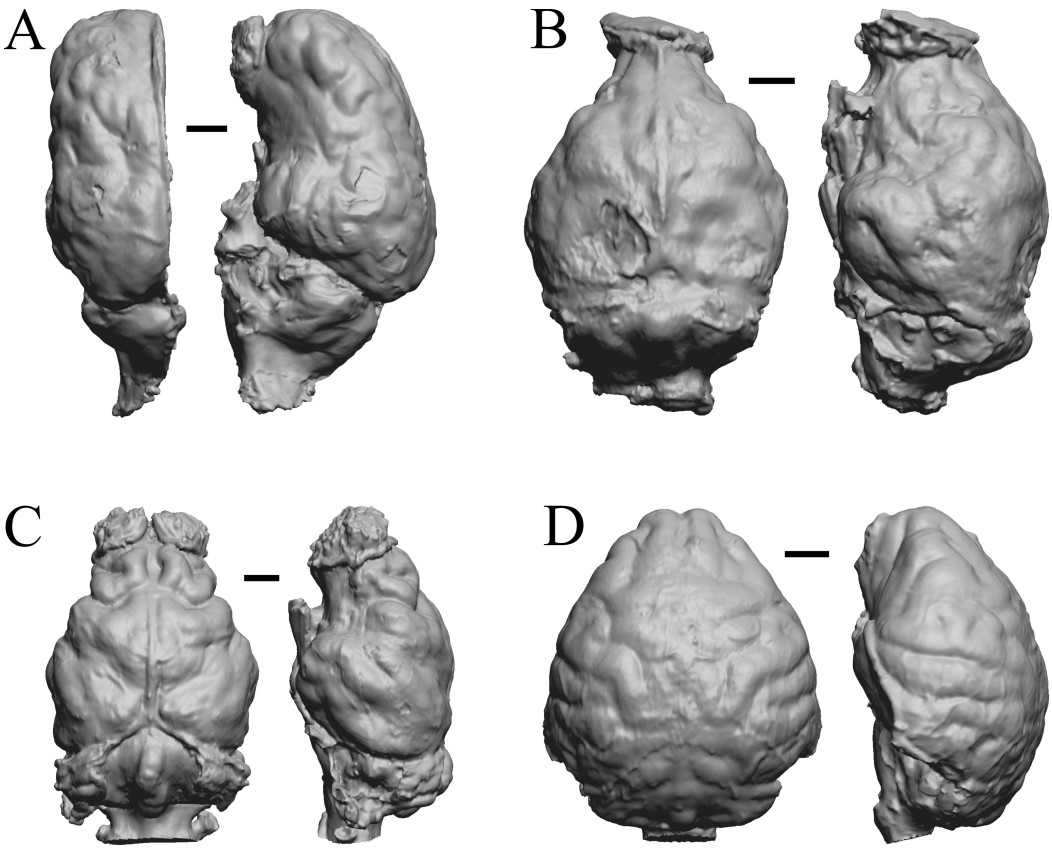

**Figure 29** *Platygonus, Sthenurus, Thylacoleo, Archaeolemur* **endocasts.** All endocasts in dorsal (left) and left lateral (right) views with rostral pole pointed at the top of the figure. (A) *Platygonus compressus* (CM VP 12888 = FMNH PM 59058. (B) *Sthenurus cf. orientalis* (FMNH PM 59245). (C) *Thylacoleo carniflex* (SAMA P18681 = FMNH PM 59244. (D) *Archaeolemur majori* (AMNH FM 30007 = FMNH PM 59258). Scale bars = one cm. Further details may be found in Additional Information.

## CONCLUSIONS

This perspective study offers researchers an opportunity to consider our large dataset (172 cranial endocast specimens and incorporating 41 extant species, of broad patterns of relative brain size and neocorticalization across vertebrate evolution) and historical analyses in future paleoneurologic research. Following earlier methods of HJJ, this perspective analysis shows that, on average, mammal neocorticalization increased at about 5% additional neocortex per 10 million years. About 60 million years ago, mammalian neocorticalization averaged about 20%, increasing to a present average of 50%, with a maximum at about 80% in primates reached within the past 10 million years. Compared to results of previous bivariate analyses, these data redefine the observed boundary between mammals and reptiles and confirm that measurements on a single species specimen adequately represent the brains of the entire species. However, these results are products of traditional analyses

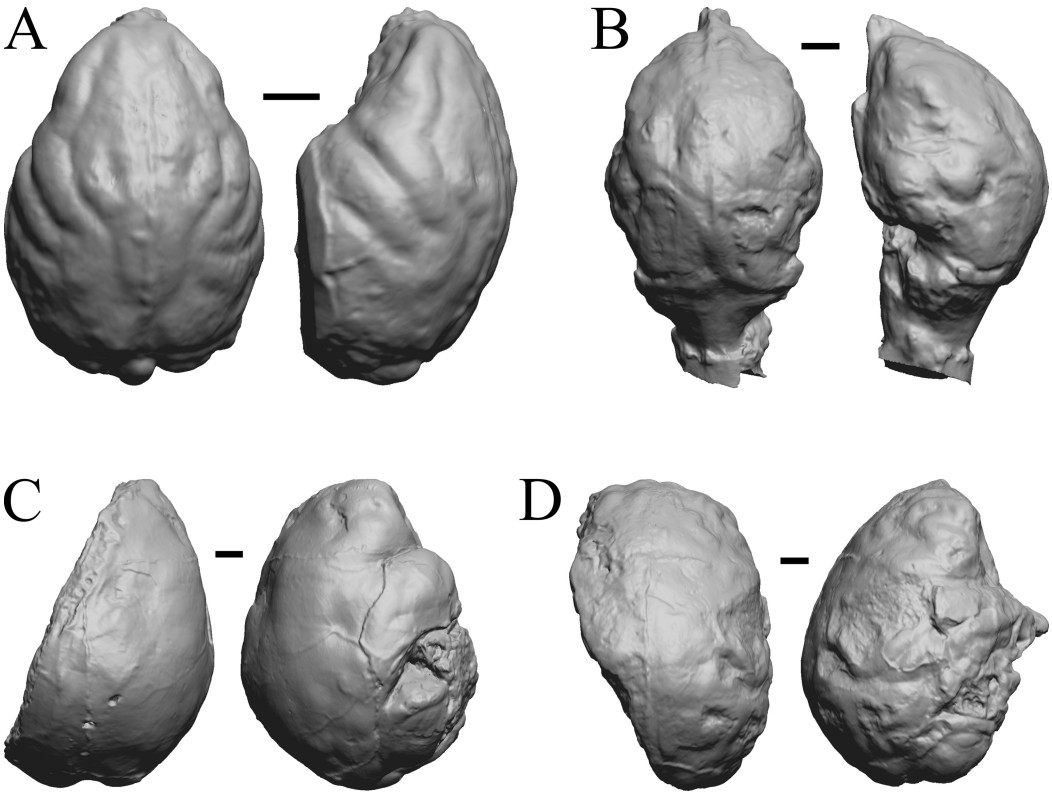

**Figure 30** *Pachylemur insignis, Palaeopropithecus maximus, Australopithecus robustus, Australopithecus africanus* **endocasts.** (A) *Pachylemur insignis* (FMNH PM 59253) in dorsal (left) and left lateral (right) views with rostral pole pointed at the top of the figure. (B) *Palaeopropithecus maximus* (FMNH PM 59250) in dorsal (left) and left lateral (right) views with rostral pole pointed at the top of the figure. (C) *Australopithecus robustus* partial endocast (SK1585) in dorsal (left) and right lateral (right) views with rostral pole pointed at the top of the figure. (D) *Australopithecus africanus* Taung 1 in dorsal (left) and right lateral (right) views with rostral pole pointed at the top of the figure. Scale bars = one cm.

and should only be considered when viewed alongside other notable studies, especially *Bertrand et al., 2022*, for updates and context. We encourage future researchers to revisit these findings with modern statistical methods, as well as potentially remove the La Brea specimens and any incomplete natural endocasts from the fossil dataset, as they are likely too recent in age and skew results. In conclusion, this perspective paper draws on the long history of interpreting endocasts as brains in mammals but exploits their quantitative analysis using digitization technology. Our analysis of brain evolution supports previously published allometric relationships and encephalization patterns in living species, and it provides new trajectories for studying brain evolutionary trajectories, interclass boundaries, and interspecies homogeneity.

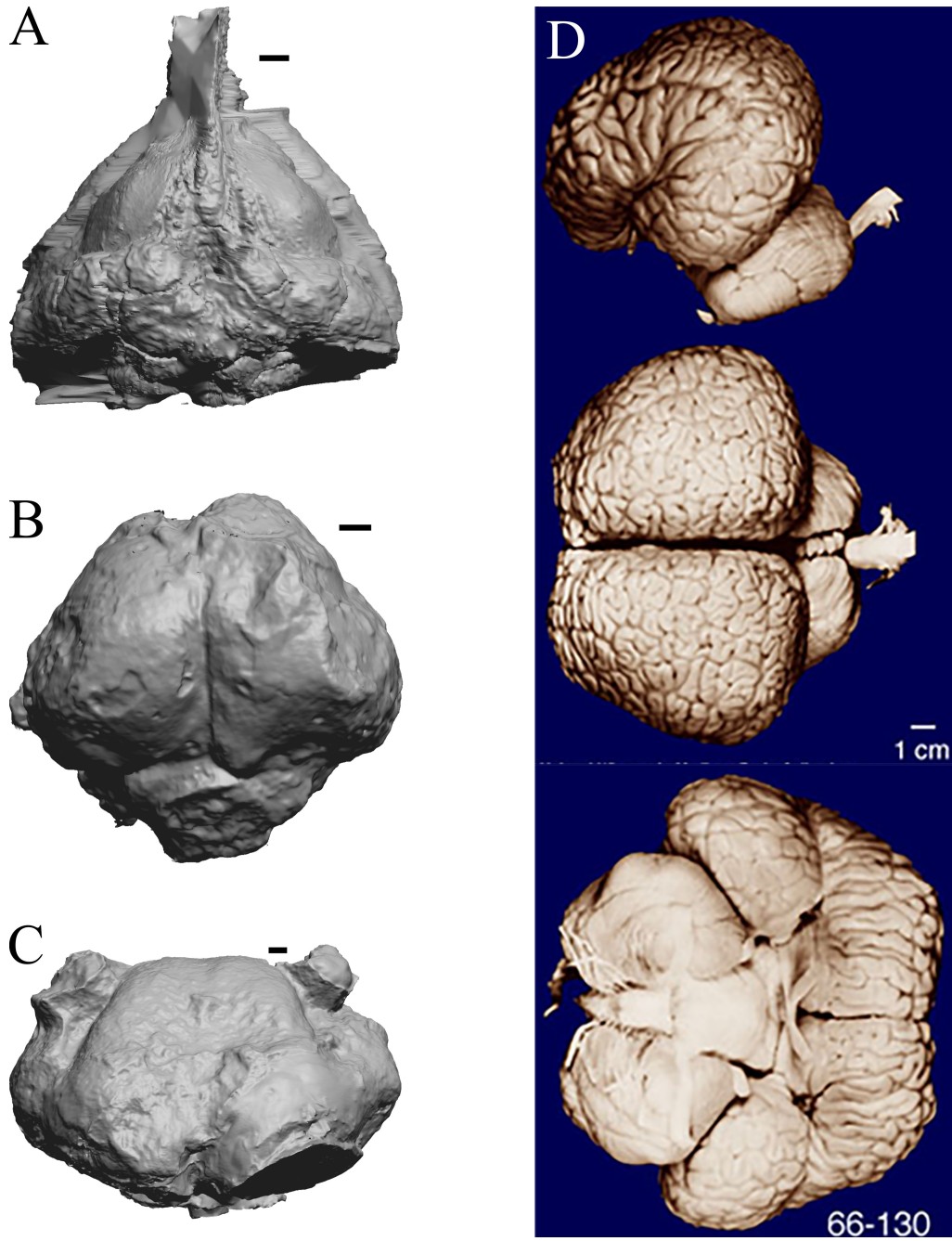

**Figure 31** *Dorudon, Argyrocetus, Aulophyseter, Tursiops.* Endocasts and brain. (A) *Dorudon atrox* endocast (NHMUK PV M 10173 b) in dorsal view with rostral pole pointed at the top of the figure; endocast shows some non-neural material. (B) *Argyrocetus joaquinensis* endocast (USNM 11996) in dorsal view with rostral pole pointed at the top of the figure. (C) *Aulophyseter morricei* endocast (USNM 11230) in dorsal view with rostral pole pointed at the bottom of the figure. (D) Three views of brain of *Tursiops truncates* (NMHM Vertebrates WISC 66-130): top: left lateral view, rostral to left; center: dorsal view, rostral to left; bottom: ventral view, rostral to right. Images in (D) reproduced with permission from http:// brainmuseum.org, with copyright retained by said party. Scale bars = one cm.

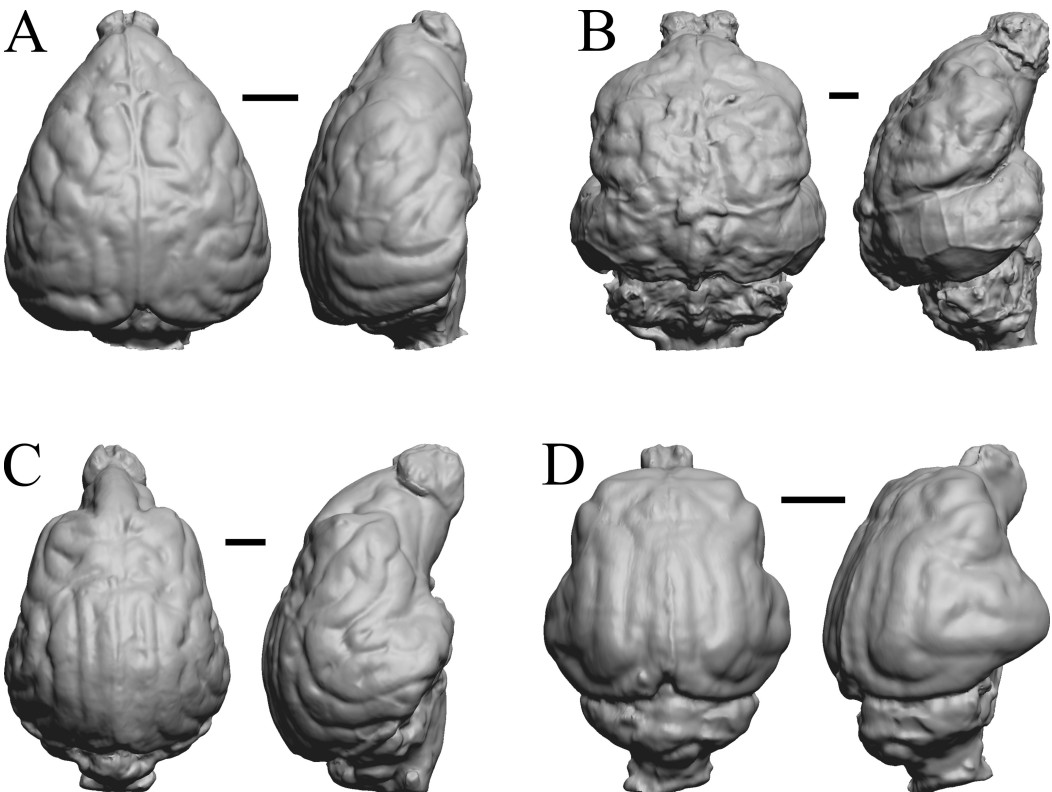

**Figure 32** *Aonyx, Ursus* **(Black Bear)**, *Canis latrans, Felis catus* **endocasts.** (A) *Aonyx (Amblyonyx) cineria* (Radinsky Specimen 358) in dorsal (left) and right lateral (right) views with rostral pole pointed at the top of the figure. (B) *Ursus americanus* in dorsal (left) and right lateral (right) views with rostral pole pointed at the top of the figure. (C) *Canis latrans* (LACMHC 3200-7) in dorsal (left) and right lateral (right) views with rostral pole pointed at the top of the figure. (D) *Felis catus* (FMNH Mammals 146456 = Radinsky Specimen 101) in dorsal (left) and right dorsolateral (right) views with rostral pole pointed at the top of the figure. Scale bars = one cm.

## ACKNOWLEDGEMENTS

Acknowledgments from the perspective of the first author, now deceased, which we have maintained out of respect and admiration for his multiple decades of work on this study: I acknowledge first Bob Martin, Bill Simpson, and the late Bill Turnbull and his widow, Hedy, who made my frequent visits to FMNH a special pleasure. I was often joined by my late wife, Irene, before she succumbed to Alzheimer's. Hedy Turnbull and Anjali Goswami made my last visit especially memorable by spending much time with Irene. Special thanks to Anjali for photographing my specimens at FMNH. The Hanse-Wissenschaftskolleg at Delmenhorst, Germany, where I was a Fellow in 1998, enabled me to buy my laser scanner, and institute members helped me learn to use it. Among other colleagues at museums and institutes at which I worked, I thank Susan Bell at AMNH, Robert Purdy and Mike Brett-Surman at USNM, and Chris Morris at YPM. John Harris at LACM helped with

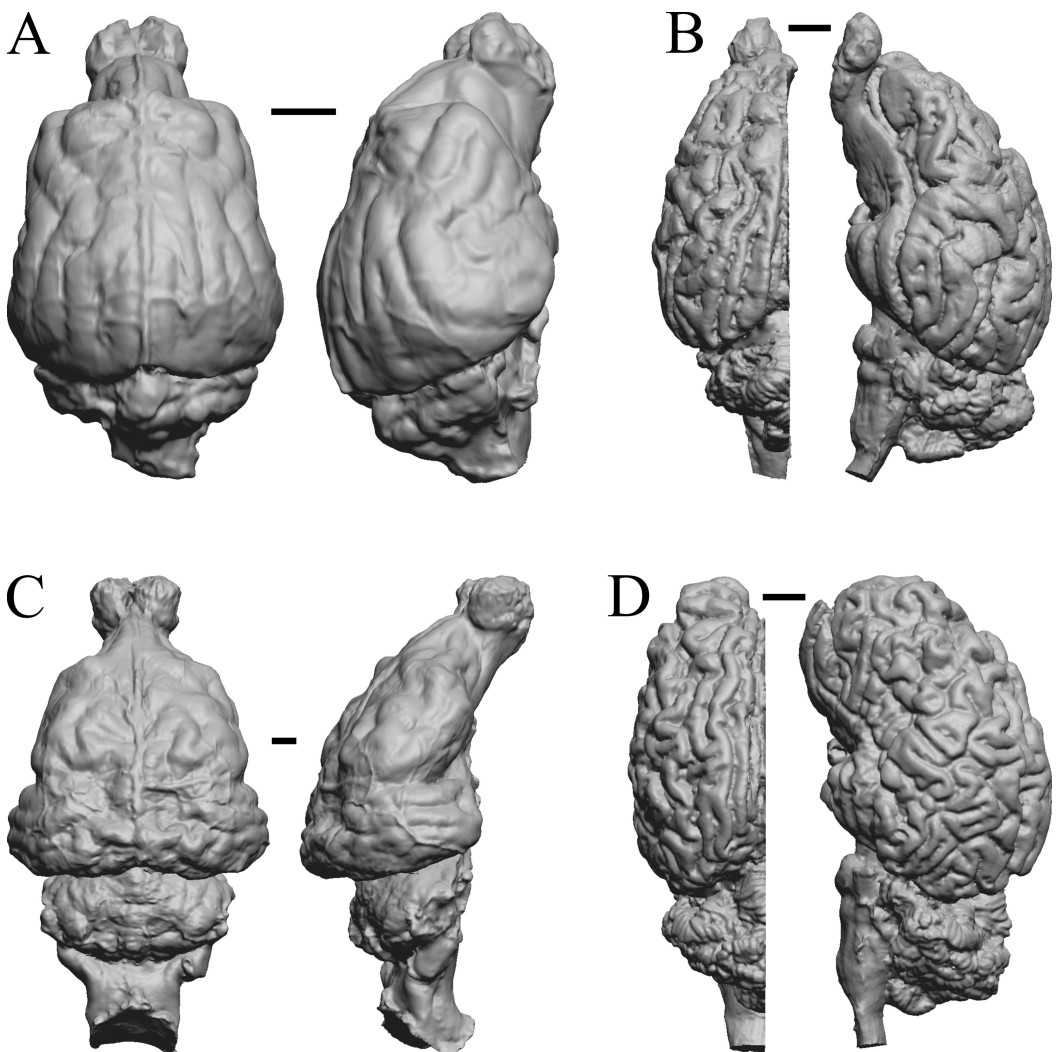

**Figure 33** *Cerdocyon, Odocoileus, Ursus* **(Kodiak),** *Lama.* (A) *Cerdocyon thous* endocast (AMNH Mammals 36501 = FMNH Mammals 146294 = LBR/Rad 294) in dorsal (left) and right lateral (right) views with rostral pole pointed at the top of the figure. (B) *Odocoileus virginianus* braincast (NMHM Vertebrates WISC 67-81) in dorsal (left) and left lateral (right) views with rostral pole pointed at the top of the figure; approximately half of the braincast was available. (C) *Ursus* endocast (possibly LACM Mammals) in dorsal (left) and right lateral (right) views with rostral pole pointed at the top of the figure. (D) *Lama glama* braincast (NMHM Vertebrates WISC 65-139) in dorsal (left) and left lateral (right) views with rostral pole pointed at the top of the figure; approximately half of the braincast was available. Scale bars = one cm.

many specimens. My French colleagues Stéphane Peigné and Thiery Smith assisted with their specimens, and Mark Uhen and Phil Gingerich helped me with whale fossils. I thank Kris Carlson of Witwatersrand for a chimpanzee endocast, and Maria Teresa Dozo, John Flynn, and Denis Croft for help with Neotropical species. Phil Gingerich loaned me the *Pachyaena* endocast that I scanned, and Jack Johnson helped with dolphins and brains in general. Gregg Gunnell checked my geological dating, and Dean Falk helped me with

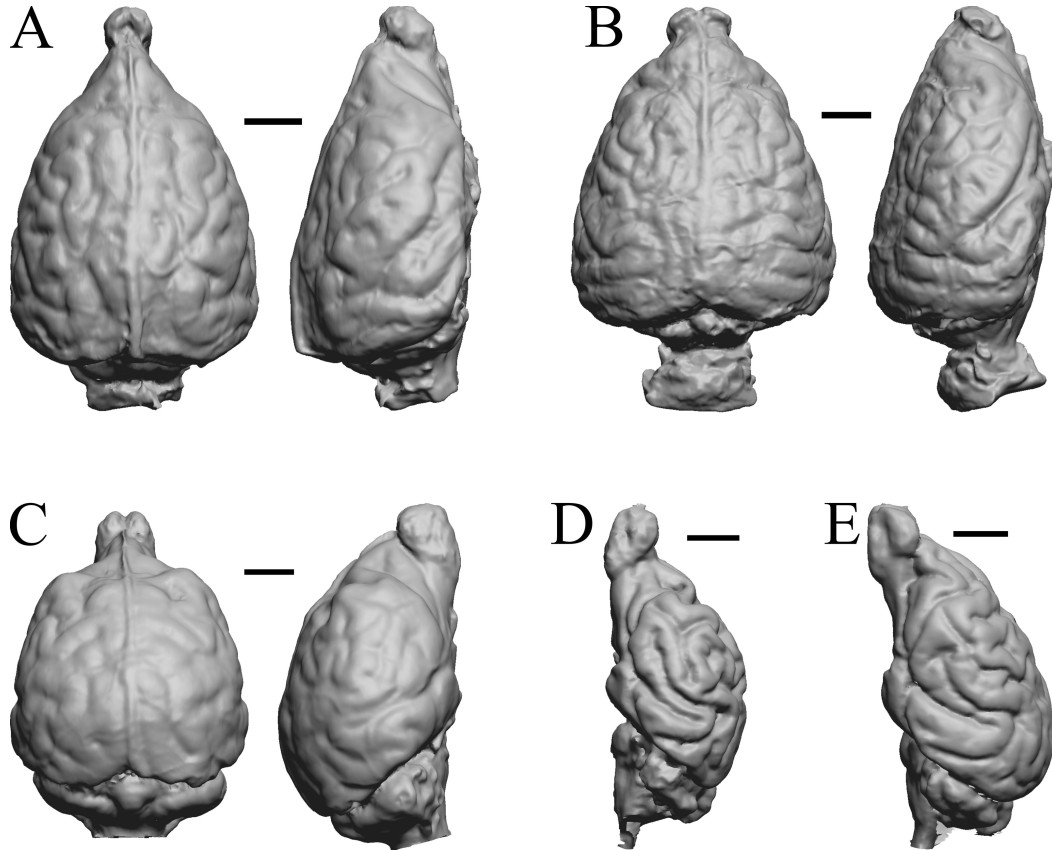

**Figure 34** ***Lutra lutra, Lontra canadensis, Procyon.* endocast and braincast, *Nasua*.** (A) *Lutra lutra* endocast (Radinsky Specimen 366) in dorsal (left) and right lateral (right) views with rostral pole pointed at the top of the figure. (B ) *Lontra canadensis* endocast (FMNH Mammals 146394 = Radinsky Specimen 129) in dorsal (left) and right lateral (right) views with rostral pole pointed at the top of the figure. (C) *Procyon lotor* endocast (FMNH Mammals 146352 = Radinsky Specimen 154 = AMNH Mammals 8335) in dorsal (left) and right lateral (right) views with rostral pole pointed at the top of the figure. (D) *Procyon lotor* braincast (NMHM Vertebrates WISC 61-824) in left lateral (right) view with rostral pole pointed at the top of the figure. (E) *Nasua narica* braincast (NMHM Vertebrates WISC 62-404) in left lateral (right) view with rostral pole pointed at the top of the figure. D and E compare the scans of the raccoon (D) and coati (E) brains. Scale bars = one cm. Further details may be found in Additional Information.

hominid evolution. Xiaming Wang of LACM, Ted Macrini at St. Mary's University, Texas, Blaire Van Valkenburgh of UCLA, and Michel Hofman of the Amsterdam Brain Institute all deserve thanks, along with Marcus Eriksen and Jack Horner, who each let me scan their dinosaur endocasts for Fig 58. My daughter, Elizabeth Jerison Terry, helped in many ways in the preparation of the manuscript and tables, and Phil Dench of HEADUS Computing programmed my software and hardware, and helped me scan the *Tyrannosaurus rex* endocast. Finally, I must thank Liesl Erman who supported me through the difficult final editing of the entire manuscript, its figures and legends. All deserve special thanks. To the

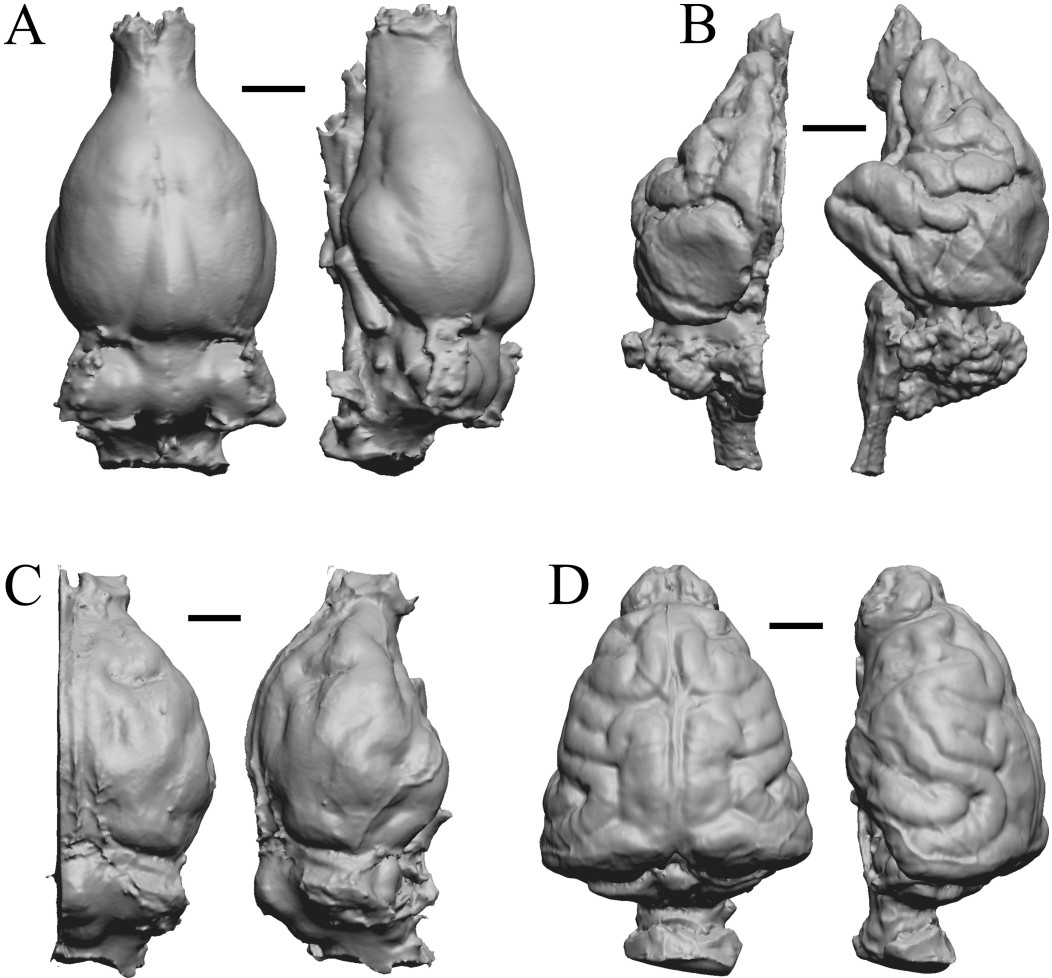

**Figure 35** ***Phascolarctos, Macropus, Vombatus, Taxidea.*** (A) *Phascolarctos cinereus* endocast (Maciej Henneberg Lab, University of Adelaide) in dorsal (left) and left lateral (right) views with rostral pole pointed at the top of the figure. (B) *Macropus fulginosus* left hemisphere braincast (MSU 64023) in dorsal (left) and left lateral (right) views with rostral pole pointed at the top of figure. (C) *Vombatus ursinus* right hemisphere endocast (NMV C7780) in dorsal (left) and right lateral (right) views with rostral pole pointed at the top of the figure. (D) *Taxidea taxus* endocast (Radinsky Specimen 360) in dorsal (left) and left lateral (right) views with rostral pole pointed at the top of the figure. Scale bars = one cm. Further details may be found in Additional Information.

many colleagues who helped on specific emails and whose 'personal communications' I cite, thank you all.

Additional acknowledgments from the other authors: we thank the following collections staff for their help with confirming specimen identifications after our first author's passing: Neil Duncan (American Museum of Natural History—Mammals); Carl Mehling, Jin Meng, and Ruth O'Leary (American Museum of Natural History—Vertebrate Paleontology); Amy Henrici and Matt Lamanna (Carnegie Museum of Natural History—Vertebrate

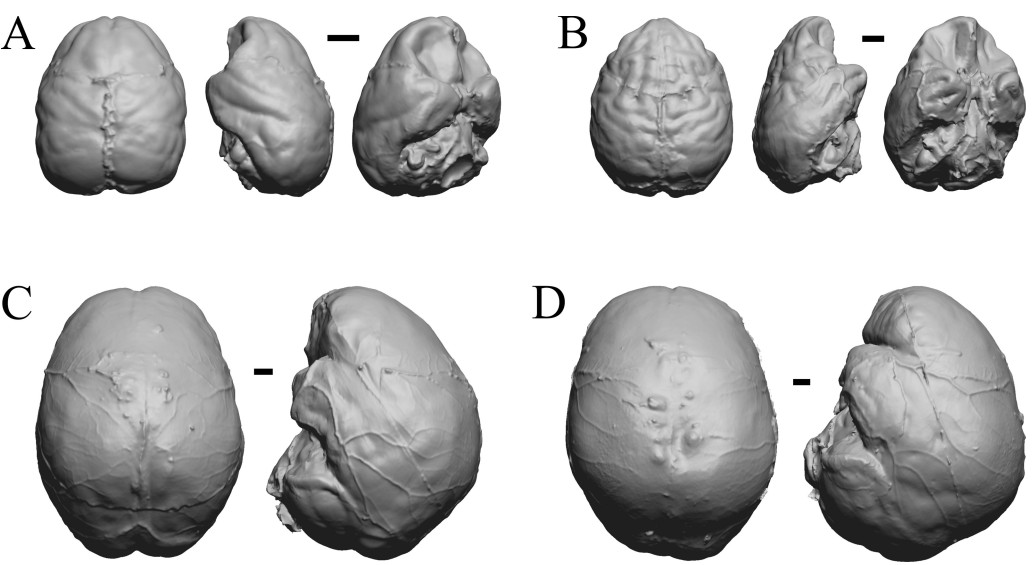

**Figure 36** *Chiropotes*, **Mandrill**, *Homo*-**Falk A**, *Homo*-**Falk B.** Four primate endocasts. (A) *Chiropotes albinansa* (FM 94927) in dorsal (left), left lateral (middle), and right lateroventral (right) views with rostral pole pointed at the top of the figure. (B) *Mandrillus sphinx* (AMNH Mammals 274) in dorsal (left), left lateral (middle), and right lateroventral (right) views with rostral pole pointed at the top of the figure. (C) *Homo sapiens* (Falk A) in dorsal (left) and left lateral (right) views with rostral pole pointed at the top of the figure. (D) *Homo sapiens* (Falk B) in dorsal (left) and left lateral (right) views with rostral pole pointed at the top of the figure. The ventrolateral view exposes more of the rhinal fissure, though it is not easy to trace it in this figure; the fissure is often hidden in more familiar lateral views in primates. Scale bars = one cm. Further details may be found in Additional Information.

Paleontology); Adam Ferguson, Lawrence Heaney, and Kate Webbink (Field Museum of Natural History—Mammals); Bill Simpson and Kate Webbink (Field Museum of Natural History—Vertebrate Paleontology); Guillaume Billet (Muséum national d'Histoire naturelle—Vertebrate Paleontology); Mark Omura (Museum of Comparative Zoology—Mammals); Kevin Rowe (Museums Victoria—Mammals); Archibald Fobbs (National Museum of Health and Medicine); Aisling Farrell and Sam McLeod (Natural History Museum of Los Angeles—Vertebrate Paleontology); Kayce Bell and Shannen Robson (Natural History Museum of Los Angeles—Mammals); Pip Brewer, Christopher Dean, and Rachel Ives (Natural History Museum, London—Vertebrate Paleontology); Rainer Brocke (Senckenberg Gesellschaft für Naturforschung—Palaeontology); Holly Little and Amanda Millhouse (Smithsonian National Museum of Natural History—Paleobiology); Mary-Anne Binne and Steve Donnellan (South Australian Museum—Palaeontology); Matthew Brown (Texas Vertebrate Paleontology Collections); Patricia Holroyd (University of California Museum of Paleontology—Vertebrate Paleontology); Sonia Sequeira and Bernhard Zipfel (University of Witswatersrand); Vanessa Rhue (YPM Vertebrate PaleontologyYale Peabody Museum of Natural History—Vertebrate Paleontology). We thank all who have persisted in their support through several years of complex communication, logistical challenges,

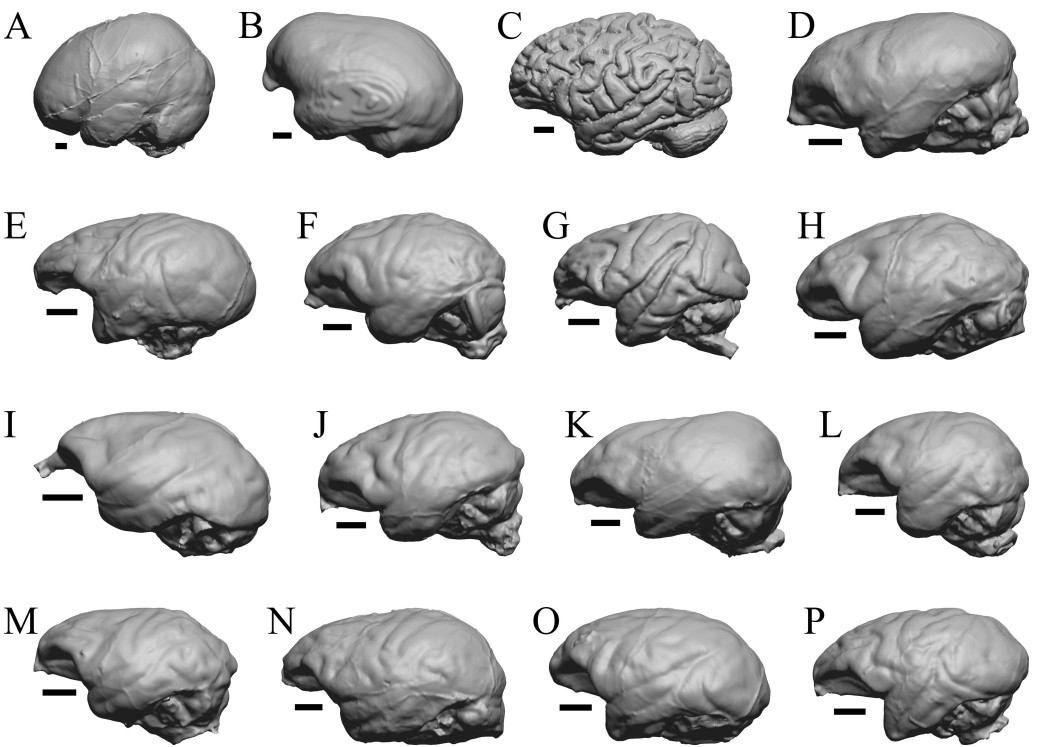

**Figure 37  Primate left hemisphere endocasts and braincasts.** Endocasts and braincasts are in left lateral views. (A) *Homo sapiens* endocast (Falk B). (B) *Pan troglodytes* endocast (unidentified MCZ Mammals specimen). (C) *Pan troglodytes* braincast (NMHM Vertebrates WISC 63-307). (D) *Colobus guereza* endocast (AMNH Mammals 52217). (E) *Erythrocebus patas* endocast (AMNH Mammals 52574). (F) *Hylobates lar* endocast (Falk 386). (G) *Macaca mulatta* braincast (WISC 62-133; 69-307). (H) *Nasalis larvatus* endocast (MCZ Mammals 37328). (I) *Pithecia monachus* endocast (AMNH Mammals 75981). (J) *Presbytis johnii* endocast (AMNH Mammals 54644). (K) *Rhinopithecus (Pygathrix) avunculis* endocast (MCZ Mammals 13681). (L) *Pygathrix nigripes* endocast (AMNH Mammals 69555). (M) *Simias concolor* endocast (AMNH Mammals 103359). (N) *Theropithecus gelada* endocast (FMNH Mammals 8174). (O) *Cercocebus albigena* endocast (AMNH Mammals 52583). (P) *Cercopithecus pygenthus* endocast (AMNH Mammals 52468). Scale bars = one cm. Further details may be found in Additional Information.

a pandemic, and the grief of Dr. Jerison's passing to help usher this study to publication; most especially, we off our deep gratitude and heartfelt sympathies to Harry's daughter, Dr. Elizabeth F. Jerison Terry, and family friend Liesl Erman. Finally, we extend appreciation to our subject editor, Dr. Brandon Hendrick, and our two reviewers, Drs. Ornella Bertrand and Jason Bourke, for their time and effort in supporting and improving this study.

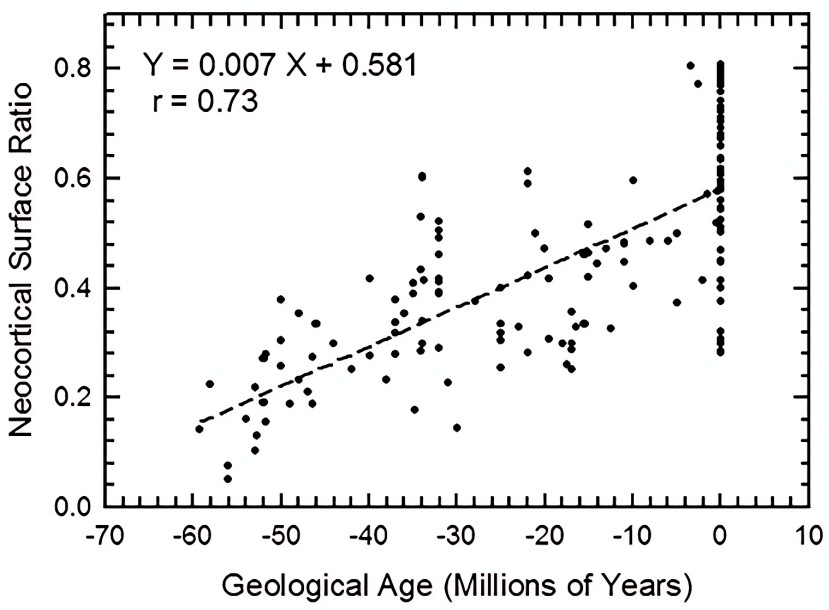

**Figure 38 Neocorticalization and geological age.** Neocorticalization as a function of geological age in 155 scanned specimens from extinct and extant taxa.

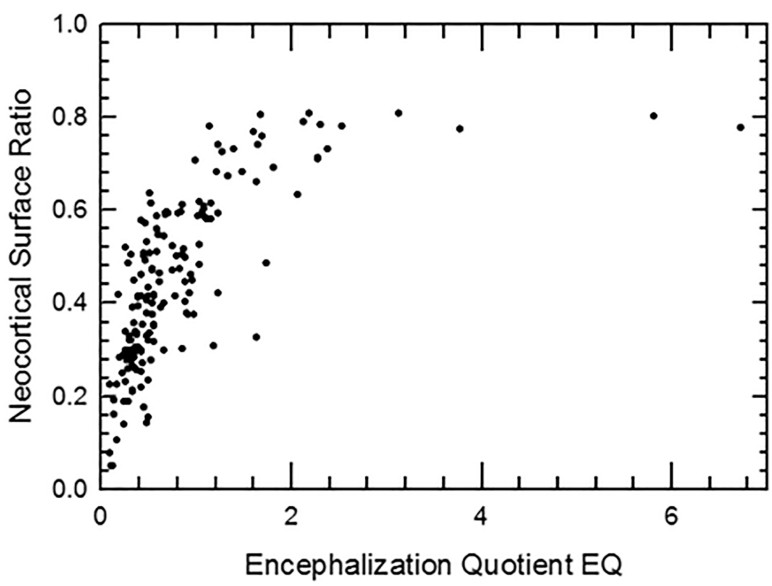

**Figure 39 Neocorticalization and encephalization.** Neocorticalization as a function of encephalization; maximum is about 81%.

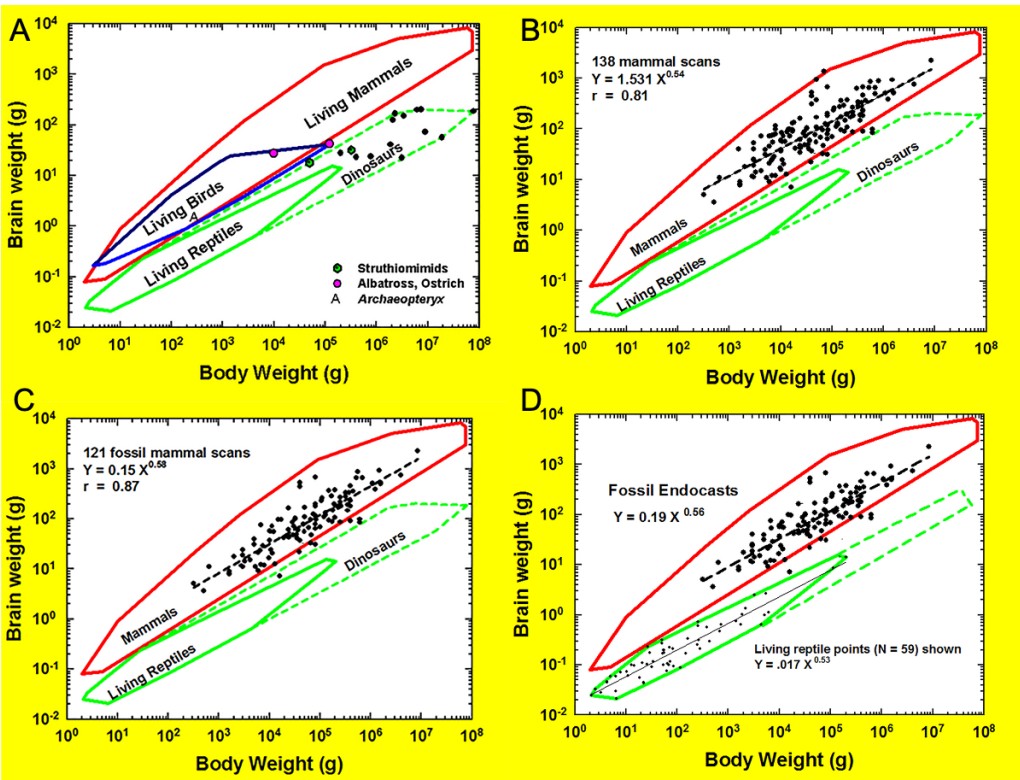

**Figure 40 Amniote allometry.** (A) Brain-body relationships in amniotes. Convex polygons enclose data on living species: mammals ($N = 647$), birds ($N = 219$), and reptiles ($N = 59$). Additional data points for the late Jurassic bird (*Archaeopteryx lithographica*), living albatross (*Diomedia exultans*), living ostrich (*Struthio camelus*), and fifteen non-avian dinosaurs including Struthiomimids, Late Cretaceous "ostrich-dinosaurs" (from (*Jerison, 2007*), by permission). (B) Amniote brain-body polygons with data on 155 scanned mammals and polygons of reptiles, including dinosaurs (see Hopson 1979). (C) Encephalization in 122 fossil mammal species shown within allometry polygons. Amniote brain-body polygons with data on 122 fossil mammals. (D) New reptile polygon. Amniote brain-body polygons with fossil mammal and living reptile data; revised reptile polygon based only on brain size.

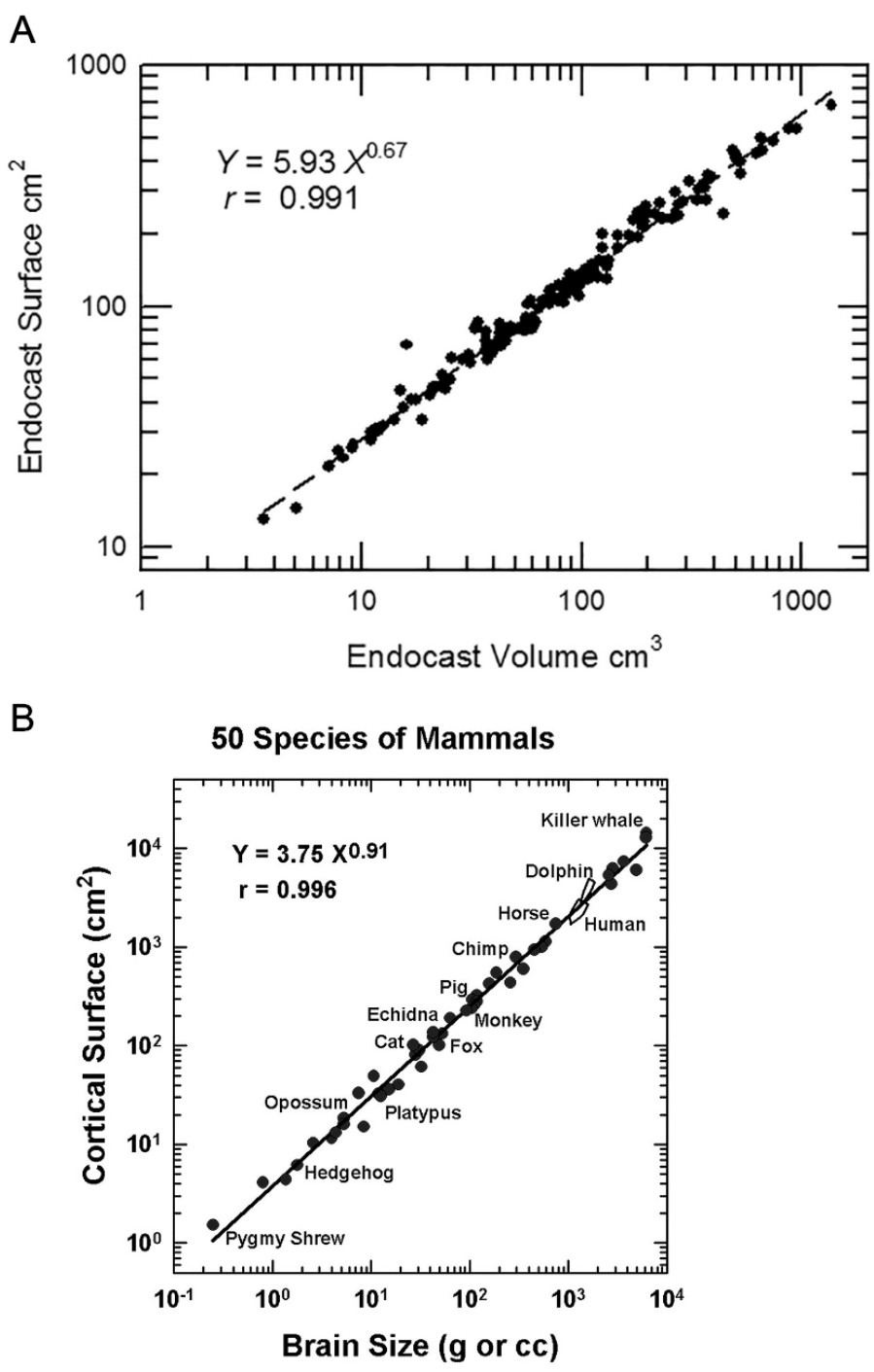

**Figure 41 Surface area–volume relationship in endocasts and living mammals.** Endocast surface area as a function of endocast volume. (B) Cortical surface area (including that buried within sulci and fissures) as a function of brain size in fifty species of living mammals. Correlation: $r = 0.996$. Bivariate regression: $Y = 3.75 X 0.91$. Labeled species indicate the sample diversity. Human and dolphin data are presented as minimum convex polygons enclosing 23 brains for humans and 13 brains for dolphins. (Graph from *Jerison (1991)*, by permission).

### Funding

This work did not receive any funding support.

### Competing Interests

Andrew A. Farke is an Academic Editor and Section Editor for PeerJ.

### Author Contributions

- Harry J. Jerison conceived and designed the experiments, performed the experiments, analyzed the data, prepared figures and/or tables, authored or reviewed drafts of the article, and approved the final draft.
- Catherine M. Early analyzed the data, prepared figures and/or tables, authored or reviewed drafts of the article, updating and processing the figures and scans used in the manuscript, and approved the final draft.
- Andrew A. Farke analyzed the data, prepared figures and/or tables, authored or reviewed drafts of the article, updating the text and nomenclature, revision of taxonomy in paper, and approved the final draft.
- Ashley C. Morhardt analyzed the data, prepared figures and/or tables, authored or reviewed drafts of the article, substantial overhaul the text; updates to nomenclature, revision of taxonomy in paper, incorporation of new research into the overall manuscript, and approved the final draft.

### Data Availability

The raw data are available in Tables 1 and 2 and the Supplementary File.

### Supplemental Information

Supplemental information for this article can be found online at http://dx.doi.org/10.7717/peerj.19826#supplemental-information.

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
