# Peer review of "Digitized endocasts and brains: a perspective on measurements and historical analyses of the evolution of 172 fossil and extant amniote specimens"

_PeerJ, doi:10.7717/peerj.19826_

## Round 0.1 · original submission · Major Revisions

Dear authors,

Thank you for your submission to PeerJ. Based on the comments from two reviewers, the manuscript will require major revisions prior to publication in PeerJ.

Given the figure heavy nature of this paper, I think the main revision that is required is to add annotations to the figures. This would help readers who are experts in neuroanatomy, but also make the paper easier to understand for non-expert scientists. Both reviewers made note of this and though it will constitute quite a bit of work, I think it is important. Also, as both reviewers note, the scale bars are not defined in any figure.

The reviewers have also noted a number of places where the text of the manuscript can be improved.

When you submit your revisions, please include a tracked changes version of your revisions, a clean version of the manuscript, and a response to reviewer document going through each of the individual suggested changes.

Thank you again for your submission. Please let me know if you have any questions.

Best,

Brandon P. Hedrick, Ph.D.

·

Basic reporting

This manuscript represents a culmination of decades of work by the lead author. Whereas the statistical methods used have fallen out of favour over the years (which the authors mention and discuss), the breadth of data provided by the authors is a welcome addition to the neuroanatomical literature. Nonetheless, the manuscript in its current form does require some significant reformatting to effectively convey its information.

There are issues with narrative voice in the manuscript. The authors should decide on either a passive or active voice and maintain this throughout the length of the manuscript.

This is an image-heavy descriptive paper, which places a lot of weight on these images. Most if not all of the images presented have associated scale bars, but no mention in the captions on what the scale bars represent. Similarly, many images show endocasts from two or more views, yet the captions give no mention on what the orientation is for these views. Adding orientation information along with scale bar information would greatly help the figures. Lastly, the image captions show mistakes such as referring to a coloured line or subfigure that is not present (see Additional Comments for specifics). Many of these captions provide far more detail than is needed for the caption. In contrast to these overly descriptive captions, the descriptive text that calls on these figures are often very simplistic, essentially serving only to call on the figures. I suggest the authors swap the captions with the descriptions for better readability.

As a neuroanatomy paper, the intended audience should be well-versed in the general anatomy of the brain. Nonetheless, there are a few instances in the manuscript where more species-specific neuroanatomical terms (e.g., frontal pole) are used. I recommend the authors either direct the reader to a reference for these anatomical terms or even better, add labels to the relevant figures.

The authors break their data down by geological slices. Can we have a reference for the specific age ranges used?

The section comparing the mammal data to reptile data should mention the caveat that the reptile data comes from a substantially smaller sample size (~11x smaller sample size than the mammals and 3x smaller than the bird data).

When referring to extinct animals, especially multi-tonne dinosaurs, I think it is relevant to remind readers of the variability in these mass estimates. I suggest citing Campione and Evans 2020, as these authors specifically tackled variability of body mass estimations for dinosaurs. A section tackling the sensitivity of EQ measurements to this variability would be beneficial to the manuscript.

Experimental design

The only thing I have to suggest here is to let readers know that the laser scanner used was a surface scanner. Possibly mention the limitations of surface scanning and how the authors got around it (much of the workaround is already mentioned in the text, so the limitations portion should segue just fine).

Validity of the findings

no comment

Additional comments

These are a list of specific—but small—issues that should be a quick fix for the authors.

Line 38: Remove “Yet”. Just start sentence with: Endocasts
Lines 42–43: This sentence reads a little awkward. Suggest cutting out the “and in no other vertebrates” part.
Line 46: The rhinal fissure and its relationship to the neocortex is mentioned here, but would benefit from a figure call. I think figure 1 is good for this and has the benefit of not interfering with the figure numbering.
Line 48: Change “phylogenic” to “phylogenetic”
Line 51: Consider removing “However” and just starting the sentence with “Many birds”
Line 56: Shouldn’t the Archimedes’ method read more like the displacement of water volume during immersion? The statement about loss of weight seems a strange way of phrasing it.
Line 74: Replace “such as” with “than”
Line 101: Braincasts are mentioned but never defined. A brief definition of braincast—as was done with endocast—would help here.
Line 108: Might consider using initials rather than first author.
Lines 110 and 115: Capitalize Headus.
Lines 141,143 and 153: Equations are better labeled 2a and 2b instead of 2 and 2a.
Line 175 and Figure 3A: Arrows pointing to the rhinal fissure would help here.
Line 270: Remove nested parenthetical. Can just be: (Jerrison 2007; Fig 40A)
Line 272: Change “beingless” to “being less”
Line 284: I would dump the whole “non-avian” bit entirely here. I don’t think you need to justify a paraphyletic Dinosauria here. It wasn’t justified for reptiles and that didn’t hurt the thrust of the paper. I wouldn’t bother with it for dinosaurs either.
Line 286: The description about Arctocyon having a very brain-like endocast would benefit from a call to figure 6.
Line 308: Can we get a reference for the estimated mass of the dinosaurs here?
Line 327: Would add “in mammals” just after “almost deterministic of brain volume”.
Lines 352–354: Sentence flows strangely. Consider rewording.
Line 359: Change “neocorticalizationacross” to “neocorticalization across”
Line 363: Change “Comparaed” to “Compared”
Line 365: Did you mean “specimen” rather than “species” here?
Lines 390–391: Can you reword the last sentence. It reads weird (constraints on their interpretations as brains?)
Line 404: Weight should not be interchangeable with volume unless we are referring to a proxy for weight (i.e., assuming that neural material is of equal density in all species), in which case, it would be good to add this qualifier.
Lines 438–440: This last sentence is difficult to understand. Can the authors please rephrase it?
Lines 450–459: This concept would benefit from a figure.


Figure 1: Image quality is pretty low. Can the authors provide a higher DPI figure here? Caveat: this could be an issue with the reviewer image quality provided by PeerJ
Figure 2: Needs information on orientation of each image.
Figure 3: The caption refers to a green line, but all the lines in the figure are black.
Figure 4: Caption refers to an image D, but there is no image D in the figure.
Figure7: Caption needs to describe views of A and B.
Figures 8–10: Need to add description of views used in images
Figure 17: Please reword section to remove the first-person narrative part.
Figure 24: Caption mentions the likelihood of significant overestimate to endocast size but doesn’t explain why (also, see section in Basic Reporting about figure captions).
Figure 30: Mentions frontal pole, which might throw off some readers. Suggest adding labels in this instance.
Figure 41: Need accession numbers for the dinosaurs
Figure 41a: Show lines used for brain sectioning described in lines 310–313.

·

Basic reporting

-English is clear, paper is well-written.
-More references could be added. Please see attached review.
-Figures are nice and tables are detailed but the authors should make sure that the digitized endocasts are put shared in the repository such as Morphosource. Figures should be labelled, and scale bar added. Please see attached review for more details.
-Yes the paper is self-contained, relevant results to hypotheses present.

Experimental design

-Paper represents an important contribution to Paleoneurology.
-Research question is clear and filll a gap in the field of Paleoneurology
-Statistics are robust.
-Methods are good but some details should be added such as how the neocortex and olfactory bulb size were obtained. Please see attached review for more details.

Validity of the findings

-All data are provided to reproduce the analyses. The digitized endocasts should be put in a repository.
-Conclusions are representative of the findings.
-A limitation section should be added to the study. Please see attached review for more details.

Additional comments

From what I can read, the authors are not allowed to publish the images from http://neurosciencelibrary.org/index.html because of copyright issues.

Please see attached review for additional suggestions.

---

## Round 0.2 · Minor Revisions

Dear authors,

Thank you for your submission to PeerJ. Based on two reviews, I feel that this paper will be publishable in PeerJ following minor revisions. One reviewer noted substantial advancements in the field from the time that this paper was initially reviewed and now.

I understand the desire to publish this as a tribute to Jemison and appreciate that methods such as PCMs need not be employed. However, echoing reviewer comments, I do think it is more important to present the results that are novel in light of the caveats that are variously found throughout the paper. As such, I feel like this paper might be better presented as a perspective manuscript given the difficulty in incorporating new data and PCMs. The paper is written like this in many ways already so I don't think this will require substantial rewrites. Framing it this way will clarify to readers who may be new to the field that this paper is meant to stimulate ideas, consolidate data, and function as a tribute rather than act as a primary research manuscript.

A few additional small comments:
Line 82-84: I guess I would argue that PCMs are no longer a particularly new idea. You might reword this away from it being a new thing that was not done in favor of the other reasons mentioned.
Line 92: missing period
Line 196–198: Do you have citations supporting your third reason?
Line 427: species?
Line 476–478: This is question specific. Relatively high within species variability may be important at a smaller taxonomic level (e.g., comparing within family), but perhaps is less relevant across mammalian orders or Mammalia as a whole.

Best,

Brandon P. Hedrick, Ph.D.

·

Basic reporting

I'm glad to see the return of this massive testament to Dr. Jerison's work. The authors did a great job on the rewording, adjusting and trimming of the original manuscript. This revision now reads more concise while still retaining its massive amount of information.

Experimental design

My initial critiques were all successfully addressed by the authors. I believe that this revision now reads cleaner and tighter.

Validity of the findings

Findings are all appropriately addressed in the text and appear to stay well within the confines of the data.

Additional comments

It is a shame that Dr. Jerison is no longer with us. However, this posthumous, final publication should serve as a lovely and informative tribute to his work in paleoneurology.

·

Basic reporting

Review of the manuscript entitled: “Digitized endocasts and brains: measurements and analyses of the evolution of 172 fossil and extant amniote specimens”
Thank you for giving a new opportunity to review this manuscript and thank you to the authors for incorporating the majority of my suggestions. Because it was first reviewed in 2021, I hope you can understand that I have new comments related to articles that have been published in the last 4 years and that are relevant to different aspects of the paper. There is quite a bit of areas in text that need to be better justified or clarified. My comments below are meant to be constructive feedback.
I understand the authors want to compare the graphs here with the data that were generated using the same methods by Jerison. However, this is highly problematic because you are not incorporating newly published data including all of the virtual endocasts of fossils ever published (from what I understand the sample is) and not keeping up with current hypotheses about brain evolution. For example, the current paper does not have many Paleocene mammals and therefore fails to see the decrease in relative brain size found by Bertrand et al. (2022).
So, I think it is important to better frame this paper with this in mind. It is crucial to better justify throughout the manuscript, abstract and conclusion, what aspects of the papers are providing 1) truly novel results, 2) what aspects are supporting previous results, 3) acknowledge that virtual endocasts are not included and therefore some of the conclusions on the quantitative analyses here might be outdated and 4) the fact that many of the natural endocasts cannot be included in future quantitative analyses because of incompleteness.
Something that should be put more forward in this paper is the fact that natural endocasts can be great for morphological descriptions and identifying important features such as the exposure of the midbrain including the colliculi as well as sulci pattern on gyrencephalic brains.
I am sorry, but I will not be able to look in the supplementary data, figures and table legend in depth because of time constraints.
Detailed comments: The line numbers correspond to the PDF version of the manuscript
The abstract needs to be updated to include more recent work. An increase in neocortical surface area that occurred independently in mammals throughout the Paleocene to Eocene has been published more recently (Bertrand et al., 2022, see below). It would be more accurate to write “as previously reported, on average, neocorticalization of mammals increased over time…”
Bertrand, O. C. et al. Brawn before brains in placental mammals after the end-Cretaceous extinction. Science 376, 80-85 (2022).
Lines 58-60: “Encephalization, the evolutionary increase in brain complexity or relative size reflecting environmental adaptations, is … independent of their phylogenetic details”
This sentence needs clarification. As it stands, I would not agree with that especially in mammals. There are clear significant differences between the slope and intercept of various mammalian clades today. Please see Smaer et al. (2021). Bruger et al. (2019) proposed different equations for various mammalian clades.
Smaers, J. B. et al. The evolution of mammalian brain size. Sci. Adv. 7, eabe2101, doi:10.1126/sciadv.abe2101 (2021).
Burger, J. R., George, M. A., Jr., Leadbetter, C. & Shaikh, F. The allometry of brain size in mammals. J. Mammal. 100, 276-283, doi:https://doi.org/10.1093/jmammal/gyz043 (2019).
Lines 62-63: Barton and Harvey (2000) are proponent of the mosaic model of evolution and the concerted so I am not sure why this publication is cited here. The work of Finlay would be more appropriate here (Finlay and Darlington, 1995 for example but there are others).
For example: Finlay, B. & Darlington, R. Linked regularities in the development and evolution of mammalian brains. Science 268, 1578-1584, doi:10.1126/science.7777856 (1995).
Lines 73-75: I think this is an important goal and I agree with the authors, but it would be good to mention previous work that has starting doing so using virtual endocasts of fossils from the Paleocene and Eocene (see Bertrand et al., 2022).
Lines 78-79 and Lines 84-86: Please cite and discuss the results from Bertrand et al. (2022).
Line 76: I would add that there are limitations in using natural endocasts from fossils because sometimes regions are not well preserved and broken. It would be good to justify how the authors dealt with this issue specifically in the statistical analyses. It is not clear if all of the specimens here were used in the quantitative analyses or only the ones that are well preserved.
Lines 137-138: Again, a lot of specimens are incomplete. Please specify which ones were measured and which ones were too damaged.
Lines 148-149: Do you mean that it is not visible in cetaceans? Please clarify this sentence.
Lines 177-178: Just something to clarify, did the authors used the EQ equation from Jerison (1973)? If that’s the case, it is important to acknowledge that this equation was done on a very limited sample compared to the equations that have been produced more recently in Burger et al. (2019; all mammals) and Lopez-Torres et al., (2023; Eurachontoglires). Why not sure a more updated equation like the one from Burger et al (2019) for all mammals?
Burger, J. R., George, M. A., Jr., Leadbetter, C. & Shaikh, F. The allometry of brain size in mammals. J. Mammal. 100, 276-283, doi:https://doi.org/10.1093/jmammal/gyz043 (2019).
López-Torres, S. et al. The allometry of brain size in Euarchontoglires: clade-specific patterns and their impact on encephalization quotients. J. Mammal. 105, 1430-1445, doi:10.1093/jmammal/gyae084 (2024).
Lines 187-188: I am not sure that I understand what that means. Could you please clarify?
Lines 188-189: Could you provide an example for point 3? I am not sure how this can be considered more recent to measure brain and brain region sizes in 2D.
Lines 194-196: I am not sure I see a need to do that when we can scan and segment complete virtual endocasts. Those different natural endocasts are damaged and therefore are problematic to include in quantitative analyses.
Line 198: I am not sure that it is fair to say that this work would represent a foundation for future studies when so much as been done since 1973 in the field of paleoneurology regarding ideas and discussions. If the authors want to keep this part, please make a clearer case for it in the paper. Please also consider reading the recently published book by Dozo et al. (2023). There is a wealth of information about paleoneurology and should be cited in the introduction.
Dozo MT, Paulina-Carabajal A, Macrini TE, Walsh S (2023) Paleoneurology of amniotes: New directions in the study of fossil endocasts, Cham: Springer.
Line 206: Please also cite Bertrand et al. (2022), we looked at relative brain size and the size of brain regions through time in mammals.
Bertrand, O. C. et al. Brawn before brains in placental mammals after the end-Cretaceous extinction. Science 376, 80-85 (2022).
Lines 298-299: Relative brain size has been shown now to not be a good way to estimate “intelligence”. Even the term “intelligence” should be avoided. I would say that it is quite an outdated idea and an oversimplification of behaviour. See van Schaik et al. (2021) and add a few lines about this aspect in the manuscript.
van Schaik, C. P., Triki, Z., Bshary, R. & Heldstab, S. A. A farewell to the encephalization quotient: A new brain size measure for comparative primate cognition. Brain Behav. Evol. 96, 1-12, doi:10.1159/000517013 (2021).
Lines 305-307: Could you please rephrase this sentence? The meaning is not clear.
Lines 307-309: Do you mean changes in cortical organization? I would modify the sentence so it is clearer.
Lines 309-311: There have been papers showing a link between ecology and brain region size using endocasts and the fossil record. Admittedly not many but some. I am not trying to just cite my papers, there is just not so many that have been doing this yet. We found that there is a correlation between arboreality and the increase in the size of the neocortex and petrosal lobules when squirrels transition to an arboreal lifestyle using extant and extinct taxa. We also looked at the influence of diet in the Trogosus paper and auditory capabilities in the Incamys paper. This is just to provide some background on the topic.
Bertrand, O. C., Püschel, H. P., Schwab, J. A., Silcox, M. T. & Brusatte, S. L. The impact of locomotion on the brain evolution of squirrels and close relatives. Commun. Biol. 4, 1-15, doi:https://doi.org/10.1038/s42003-021-01887-8 (2021).
Bertrand, O. C. et al. The virtual brain endocast of Trogosus (Mammalia, Tillodontia) and its relevance in understanding the extinction of archaic placental mammals. J. Anat. 244, 1-21, doi:https://doi.org/10.1111/joa.13951 (2024).
Bertrand, O. C. et al. The virtual brain endocast of Incamys bolivianus: Insight from the neurosensory system into the adaptive radiation of south American rodents. Pap. Palaeontol. 10, e1562, doi:https://doi.org/10.1002/spp2.1562 (2024).
Lines 312-313: If I recall, there is potentially an issue in the way the brain was preserved for the koala in that original publication and it may have shrunken post-mortem before they measured the brain. There are more recent images published in Taylor et al. (2006) that show that the brain is against the endocranial cavity (Fig. 1).
Taylor, J., Brown, G., De Miguel, C., Henneberg, M. & Rühli, F. J. MR imaging of brain morphology, vascularisation and encephalization in the koala. Australian Mammalogy 28, 243-247 (2006).
Lines 313-315: Please add a reference for this sentence.
Lines 327-329: I would rephrase this sentence. How variation in brain regions demonstrates why brain as a whole evolved? Maybe flesh out this idea a bit more to clarify what you mean.
Lines 332-333: A reference is needed here for this statement. I was able to compile quite a bit of olfactory bulb data throughout the years on fossil endocasts (see for example Bertrand et al., 2022).
I think it would be fairer to say that it is an issue for natural endocasts but not for virtual endocasts. The circular fissure is a narrow point and it can break easily, therefore the olfactory bulbs are not always preserved in natural endocasts.
Lines 341-342: Enlarged colliculi do not automatically mean that an animal has echolocation. It just means that it potentially uses more the midbrain to survive than the visual and auditory cortices. Additionally, please specify if the inferior or the superior colliculi are enlarged. We published on this in the Incamys paper (Bertrand et al., 2024) and found that the inferior colliculi were possibly enlarged in that Oligocene rodent. In squirrels, the superior colliculi are enlarged instead. Also, it is important to keep in mind that it is difficult to design experiment that would show if indeed koalas use their vision or audition a lot. We do not have the same “Umwelt” as koalas, so we might not see how they would use these functions yet.
Lines 352-363: These results should be compared with Bertrand et al. (2022). We measured the neocortical size of Arctocyon on a virtual endocast and found 10.3% (Table S1) but this is in relation to the whole endocast (including the olfactory bulbs). We also found that some carnivoramorphs also had a relatively big neocortex. Feel free to check Table S1 for making comparisons with the current work. We also found an increase in the relative size of the neocortex through time (Fig. 3C, F, I of the same paper) with some interesting differences between stem taxa and crown clades of the Eocene. We also look at the middle Eocene specifically in the Trogosus paper (Bertrand et al., 2024).
Lines 402-403: This point makes me think of one important finding from Bertrand et al. (2022) is that we found a decrease in relative brain size in the Paleocene in comparison to the Mesozoic. I think it would good to discuss this point and how this affects the conclusion of the current study.
Lines 421-423: I would encourage you to check Caspar et al. (2024) for the part about which rule dinosaurs follow. This paper can be discussed in relation to the results here.
Caspar, K. R. et al. How smart was ? Testing claims of exceptional cognition in dinosaurs and the application of neuron count estimates in palaeontological research. Anat. Rec. 307, 3685-3716, doi:https://doi.org/10.1002/ar.25459 (2024).
Lines 454-457: For this particular point, you could check the very insightful paper from Boch et al. (2024). They look at sulci pattern and foraging behaviour in this group. This paper could be discussed in relation to the statement in the present paper.
Boch M, Karadachka K, Loh KK, et al. (2024) Comparative neuroimaging of the carnivoran brain: Neocortical sulcal anatomy.). eLife Sciences Publications, Ltd.
Lines 458-459: There are other regions that can be measured: the olfactory bulbs and the petrosal lobules with great accuracy. More broadly, there is quite a bit in the literature showing the impact of ecology on diverse brain regions. Check De Casien et al. (2019) for example. They look at diverse brain regions and the link to ecology and behaviour.
DeCasien AR, Higham JP (2019) Primate mosaic brain evolution reflects selection on sensory and cognitive specialization. Nat. Ecol. Evol., 3, 1483-1493.
Lines 467-468: I am not sure that I understand this sentence, why could researchers not study neocorticalization and brain surface areas before digitization?
Lines 481-486: Please include a few sentences on how this compares to the results from Bertrand et al. (2022). This is true that we did not include extant species, but we had 137 fossil taxa, which is a few more than the current study if the 172 endocasts include extant species.
Lines 489-490: I would add that it is important to make sure that future researcher keep in mind that many of these natural endocasts are incomplete (missing part of the olfactory bulbs) and a lot of them cannot be incorporated with data of virtual endocasts.

Yours sincerely,
Ornella Bertrand

Experimental design

See above (1)

Validity of the findings

See above (1)

Additional comments

See above (1)

---

## Round 0.3 · accepted · Accept

Dear authors,

Thank you for your careful attention to reviewer comments. I think that this paper is quite strong now and agree with your changing it to a perspective. I just had a few tiny type-editing comments that can be fixed in the proof process:

Line 199: space “data)in”
Line 203: “intelligence)and”
Line 326: delete “=”
Line 505: double period
Line 516: grammar

Thank you for your submission to PeerJ.

Best,

Brandon P. Hedrick